cognition/psychology/behaviour

language evolution, multimodality, gesture, vocalization, interaction

**Author for correspondence:**
Vinicius Macuch Silva
e-mail: vinicius.macuch.silva@uni-osnabrueck.de

# Multimodality and the origin of a novel communication system in face-to-face interaction

Vinicius Macuch Silva[1], Judith Holler[2,3], Asli Ozyurek[2,3,4] and Seán G. Roberts[5]

[1]Institute of Cognitive Science, Osnabrück University, Osnabrück, Germany
[2]Max Planck Institute for Psycholinguistics, Nijmegen, The Netherlands
[3]Donders Institute for Brain, Cognition and Behaviour, Radboud University Nijmegen, Nijmegen, The Netherlands
[4]Center for Language Studies, Radboud University Nijmegen, Nijmegen, The Netherlands
[5]Department of Archaeology and Anthropology (excd.lab), University of Bristol, Bristol, UK

VM, 0000-0002-3370-4157; JH, 0000-0003-0671-6651; SGR, 0000-0001-5990-9161

Face-to-face communication is multimodal at its core: it consists of a combination of vocal and visual signalling. However, current evidence suggests that, in the absence of an established communication system, visual signalling, especially in the form of visible gesture, is a more powerful form of communication than vocalization and therefore likely to have played a primary role in the emergence of human language. This argument is based on experimental evidence of how vocal and visual modalities (i.e. gesture) are employed to communicate about familiar concepts when participants cannot use their existing languages. To investigate this further, we introduce an experiment where pairs of participants performed a referential communication task in which they described unfamiliar stimuli in order to reduce reliance on conventional signals. Visual and auditory stimuli were described in three conditions: using visible gestures only, using non-linguistic vocalizations only and given the option to use both (multimodal communication). The results suggest that even in the absence of conventional signals, gesture is a more powerful mode of communication compared with vocalization, but that there are also advantages to multimodality compared to using gesture alone. Participants with an option to produce multimodal signals had comparable accuracy to those using only gesture, but gained an efficiency advantage. The analysis of the interactions between participants showed that interactants developed novel communication systems for unfamiliar stimuli by deploying different modalities flexibly to suit their needs and by taking advantage of multimodality when required.

# 1. Introduction

Theories of language origins and evolution have been polarized in terms of speech or gesture first accounts of early human communication. More recently, multimodal accounts placing emphasis on both speech and gesture have entered the theoretical landscape of linguistic evolutionary research. Support for such multimodal accounts of language evolution stems primarily from the understanding of modern linguistic behaviour: speech and gesture play important roles in the acquisition, processing and situated use of language. However, despite extensive investigation of how vocal and visible bodily behaviour are combined in modern language, one dimension relating to multimodal communicative behaviour remains little explored: how is it that vocal and gestural signals might come together in establishing communication from the ground up and creating language anew? Focusing on such an evolutionary dimension of language, we report a laboratory experiment testing the extent to which vocal and gestural signals can be used to develop a communication system in the absence of prior communicative conventions, and use rigorous statistical methods to analyse the results. In addition to quantifying communication in terms of both descriptive accuracy and efficiency, we also explore when multimodal signals are used, and how the different modalities within multimodal signals relate to each other in interaction

## 1.1. Modality and the origins of language

One of the central issues in the field of language evolution is the role of modality in the emergence of human communication. There are two possible extreme positions, one stressing the importance of manual action and visible gesture, the other defending the centrality of speech and vocal communication [1]. The former suggests that gesture preceded and led to the origin of spoken language (e.g. [2–5]). The so-called 'gesture-first' theories of language evolution have gained mounting popularity in recent years, mostly due to advances in comparative and neurobiological research which highlight the ties between manual action and visual-gestural signalling in both humans and non-human primates [6].

Evidence that language can emerge through visual-gestural communication can be found in modern-day sign systems. Indeed, homesign, impromptu communication systems created by deaf children born to hearing parents, exemplify how gestural communication, mostly iconic in nature, can give rise to systematically structured communication systems. Goldin-Meadow *et al.* [7], for instance, showed that deaf children can establish new communicative conventions by spontaneously communicating with their hearing parents. Interestingly, not only are children able to create new inventories of form-meaning associations out of hand gestures, these inventories also develop combinatorial structure both at the lexical and syntactic level. In other words, sets of motivated gestural displays created by deaf children can adopt systematic structure akin to that of language over repeated use in situated communication. In fact, emerging sign languages are living proof that, given sufficient use and transmission, unconventionalized sign systems can evolve and acquire increasingly more complex linguistic structure (e.g. [8,9]). However, it is unclear how to connect evidence that visible gesture can lead to the emergence of systematic communication systems to the origins of language and the move from gesture to vocal communication in the evolution of language.

On the other side of the spectrum are theories which suggest that, since vocal signalling is the main form of conventionalized communication across modern human populations, language must have been realized in the vocal modality from its very onset. Darwin argued that the widespread abilities to produce and perceive sounds across primate species pointed to a central role of acoustic signals and that 'musical sounds afforded one of the bases for the development of language' [10, p. 572]. More recent research on musical protolanguage continues to emphasize the importance of vocalization [11–13]. There is increasing evidence for flexible use of vocal calls in primates (e.g. [14–16]), and research showing that it may be easier to recognize and treat vocalizations as symbolic signals compared to gestures [17–19].

Bridging the gap between the two opposing sides is another approach to the emergence of language, one which explicitly acknowledges that modern human communication is multimodal, rather than consisting of speech only. Multimodal theories of language evolution have as their central premise the fact that both vocal and gestural modes of communication are core to modern human communication and its evolution (e.g. [20–24]). Evidence in support of such theories is drawn from modern human communicative behaviour (see section below), as well as from neurobiological findings which point to

the deep connection between vocal and visual behaviour not only in modern humans but possibly too in other species, both extant and extinct, in the phylogenetic line leading to hominins (e.g. [20,24–27]). Moreover, as highlighted by ethological and comparative research, humans are not alone in being multimodal communicators: non-human apes as well as other primates seem to combine both vocal and gestural signals in their communication ([28–32]; see also [33]). In general, signals used in important contexts such as warning or mating are often multimodal in order to be robust [34–38]. In this light, it seems quite plausible that early human communication was inherently multimodal in early stages of evolution, just as it is today. Of course, there are questions about whether the ability to *learn* vocal signals was present for our ancestors before the emergence of symbolic language (e.g. [39]). However, Perlman [24] argues that there is at least some evidence of vocal learning in non-human primates, and that this might plausibly allow a fully multimodal system right from the start (see also [21,27,40]).

We should note that, although 'gesture-first' and 'speech-first' views are often pitted against each other, in reality many suggest a middle ground. For example, Arbib [41] suggests that gesture was important at a very early stage of language evolution because it provided a scaffolding for the evolution of vocal learning, but that communication would have been multimodal as soon as an ability to learn vocalizations was in place. Similarly, Fay *et al.* [42] interpret the findings of their experiment to support a multimodal origin of language. However, we note that the polarization has crept back into the debate [24] and we discuss potential reasons for this below. Our current experiment aims to add further empirical data to this debate, in the hope of reclaiming the middle ground.

## 1.2. Modern human communicative behaviour

Multimodal theories of language evolution are supported primarily by evidence from modern human communicative behaviour. In face-to-face scenarios, people communicate by drawing not only on their voice but also on visible signals produced with bodily articulators such as the face and the hands. Situated human communication can thus be characterized as a multimodal form of signalling, one which involves the co-articulation of vocalizations and gesture.

Many accounts of language use in spontaneous interaction underscore such a multimodal character of linguistic communication (e.g. [43,44]). Naturalistic research conducted in the field has shown that people of different cultural and linguistic backgrounds employ verbal as well as visual signals when communicating to each other in everyday contexts (e.g. [44–46]), and it has been shown that gesture plays a key role in spoken language acquisition (e.g. [47,48]), for example, by scaffolding early word learning (e.g. [49]). Bohn *et al.* [50] found that 24-month-olds could understand multimodal iconic signals, but not signals that were unimodal (vocal or gestural). Evidence of how the vocal and gestural modality are combined in language also stems from experimental research on adults, where research has shown that vocal and gestural signals are integrated at behavioural, cognitive and neural levels (e.g. [51–57]). Indeed, experimental research conducted in the laboratory has demonstrated that visual behaviour is fundamental to the processing of language. During comprehension and production people combine verbal information with that derived from visual modes of communication, including eye gaze (e.g. [58–60]), manual gestures (e.g. [61,62]), as well as other visible bodily behaviour such as head movements (e.g. [63]). Arguably, the most thoroughly experimentally researched contribution is by manual iconic and other representational co-speech gestures [64–67], as well as visual deixis [68], but at least signals emitted by the face [69,70] and head [63,71], too, appear to fulfil important functions. Recent studies have shown a processing advantage for multimodal signals in humans and other animals (e.g. [72,73]; see also [74]). For example, gestures enhance neural processing of speech comprehension especially in noisy contexts [75], guide visual-spatial attention in healthy and clinical populations [76,77], and help patients with language disorders to express themselves [78,79]. Questions produced with gestures are responded to faster [80], indicating that the multimodal signal may help processing during the interaction.

## 1.3. Emergence and evolution of human language in the wild and in the laboratory

Since traces of early forms of human language cannot be unearthed, researchers interested in understanding its origins and early evolution often rely on indirect sources of evidence. One such source is the study of modern language emergence: pidgins and creoles, for instance, are studied in order to understand how impromptu communication systems can evolve into fully fledged natural

languages (e.g. [81–83]). Similarly, research on homesigns and emerging sign languages provide insightful clues about how communication can be created in the visual modality and about how it can acquire systematic linguistic structure over time (e.g. [9,84]).

Together with naturalistic approaches, experimental methods have also made their way into linguistic evolutionary research. Indeed, studying how artificial languages are created in the laboratory has become a serious method for testing hypotheses about language emergence and evolution [85]. To that end, laboratory experiments have looked at how human participants create new communication systems and at how unstructured semiotic inventories adapt to become more language-like in nature ([86–88]; for similar work conducted with computational agents, see [89,90]).

Existing experimental paradigms make use of communication games which allow experimenters to manipulate not only what participants communicate about, but also how they go about communicating. Previous studies have explored, for instance, the cognitive preconditions necessary for the emergence of language, from how people signal communicative intent (e.g. [91–93]), to how they establish common ground (e.g. [94–97]). Recently, studies have started exploring how different semiotic modalities can be used to communicate in the absence of verbal language, including whether the use of combined modalities might affect improvised communication. It is also possible to study the role of context, the relative difficulty of disambiguating individual meanings or the need to distinguish particular instances from general categories [98–102], though this is beyond the scope of the current study.

Among such experimental studies focused on the role of modality in early language emergence, only a few studies have tapped into the issue of multimodality (e.g. [42,103,104]). In these studies, participants were asked to interact and communicate about (or to just comprehend, based on pre-recorded stimuli) concepts conveyed by gestures alone or by multimodal signals (i.e. gestures and non-linguistic vocalizations), and in some cases by non-linguistic vocalizations alone. These studies found that communicative success or comprehension was better for gestural rather than vocal communication, and that multimodal communication did not provide any advantage compared to gesture alone. It is clear that the behaviour of modern, linguistic, encultured humans cannot provide direct evidence for the behaviour of early humans (modern humans have more developed theory of mind, experience with pragmatic communication and possibly other adaptations for linguistic communication). However, they do provide a useful model and these experimental results have been used to support theories of language evolution. For example, Fay et al. [42,103] made two inferences [105]: first, that gesture has more potential for motivated signs (iconicity, indexicality) than vocalizations; and second, that iconic signals can help bootstrap a communication system. They conclude that gesture would have been an important part of early communication systems, to the extent that gesture helps convey meanings through iconicity. Fay et al. [42] are careful to note that 'rather than commit to a gestural origin, we prefer an origin in which humans initially communicated using motivated signs (icons and indices), whether gestural or vocal (i.e. a multimodal origin)' (p. 1365). That is, the results were intended to suggest that motivated (iconic) signs have an advantage over abstract signs when creating a language (see also [106]), rather than claim that this process did not involve vocalizations or multimodality. However, subsequent studies have interpreted these results in a variety of ways, including that they support a 'gesture-first' theory of language evolution (see electronic supplementary material, S1). Part of the reason might be that these experimental studies compared the communicative effectiveness of modalities, rather than directly measuring the potential for each modality to create motivated signals, i.e. signals which are grounded perceptually or contextually as opposed to being abstract or arbitrary [107,108]. In any case, a frequent claim is that experimental evidence supports the view that multimodality does not provide any advantage during the process of language emergence above and beyond the gestural modality. Our study seeks to provide further experimental evidence that tests this interpretation.

In previous experimental studies, participants were asked to communicate familiar meanings, such as the act of sleeping, the emotion of pain or objects such as 'rock' and 'fruit'. Many of these concepts are already associated with conventionalized gestures. For example, Ortega et al. [109,110] asked hearing participants to produce gestures for a range of meanings. For many meanings, the gestures produced were highly consistent across participants. For example, 'sleeping' is frequently depicted by pressing the hands together palm-to-palm and laying them under a tilted head with eyes closed (Ortega et al. found that 95% of participants produced this gesture for 'sleeping'; G Ortega 2017, personal communication). Other meanings, such as 'fruit', might not exhibit such consistent production patterns, however, these are usually depicted by hearing gesturers on the basis of easily identifiable prototypes, like a banana (depicted as the action of peeling a banana), or apple (depicted as the action

of biting a round-shaped object held in one's hand). In the experiments above, although participants were instructed that they could not use words (conventionalized sounds) in the vocal-only condition, they were not instructed that they could not use conventionalized gestures. Thus, a critical question that remains unanswered is what people would do when communicating without conventional signals, a situation that would simulate the origins of human communication most closely.

In order to simulate communicative constraints as similar as possible to those found in first encounters between individuals in the absence of any shared symbols, what is needed is a set of stimuli which pre-empt the use of existing conventional signals and have clear availability to each modality. We present an experiment which uses novel stimuli for which creating labels with conventionalized gestures and vocalizations is difficult.

# 2. The present study

In order to investigate how people communicate on the basis of spontaneously created non-linguistic signals, we had pairs of participants describe novel stimuli to one another using only non-linguistic vocalizations and visible gestures. The stimuli that the participants were asked to describe were either sounds or images (see §3.2). Note that our aim was not to test whether participants are better able to communicate about visual as compared to auditory stimuli using the visual and/or the vocal modalities. Rather, by comparing unimodal vocal and gestural signalling to multimodal signalling in an interactive communication game, we explore how the use of combined modalities might affect improvised non-linguistic communication for both sets of stimuli. Thus, we investigate:

(i) how players communicate without conventionalized meanings using vocal and gestural signals in real-time, face-to-face interaction;
(ii) whether multimodal signalling grants players any advantages over unimodal signalling in the context of such improvised communication; and
(iii) whether players are successful in mapping their partners' vocal and/or gestural signals onto stimulus items.

To this end, participants were asked to perform a referential communication task as discussed above. We measured the amount of time it took them to convey a stimulus to their partner (*efficiency*), as well as whether they guessed the target items correctly (*accuracy*). In order to address (i) above, we also explore the data to look at interactional aspects such as turn taking.

These considerations lead to several hypotheses. For example, The *direct linkage hypothesis* predicts that participants will perform better when describing stimuli which match the modality in which they are communicating. There is a direct mapping when using vocal signals to describe sounds, and an indirect mapping when using vocal signals to describe images (table 1). Cross-modal mappings are possible, of course [111–114], but on average visual stimuli would be guessed more accurately and more efficiently when participants can use gesture-only than when they can use vocalization only, and vice versa for auditory stimuli. This predicts that there will be an interaction between modality condition (vocal-only, gestural-only and multimodal) and the modality of the target stimuli (auditory or visual).

The *gestural advantage hypothesis* states that gesture trumps vocalization. As such, despite the motivated link between the stimuli and the semiotic means by which these stimuli are represented, participants relying on visible gesture may do better overall. This predicts that there should be a main effect of signal modality, with the vocal modality condition being the least accurate and efficient and no difference between the gesture modality condition and the multimodal condition. The *multimodal advantage hypothesis* predicts that participants who are able to use both vocal and gestural signals would be more accurate and efficient than participants restricted to using only one type of signal. This predicts a main effect of modality condition, over and above the interaction expected by the direct linkage hypothesis. Given that participants in the multimodal condition are free to use either unimodal or multimodal signals, there may also be an additional effect of whether multimodal signals are actually used in a given trial.

The gestural advantage prediction and the multimodal advantage predictions are not mutually compatible, and the statistical analysis of the results should be able to distinguish them, either in terms of average accuracy or efficiency, or in the rate of improvement of either over the four games. The direct linkage prediction may be confirmed independently of whether the other two predictions are confirmed.

**Table 1.** Summary of the motivated potential of each signalling modality (auditory versus visual) in relation to each type of stimulus (vocal versus gestural).

| | | signalling modality | |
|---|---|---|---|
| | | vocal | gestural |
| stimuli | auditory | direct mapping | no direct mapping |
| | visual | no direct mapping | direct mapping |

Participants in the multimodal condition might gain an advantage for two reasons. Firstly, multimodal signals might be more effective than unimodal signals. A second kind of advantage might be the flexibility to draw on these two modalities to produce unimodal visual or unimodal vocal signals as required. We find that participants do use multimodal signals, but we do not find a multimodal advantage in the first sense that using a multimodal signal predicts greater efficiency or accuracy for a given trial. We do find an advantage of the multimodal condition as a whole in bootstrapping a communication system, which we attribute to an advantage in flexibly deploying modalities

There are also some minor predictions about other factors that could affect accuracy or efficiency. Participants should improve over the course of an experiment (perhaps in a nonlinear way as they approach ceiling performance), so that the number of trials passed should predict performance. There should be no effect of the order of stimulus set, and the model should also control for random differences between participant's abilities (e.g. some might put more effort into their initial descriptions while others rely more on interaction to converge on the correct meaning) or the differences between individual stimuli (some stimuli might be easier to communicate than others). Accuracy and efficiency may be related in one of two ways. Participants that put more time into making sure they understand each other (less efficient, more interaction) might have a better chance of guessing correctly (more accurate), so that the two are negatively related. On the other hand, it is possible that a longer trial (more interactions, longer initial description) is an indication that there is a problem with communication, and therefore predict lower accuracy, so that the two are positively correlated. However, if the latter is the case, then spending time resolving communication problems in the current trial might improve efficiency in later trials. The main predictions above should hold even when considering these additional potential differences between players.

In addition to these predictions, we also have more exploratory questions. For example, does interaction improve communication? Many models of the emergence of linguistic conventions provide very minimal possibilities for interaction (e.g. [89,90]; see [115,116]). In an experiment with free face-to-face interaction, participants have the option to take time to seek further clarification. Are there differences between modalities in the extent to which participants choose to do this? Choosing to seek clarification might be characterized as a trade-off: partners reduce their efficiency in the hope of greater accuracy. Are trials with more interaction more likely to be correct? Does paying a cost in terms of efficiency now lead to benefits in the future? Other questions relate to how people use multimodal signals. Within a multimodal signal, how do the individual components relate to each other? For example, how do they overlap temporally? If gesture is a dominant modality, then perhaps vocal components of a multimodal signal only appear as short additions within a wider gestural component. We will address these additional questions in the final analysis.

# 3. Method

## 3.1. Participants

Thirty participants (10 male, 20 female, all native speakers of Dutch) participated in the experiment. All participants were recruited using the participant database of the Max Planck Institute for Psycholinguistics, and none had any command of any sign language. Participants were paid €10 each.

## 3.2. Stimuli

The stimuli used in the experiment consisted of sounds and images without existing conventional labels. Auditory stimuli consisted of eight sounds resembling both generic natural sounds and human-made/

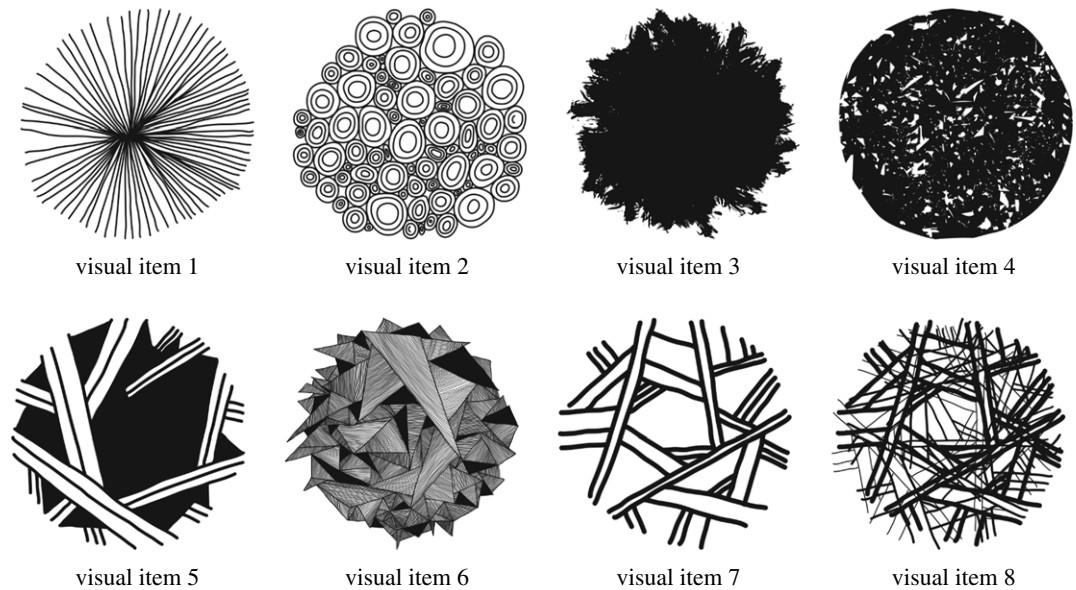

| | | | |
|---|---|---|---|
| visual item 1 | visual item 2 | visual item 3 | visual item 4 |
| visual item 5 | visual item 6 | visual item 7 | visual item 8 |

**Figure 1.** Visual stimuli set.

artificial sounds. The audio files are available in the electronic supplementary material (S6), and may be described as: (1) door creaking; (2) a tree falling; (3) air leaking; (4) a balloon bursting; (5) paper being crumpled; (6) wings flapping; (7) pages of a book being turned; (8) a bouncing 'boing' effect. We stress that these descriptions may make the sounds seem more conventional than they are, and that the participants interpreted the sources of the sounds in a variety of ways. Visual stimuli, presented in figure 1, consisted of eight images of circles filled with different patterns and shapes (e.g. lines, triangles, etc.).

The auditory and visual stimuli cannot be treated as being equally matched in many dimensions. They were designed to capture sounds and visual patterns that can be perceived in the natural environment, but which have no specific conventionalized signals in any modality. A pilot study checked that the stimuli did not elicit any conventionalized non-linguistic signs. The stimuli were pre-tested both with individual participants, in a non-interactive setting, and with pairs of participants using the same procedure as the main experiment.

## 3.3. Procedure

Dyads of participants played a referential communication task. One participant took the role of director and the other took the role of matcher, with participants switching roles after each trial. The director's task was to communicate a target item to the matcher, and the matcher's task was to guess the correct target from an array of three items.

The experiment took place in a facility dedicated to research on gestures and sign language (soundproof, no distractions). Participants were seated at a desk with a laptop (HP Probook 470, screen resolution $1600 \times 900$). They could observe stimuli by clicking buttons on the screen in a custom web program (available at https://github.com/seannyD/multimodalCommunicationGame and https://doi.org/10.5281/zenodo.3333408). Auditory stimuli were presented through headphones and image stimuli were presented on the screen. Clicking a button would play the audio stimuli or reveal the image stimulus for 1 s before hiding the image again. In order to match the availability of both stimuli types, only one audio stimulus could be heard at any one time. The director saw one button for the target stimulus and one button to allow them to move on to the next trial. The matcher saw three stimulus buttons, relating to the correct target and two other stimuli from the set. The selection of alternative stimuli and the order of stimuli within the array were counterbalanced. The matcher also had a button below each stimulus button that allowed them to indicate their guess (and which moved them on to the next trial). As in previous experiments, the director could not observe the matcher's guess and the matcher was not told whether their guess was correct. This meant that no interaction or learning could happen through the experimental program, encouraging the participants to interact in person.

Trials were administered in two blocks. In one of the blocks, all targets and distractors were drawn from the *visual stimuli* set and in the other block they were drawn from the *auditory stimuli* set. The order

(a)    (b)

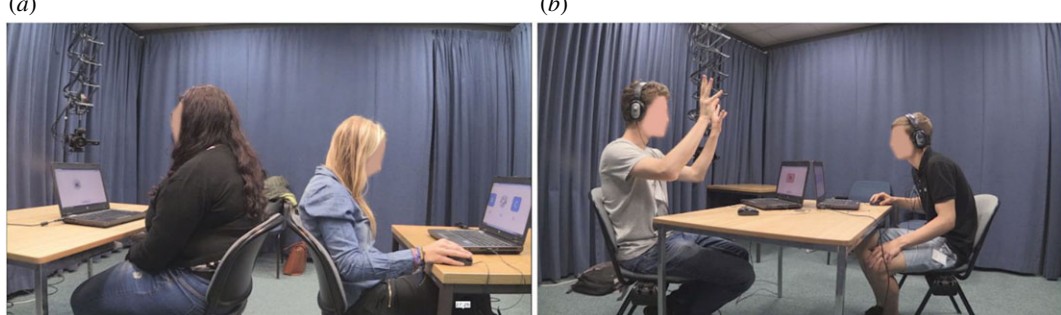

**Figure 2.** Experimental set-up for the vocal modality condition (a) and gestural and multimodal conditions (b).

of blocks was counterbalanced across the experiment. As in other experimental semiotic studies [88], participants were asked to repeatedly describe items. In our study, each block consisted of four games and each game consisted of 16 trials (each participant was given each of the eight stimuli as a target item to direct, in a random order). This produced a total of 128 trials per dyad.

Each dyad was randomly allocated to one of three *modality conditions*: vocal, gestural or multimodal ($n = 5$ per condition). In the vocal modality condition, players were instructed that they could only produce non-linguistic sounds. These sounds could be produced with the mouth or any other bodily articulator (e.g. whistling, clapping hands, hitting the desk, so 'vocal' may not be a completely accurate label, but is convenient since we use 'auditory' to refer to one of the stimuli conditions), and could not consist of spoken words or any other form of verbal language. In this condition, players sat at desks in front of their computers, but back to back and had no visual contact whatsoever (figure 2). In the gestural modality condition, players could only produce visible body movements. These visible movements could be produced with the hands or any other bodily articulator, including the face and the torso (e.g. waving arms in the air, tracing shapes on the table, nodding). In this condition, players faced one another, their individual laptops being placed on a single table located between both players, which allowed a clear view of each other's head, arms and torso (figure 2). In the multimodal condition, players could produce both sounds and visible body movements, and were free to produce either unimodal or multimodal signals. The set-up was identical to that of the gestural modality condition, which meant that players sat facing one another and had full visual contact.

Similarly to previous studies (i.e. [42,103]), players were able to interact freely within a trial, as long as they respected the restrictions of their particular condition. In order to avoid confusion about when a trial had ended (and subsequently which trial they were engaged in), participants were instructed that they could give a specific signal to mean that they understood the referent their partner was trying to convey and were ready to move on to the next trial (a 'thumbs-up' when gesture could be used or saying 'OK' when vocalizations could be used). A trial, therefore, could be as short as the director issuing a description followed by the matcher acknowledging understanding. However, it could also involve players taking various turns in communicating before the matcher finally made a guess.

From the conversation analysis literature, we might expect various kinds of responses from the matcher. The preferred response is to show that they understood. Otherwise, the matcher might initiate a 'repair sequence' by signalling that they did not understand and inviting the director to work with them to resolve the issue [117]. There are broadly three types of repair: 'open', 'restricted' and a 'candidate understanding' [112,113]. The matcher can initiate open repair by simply signalling that they do not understand (e.g. raising pitch to signal a question, raised eyebrows, furrowed brows or a 'freeze look', [118]). Restricted repair involves signalling that there is a problem with a particular part of the original signal. In spoken language, this can be done using questions (e.g. A: 'Sibby's sister had a baby', B: 'Who's sister?', and similar strategies exist in signed languages). Restricted repairs are harder to do without an established communication system, but can be done for example by pointing at the space where (part of) a gesture was produced, or repeating (part of) a vocalization up to the point where there was a problem. Finally, matchers might offer a candidate understanding which is an alternative description of the stimuli they think the director means that the director can confirm or clarify.

For each trial, the experimental program tracked matcher's guesses, the timing of responses and the trial, game and block numbers. The experiment was recorded using three Canon XF 205 HD genlocked cameras (frame rate 29.97 frames s$^{-1}$). One camera captured both players together from the side, while the remaining two cameras gave a frontal view of each player separately.

## 3.4. Coding and measures

For each trial, the experimental program provided the accuracy (whether the matcher chose the correct stimulus), the time that the target was first displayed on the director's screen, the time that the matcher made a choice by clicking a candidate, and which participant was playing which role. These data were linked to an ELAN annotation file (version 4.9.4, [119]) using the *pympi* library for Python [120] in order to align the timecodes with the video of the communicative acts of the participants. Manual coding in ELAN was carried out to define the start time and end time of each signal and each turn. For the purposes of this study, a signal was defined as an intentional, ostensive production in either the auditory or visual modality. Auditory productions were defined as sounds produced by the mouth and vocal tract or airways (e.g. speech sounds and whistles), but could also include any other sounds produced by the participants (e.g. knocking on the table). Since the vast majority of signals were vocal, and to distinguish them from auditory *stimuli*, we label these 'vocalizations'. False starts and hesitations were not included in the production time window. Visual productions or 'gestures' could include movements produced with the body including the hands, head and the face. These included visible movements which were judged to be intentionally communicative (following e.g. [43]) including pointing gestures or freeze looks ostensively directed at one's partner (which may help alignment in interaction, e.g. [58–60,121,122]), but not non-communicative movements such as head-scratching or hair fiddling (both used instrumentally rather than as part of a depictive act), nor signals aimed at getting the initial attention of the participant like waving (our reliability tests in §3.4.1 show high agreement between coders for what constitutes a signal and for turn modality categorization). For manual gestures, following Kita *et al*. [123] and other studies, the annotation window did not include gesture preparation and retraction, nor extended freezes (i.e. 'holds') before an initial stroke or after a final stroke. That is, annotations started at the first identifiable moment of a visual depiction (the 'stroke') and ended at the beginning of a halt or retraction. For instance, a player raises one of their hands and positions it in line of sight of their partner, then reviews the target item before moving the hand around in circles, then lowers their hand to the desk. The signal would be annotated as starting at the moment the hand begins making circles and ends just before it is lowered to the desk. Communicative acts which matchers produced to signal that they understood the referent were not annotated, as this was a routine which players were explicitly told to follow and which itself followed a pre-specified format ('thumbs-up', 'OK'). After the experiment was run, a random 10% of multimodal director first turns were coded for what part of the body was used for the visual component of the signal; 96% involved hands or arms, 21% involved the body and 37% involved eyes, eyebrows or gaze (63% had multiple visual components, none involved only eyes/eyebrows/gaze).

Signals were grouped together under turns, which were defined, as in studies of conversation analysis (e.g. [124]), as the time in which a person holds the floor while speaking [124]. For the purposes of this experiment, a turn consisted of any sequence of communicative signals produced by a player within a trial without interruption or intrusion from the side of their partner (from the start of the first signal to the end of the last signal). For instance, a director might describe an item by producing a vocalization, then pause, and upon receiving no response from their partner vocalize once again. For the purposes of coding the present data, these would be regarded as a single turn. Therefore, each turn included at least one signal, but could include several signals from both modalities. Each turn was categorized as 'unimodally vocal' (a turn with only vocal signals), 'unimodally gestural' (a turn with only gestural signals) or 'multimodal' (a turn with both vocal and gestural signals that at least partially overlapped in time). It was possible to produce a 'mixed' turn which contained non-overlapping signals from two modalities, but in practice only one turn of this type was observed.

The coding above allowed the measurement of various variables for each trial such as the number of turns by each participant, the modality of each turn and the length of the director's first turn. The trial length (the efficiency of the trial) was defined as the amount of time between the start of the first turn by the director to the time at which the matcher made a choice using the experimental program. In total, 1882 trials were coded (32 trials; 2%, were excluded due to experimenter error or equipment malfunction).

### 3.4.1. Reliability of coding

Two hundred and thirty-nine trials were coded by a naive second coder according to the specification above (12.7% of all trials, from nine dyads, about 30 min in randomly chosen sections, covering all combinations of stimulus type and modality condition). For 95% of trials, the two coders agreed on the number of turns in the trial (Cohen's weighted kappa = 0.84, 'almost perfect' agreement according

to [125]). The trial length (efficiency) values for the two coders were correlated with $r = 0.80$ (a permutation test showed that the true differences between judgements were significantly smaller than permuted data, $z = -22.1$, $p < 0.001$; $r = 0.98$ for the 95% of trials where coders agreed on the number of turns), and this was not biased by modality condition or stimulus type (see the electronic supplementary material). The mean difference between the onset of signal annotations between coders was 211 ms (interquartile range = [80 ms, 309 ms], a permutation test showed that the true differences were significantly smaller than permuted distances $z = -21.2$, $p < 0.001$). This differed slightly by modality of the signal (mean for acoustic = 141 ms, mean for visual = 256 ms), perhaps because of the different time resolution of the different modalities (29.97 frames per second for video, 44 000 kHz for audio), but differences were still over 10 times smaller than the standard deviation of turn lengths. There were no significant differences by condition. Comparison for the offset of signals was similar (mean difference = 214 ms [73 ms, 276 ms], $z = -26.0$, $p < 0.001$). The modality of a turn is only possibly contentious for the multimodal signal condition. Ninety-eight multimodal condition trials were coded by both coders and showed 'almost perfect' agreement in assignment of turn modality categories (Cohen's kappa = 0.81, 85% of turns coded identically).

## 3.5. Statistical procedures

### 3.5.1. Confirmatory analyses

Mixed effects modelling was used to analyse the data (*lme4* package in R, [126,127]), with estimations of significance coming from model comparison tests. The two dependent variables in this study were the *accuracy* of each trial (correct or incorrect guess by the matcher) and the *efficiency* of each trial (the length of time from the start of a director's first turn up until the moment their partner made a guess). Significance was assessed by model comparison. Starting with a null model with only random effects, *a priori* selected control variables were added to produce a baseline model, then each main fixed effect was added. A likelihood ratio test was used to assess the additional improvement that each fixed effect provided in model fit (taking into account the additional number of model parameters). Comparisons between different modality conditions were obtained by fitting equivalent models with different intercept conditions (relevelling). Variance explained by each fixed effect was estimated by pseudo-$R^2$ [128]. See the electronic supplementary material, S4 and S5 for more details.

#### 3.5.1.1. Accuracy

Accuracy for each trial was coded as a binary variable (correct or incorrect) and used as the dependent variable in a binomial mixed-effects model. The fixed factors included: modality (vocal versus gestural versus multimodal), stimulus type (auditory versus visual), the interaction between modality and stimulus type, number of trials passed, trial length (to control for participants being more accurate just because they took more time to communicate), director's first turn length, whether the director's first turn was multimodal, whether the matcher produced a turn, the cumulative number of trials for which matchers had produced a turn in the stimulus type block, block order, the interaction between modality and trial length, and the interaction between modality and the director's first turn length. The random factors included a random intercept for each player nested within dyad, with random slopes for stimulus type and accuracy, and a random intercept by item with a random slope by modality. The final model reported below only included fixed effects that significantly improved the fit of the model. Given the hypotheses listed in §1, we expect the following main effects and interactions:

*Direct linkage hypothesis*: interaction between stimulus type and modality (for auditory stimuli, vocal > gestural; for visual stimuli, gestural > vocal);
*Gestural advantage hypothesis*: main effect of modality (gestural and multimodal > vocal);
*Multimodal advantage hypothesis*: main effect of modality (multimodal > gestural and vocal).

#### 3.5.1.2. Efficiency

Trial length was used as the dependent variable in a Gaussian linear mixed effects model (log transformed to fit the Gaussian assumptions). The fixed factors included: the modality condition (vocal-only, gesture-only, multimodal), stimulus type (auditory, visual), the interaction between modality and stimulus type, number of trials passed (linear and quadratic), accuracy, whether the matcher produced a turn, the cumulative number of trials for which matchers had produced a turn in

the stimulus type block and various interactions that significantly improved the fit of the model (see electronic supplementary material, S5). Note that accuracy was included in the efficiency model and that efficiency was included in the accuracy model. We expect the effects to agree.

Since participants in the multimodal condition were allowed to produce unimodal signals as well as multimodal signals, the 'modality condition' variable may not necessarily reflect the actual use of multimodality. Therefore, we also included an additional factor in the model indicating whether the director's first turn was multimodal. This is always false for participants in the vocal and gestural conditions, and might be true in the multimodal condition. In this way, we could test whether multimodal descriptions actually lead directly to shorter trials compared with a general advantage for having the option to produce multimodal signals (we only considered initial descriptions because the efficiency and accuracy measures are at the level of the trial, so any predictors also have to be at the level of the trial rather than individual turns, because not all trials had multiple turns and because directors and matchers produced different distributions of signal modalities). The random factors included a random intercept for each player nested within dyad, with random slopes for signal type and accuracy, and a random intercept by item with a random slope by modality. Electronic supplementary material, S5 shows that the results are robust to various alternative approaches, such as using a Poisson model or a more minimal fixed effects structure.

As for accuracy, given the hypotheses listed in §1, we expect the following main effects and interactions:

*Direct linkage hypothesis*: interaction between stimulus type and modality (for auditory stimuli, vocal > gestural; for visual stimuli, gestural > vocal);
*Gestural advantage hypothesis*: main effect of modality (gestural and multimodal > vocal);
*Multimodal advantage hypothesis*: main effect of modality (multimodal > gestural and vocal).

### 3.5.2. Exploratory analyses

In addition to the main statistical analyses, some exploratory analyses of the participants' behaviour were conducted. In order to gain more insights about communication in the multimodal condition, players' communicative behaviour in that condition was further analysed in terms of the modalities in which it was produced (i.e. vocal-only, gestural-only, or both vocal and gestural). As such, we assessed whether signalling in the multimodal condition differed from signalling in the unimodal vocal and gestural conditions.

## 4. Results

The coded data, experimental software, results and analysis are available at GitHub: https://github.com/seannyD/multimodalCommunicationGame and are archived within the Zenodo repository: https://doi.org/10.5281/zenodo.3333408. Data are also available within the supplementary material. Before reporting the quantitative results, we briefly discuss various communication strategies. As in previous experiments, participants used motivated signals. Even for cross-modal signals, participants generally found successful ways of distinguishing stimuli and all participants achieved rates of success much better than chance in all conditions and for each set of target stimuli, even when looking at just the first game (§4.1). Electronic supplementary material, S2 shows examples of signals for various stimuli.

Directors using vocalizations to describe auditory stimuli almost universally attempted to mimic the target stimuli. It is reasonably complex to produce sounds with human vocal chords to match other sounds, but a simple task for the listener to identify the meaning. Directors using gestures to describe visual stimuli used a range of strategies, some of which relate to Müller's [129] categories of gestural representation. For example:

— Using repeated movements to convey repeated patterns or complex stimuli.
— Pointing to objects (e.g. a black t-shirt for an image that was mostly black)
— 'Moulding' [129]: Using hand shapes to convey the outline of the shape (e.g. a smooth outline for image 3)
— 'Tracing' [129] outlines or parts of shapes (e.g. individual lines in image 1 or the circles inside image 2.
— Using two or three fingers like a claw scratching to convey parallel lines which distinguish some stimuli.

**Table 2.** Percentage of trials where the matcher produces at least one turn.

| | vocal-only (%) | multimodal (%) | gesture-only (%) |
|---|---|---|---|
| auditory stimuli | 0.3 | 5.5 | 8.3 |
| visual stimuli | 0 | 10.3 | 10.1 |

Directors using gestures to describe auditory stimuli used similar strategies. In general, cross-modal mappings between domains were used, such as using big gestures to convey loud sound, short movements to convey short sounds or movement to convey spectral elements (e.g. small finger movements to convey 'tinkling' in sound 7). Participants also used several strategies covered by Müller's [129] categories of gestural representation such as acting (gesturing the action associated with the perceived source, e.g. miming opening a door to convey sound 1), instrumental gestures (using gestures to convey the form of the object, e.g. a fist to represent the 'bouncing ball' of sound 8) or tracing.

While the gestures involved primarily the hands and arms, we note that facial gestures were also produced. For example, wide eyes to convey the surprise that a loud sound might make, or puffing cheeks to convey the movement of air (a kind of 'acting' gesture). Participants also used pointing, for example, pointing at one's heart to convey the rhythmic, 'beating' quality of sound 4 (we note that pointing is a conventionalized gesture, but not a conventionalized label for the specific stimuli). While the participants may have linked the sounds to gestures through a multimodal source, we note that they imagined a range of sources for the same sound. For example, sound 6 was conveyed by gesturing wings flapping or a heart beating, and the sound derived from crumpling paper was conveyed by one pair by gesturing the firing of a gun.

Directors using vocalizations devised cross-modal mappings between sounds and aspects of the images. For example, pure tones could represent lines and so several tones represent several lines (image 1) or alternating notes could represent lines at different angles. They also produced iconic sounds, such as a splashing sound to convey the apparent ink blots in the 'filled in' shapes, or vocal aspects that conveyed the imagined movements the shapes might produce (e.g. vocalizations seem to depict circular motion for image 2).

Some pairs also created signals for categories of stimuli. For instance, a pair in the multimodal condition used a fist inside a hand representing the outline of a circle to mean a 'mostly filled shape' (images 2, 3 and 4), which they then combined with another signal to identify the particular stimuli. This is a kind of systematicity which also emerges in other experiments [130–132]. This feature has been explained as a response to pressures for a communication system to be expressive and learnable by new participants (e.g. [133]), though our results have no direct implications for these theories (the innovation of systematic representations is compatible with a subsequent selection of those patterns by a pressure for learnability).

Participants interacted more in the multimodal and gestural conditions than in the vocal-only condition. Table 2 shows the proportion of trials where the matcher produced at least one turn. This was between 5% and 10% for the multimodal and gesture-only conditions, but there was only one trial in the vocal-only condition where matchers produced a turn. We discuss possible reasons for this below. Table 3 shows what kinds of repair matchers produced (data from a meta-study by Micklos *et al.* [116]). There are examples of each type of repair, though open and restricted are the most frequent types. Since the number of repairs is low, we do not claim that there are significant patterns. Electronic supplementary material, S3 includes a detailed analysis of multimodal repair sequences. Participants produced vocalizations to comment on the primary gestural aspects of the signals. An interesting possibility is that different processes of creating a symbol system could be distributed over different modalities. For example, signals could be produced vocally, but commentary on those signals (evaluation of their effectiveness, attitude to the meaning, emphasizing new information, modifying their scope, comparing them with spatial relations, etc.) might be gestural, just as happens in modern multimodal communication (e.g. [134,135]).

## 4.1. Accuracy

The final model included fixed effects for modality condition, stimulus type, their interaction, number of trials, trial length and the cumulative number of matcher responses. It correctly predicted the accuracy of 87% of trials (AIC = 1281.3, BIC = 1397.7, total variance explained = 48%, fixed effects explained 18%

**Table 3.** Distribution of repair types for 132 cases of repair initiation across conditions and stimuli types. Percentages add up to 100% in each column. The single case of a matcher responding in the vocal-only condition is not shown.

| | multimodal | | gesture-only | |
|---|---|---|---|---|
| repair type | auditory stimuli (%) | visual stimuli (%) | auditory stimuli (%) | visual stimuli (%) |
| open | 68 | 34 | 45 | 49 |
| restricted | 32 | 51 | 52 | 40 |
| candidate understanding | 0 | 14 | 3 | 11 |

**Table 4.** Fixed effects for the model of accuracy, including $\beta$ (on a logit scale), the $z$-value for the $\beta$ considering the standard error, the $\chi^2$ value of the model comparison test comparing a model with and without the given variable and the $p$-value associated with that test. Note that $\beta$ and $z$ statistics are provided for each level of each variable, but the model comparison tests whether the variable significantly improves the fit of the model, so is only included once for each variable. $^*p < 0.05$.

| variable | $\beta$ | $z$ | $\chi^2$ | $p$ |
|---|---|---|---|---|
| (intercept) | 3.074 | 4.693 | | |
| visual modality condition | −0.602 | −0.895 | 1.228 | 0.541 |
| vocal modality condition | 0.091 | 0.137 | 1.228 | " |
| visual stimuli | −0.940 | −1.161 | 2.452 | 0.117 |
| game | 0.214 | 2.736 | 53.868 | <0.001* |
| trial length | −1.020 | −6.427 | 38.979 | <0.001* |
| cumulative matcher responses | 0.115 | 2.391 | 7.117 | 0.008* |
| visual condition : visual stimuli | 0.907 | 1.189 | 4.060 | 0.131 |
| vocal condition : visual stimuli | −0.970 | −1.328 | 4.060 | " |

of the variance). Table 4 shows the model statistics. Participants in all three conditions performed above chance (figure 3, average accuracy per dyad = 86%, expected accuracy by random guessing = 33%) and improved significantly over the course of the experiment (accuracy increased 1.5 percentage points per game, $p < 0.001$, pseudo-$R^2 = 0.05$). There was no significant main effect of stimulus type, which means no single stimulus set was easier to communicate, overall, than the other ($p = 0.12$). Accuracy did not significantly differ by modality condition ($p = 0.54$). In the raw data, there appears to be an interaction between stimulus type and modality condition such that visual stimuli were guessed less accurately in the vocal-only condition (about 11 percentage points lower than visual stimuli in the multimodal condition). However, according to the model, this is not significant ($p = 0.13$). The interaction is significant if random slopes are removed from the model, suggesting that the difference in accuracy is not greater than would be expected by random differences between dyads or items. Therefore, our conclusion is that visual stimuli may be harder to guess accurately in the vocal condition, but that ceiling effects and limited power mean that we cannot reliably detect this difference.

Longer trials were guessed less accurately ($p < 0.001$, pseudo-$R^2 = 0.05$). However, for every trial where a matcher responded, subsequent guesses were more likely to be correct (10 trials where the matcher responded raised the probability of a correct guess in subsequent trials by about 4 percentage points, fixed effect of cumulative number of matcher responses, $p < 0.01$, pseudo-$R^2 = 0.003$). There were no other significant effects (see electronic supplementary material, S4). All stimuli were guessed better than chance (lowest = visual stimulus 5, 71%; highest = visual stimulus 1, 100%).

## 4.2. Efficiency

The mean trial time across the experiment was 8.7 s, ranging from 2 to 102 s. The predicted values of the final model correlated with the actual values with $r = 0.83$ (AIC = 1714.1, BIC = 1996.6, total variance explained = 69%, fixed effects explained 35% of the variance). Table 5 shows the model statistics. There was no main effect of modality condition (multimodal/gesture-only/vocal-only, $p = 0.73$) nor stimulus

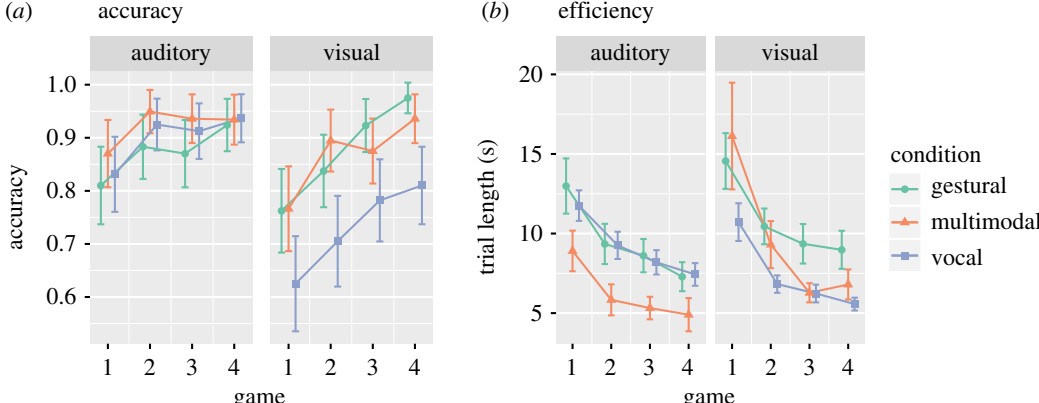

**Figure 3.** Main results, including accuracy (*a*) and efficiency (*b*). The accuracy graph shows the probability of guessing items correctly with 95% confidence intervals by trial, according to stimulus type (auditory versus visual), game (1–4) and experimental condition (gestural versus multimodal versus vocal). The efficiency graph shows mean trial length in seconds with 95% confidence intervals by trial, according to stimulus type (auditory versus visual), game (1–4) and experimental condition (gestural versus multimodal versus vocal).

type ($p = 0.46$), but there was a significant interaction between the two ($p < 0.001$, pseudo-$R^2 = 0.040$), which is explained by two patterns (figure 3). Firstly, for auditory stimuli, participants in the multimodal condition communicated more efficiently than participants in both the vocal ($\beta = 0.69$, $t = 4.4$, Satterthwaite $p < 0.001$) and gestural conditions ($\beta = 0.25$, $t = 1.7$, Satterthwaite $p = 0.1$). Secondly, for visual stimuli, participants in the vocal condition communicated more efficiently than participants in the gestural ($\beta = 0.44$, $t = 3.0$, Satterthwaite $p = 0.009$) and multimodal conditions ($\beta = 0.69$, $t = 4.4$, Satterthwaite $p < 0.001$), though they were also less accurate on average (see §4.1).

There was a significant main effect of number of trials played (trial times dropped by about 1.1 s per game, $p < 0.001$, pseudo-$R^2 = 0.11$), and a quadratic (nonlinear) effect of number of trials played ($p < 0.001$, pseudo-$R^2 = 0.020$), indicating that trial length decreased rapidly with each trial at first, then more slowly towards a steady minimum. There was a significant interaction between modality condition and number of trials seen ($p < 0.01$, pseudo-$R^2 = 0.004$) and between modality condition and the quadratic effect of number of trials seen ($p < 0.05$, pseudo-$R^2 = 0.002$) due to participants in the multimodal condition improving their efficiency at a greater rate than those in the unimodal conditions. By the final game, participants in the multimodal condition were completing trials 2.4 s faster than the gesture-only condition for auditory stimuli and 2.2 s faster than the gesture-only condition for visual stimuli.

Trials were longer if the matcher responded (by about 10 s, $p < 0.001$, pseudo-$R^2 = 0.112$) or the guess was incorrect (by about 2.5 s, $p < 0.001$, pseudo-$R^2 = 0.039$). The latter results agree with the effect of trial length in the model of accuracy. The cumulative number of matcher responses did not predict efficiency above and beyond the number of trials ($p = 0.79$).

In the multimodal condition, participants had the option of producing multimodal signals, but were not required to. There was no main effect of whether the director's first turn was multimodal ($p = 0.22$). That is, individual trials starting with a multimodal signal were no shorter than trials starting with a unimodal signal. However, the main effects for condition reported above still hold: participants in the multimodal condition (where signals may or may not have been multimodal, but crucially, participants had the option of choosing to signal multimodally) end up having shorter trial times.

### 4.2.1. Efficiency: exploratory analyses

After seeing these results, the data were further explored to try to discover the source of this difference between the conditions. At first, we suspected that multimodality might help ground the signals, and that descriptions in later games would drop one modality to make the signals shorter. However, we found that directors in the multimodal condition almost always used the same modality for a given stimulus as the last time they described it (only changing in about 10% of cases), and were equally likely to change from multimodal to unimodal or vice versa.

Next, we investigated whether multimodal descriptions were more efficient. Figure 4 shows the mean total trial times and the amount of time that the director spent producing the first turn in the trial. The

**Table 5.** Fixed effects for the model of efficiency. The variables in the model are scaled and centred, making the $\beta$-value not very transparent. Therefore, the 'estimated change' column converts the $\beta$-value back into milliseconds, to show the estimated difference that each variable makes from the intercept. For example, incorrect trials are about 2.2 s longer than correct trials on average. Other columns show the Wald $t$-value for the $\beta$ considering the standard error, the $\chi^2$-value of the model comparison test comparing a model with and without the given variable and the $p$-value associated with that test. Note that $\beta$ and $z$ statistics are provided for each level of each variable, but the $\chi^2$ statistic tests whether the variable significantly improves the fit of the model, so is only included once for each variable. $^*p < 0.05$.

| variable | $\beta$ | estimated change (ms) | Wald $t$ | $\chi^2$ | $p$ |
|---|---|---|---|---|---|
| visual modality condition | 0.499 | +4605 [847,10136] | 2.53 | 0.63 | 0.729 |
| acoustic modality condition | 0.379 | +3279 [−3204,20495] | 0.76 | " | " |
| visual stimuli | 0.410 | +3607 [1157,6784] | 3.10 | 0.54 | 0.463 |
| game | −0.158 | −1040 [−1268,−803] | −8.11 | 400.30 | <0.001* |
| quadratic effect of game | 0.062 | +452 [266,642] | 4.86 | 78.70 | <0.001* |
| matcher responds | 0.908 | +10530 [7610,14027] | 9.84 | 491.37 | <0.001* |
| cumulative matcher responses | −0.020 | −138 [−309,36] | −1.56 | 0.07 | 0.795 |
| incorrect | 0.268 | +2188 [659,4019] | 2.93 | 13.25 | <0.001* |
| multimodal T1 | 0.115 | +869 [7,1835] | 1.98 | 1.50 | 0.220 |
| visual stims first | −0.080 | −545 [−2113,1515] | −0.57 | 0.42 | 0.519 |
| visual modality : visual stimuli | −0.248 | −1561 [−2921,239] | −1.73 | 15.46 | <0.001* |
| acoustic modality : visual stimuli | −0.691 | −3551 [−4498,−2263] | −4.39 | " | " |
| visual modality : game | 0.019 | +134 [−255,546] | 0.66 | 10.51 | 0.005* |
| acoustic modality : game | 0.008 | +60 [−300,439] | 0.32 | " | " |
| visual stimuli : game | −0.002 | −11 [−364,360] | −0.06 | 0.19 | 0.664 |
| visual modality : quad. game | −0.036 | −252 [−483,−13] | −2.06 | 6.81 | 0.033* |
| acoustic modality : quad. game | −0.003 | −21 [−259,225] | −0.17 | " | " |
| visual modality : matcher responds | −0.008 | −58 [−1545,1825] | −0.07 | 2.92 | 0.403 |
| vocal modality : matcher responds | −0.106 | −715 [−4047,6228] | −0.28 | " | " |
| visual stimuli : matcher responds | 0.089 | +660 [−928,2655] | 0.76 | 0.14 | 0.706 |
| cumulative matcher responses : visual modality | 0.021 | +153 [−73,385] | 1.32 | 1.85 | 0.396 |
| cumulative matcher responses : vocal modality | −0.090 | −615 [−2835,2753] | −0.42 | " | " |
| visual modality : incorrect | −0.077 | −530 [−1924,1238] | −0.64 | 5.17 | 0.075 |
| acoustic modality : incorrect | −0.228 | −1452 [−2684,122] | −1.82 | | |
| visual stimuli : incorrect | 0.027 | +195 [−1119,1797] | 0.27 | 0.06 | 0.802 |
| visual stimuli : multimodal T1 | −0.065 | −445 [−1712,1118] | −0.60 | 0.40 | 0.529 |
| visual modality : visual stimuli : game | 0.014 | +101 [−391,630] | 0.39 | 0.62 | 0.430 |
| acoustic modality : visual stimuli : game | −0.017 | −118 [−594,393] | −0.47 | " | " |
| visual modality : visual stim : matcher responds | −0.104 | −705 [−2401,1600] | −0.67 | 0.86 | 0.355 |
| visual modality : visual stimuli : incorrect | −0.130 | −867 [−2347,1074] | −0.94 | 2.17 | 0.338 |
| acoustic modality : visual stimuli : incorrect | 0.065 | +477 [−1276,2755] | 0.48 | " | " |

difference between the total trial time and the length of the director's turn include the time spent by the matcher making their choice and any negotiation, so we will call this the *comprehension time* (lighter parts of the bars). The data are split by the modality of the director's first turn and by stimulus type, and show that there are different advantages for different combinations of stimuli and modality: if we compare trials where the director used unimodal visual turns to describe visual stimuli (bottom left panel), we see that the gestural and multimodal conditions look similar in game 1 (mean comprehension time in multimodal condition = 7616 ms, gestural condition = 6709 ms, $t = 0.38$, Bonferroni adjusted $p = 1.0$), but by game 4 the participants in the multimodal condition have shorter comprehension times (mean

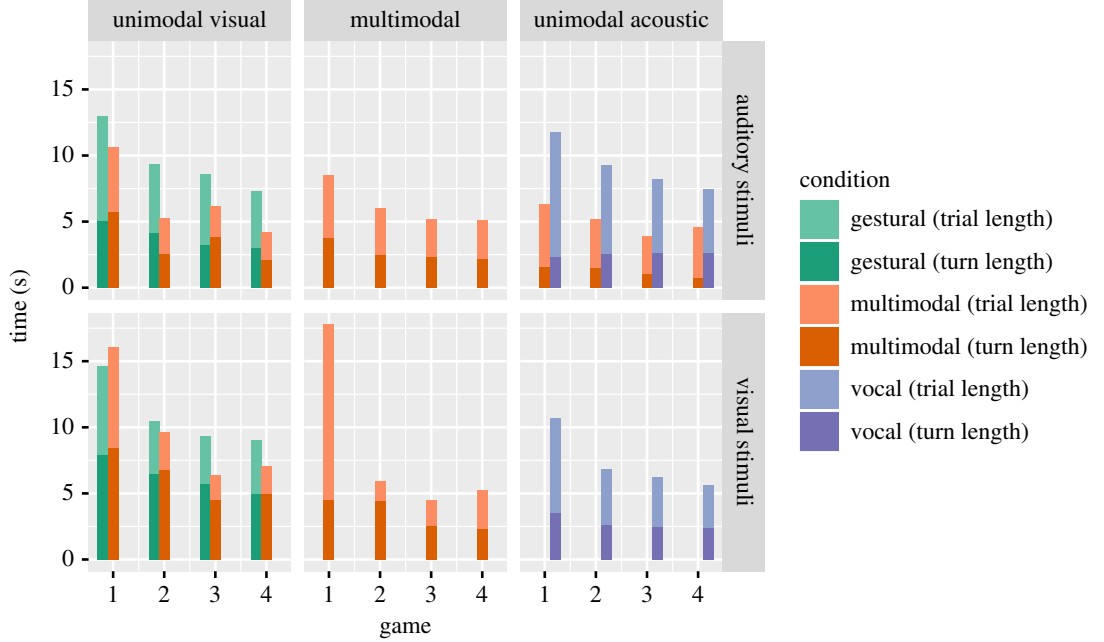

**Figure 4.** Mean efficiency in seconds split by the length of the director's first turn (darker bars), and total trial length (lighter bars). Bars are coloured by experimental condition (gestural versus multimodal versus vocal). The panels divide the data by stimulus type (rows) and the modality of the director's first turn (columns). Directors in the gestural and vocal conditions only produced unimodal turns, while directors in the multimodal condition were free to produce unimodal visual, unimodal acoustic or multimodal turns. There were too few cases of unimodal acoustic turns in the multimodal condition to plot.

comprehension time in multimodal condition = 2080 ms, gestural condition = 4010 ms, $t = 2.67$, Bonferroni adjusted $p = 0.04$). By contrast, if we compare trials where the director used unimodal acoustic turns to describe auditory stimuli (top right panel), by game 4 the advantage in the multimodal condition is mainly in the length of the director's turn (mean turn length in multimodal condition = 746 ms, vocal condition = 2667 ms, $t = 8.4$, Bonferroni adjusted $p < 0.001$). Further, comparing trials where the director used unimodal visual turns to describe auditory stimuli (top left panel), by game 4 we see an advantage for participants in the multimodal condition for both the director's turn length (mean in multimodal condition = 2154 ms, gestural condition = 3052 ms, $t = 2.8$, Bonferroni adjusted $p = 0.02$) and comprehension time (mean in multimodal condition = 2055 ms, gestural condition = 4236 ms, $t = 3.6$, Bonferroni adjusted $p = 0.003$). There were not enough observations to compare directors using unimodal acoustic turns between conditions. Finally, comparing game 4 for the multimodal turns in the multimodal condition (central column) to the unimodal turns in the gesture-only and vocal-only conditions, we see that participants in the multimodal condition have marginally shorter comprehension times for auditory stimuli (mean in multimodal condition = 2762 ms, other conditions = 4500 ms, $t = 2.46$, Bonferroni adjusted $p = 0.08$) and marginally shorter turn lengths for visual stimuli (mean in multimodal condition = 3682 ms, other conditions = 4601 ms, $t = 2.32$, Bonferroni adjusted $p = 0.09$). In summary, the advantage of being in the multimodal condition was not as simple as descriptions being shorter or comprehension being faster overall. Instead, there were different advantages for different types of stimuli. We suggest that participants in the multimodal condition could flexibly and strategically deploy their modalities in order to make the best use of these advantages and in relation to the different modalities' and stimuli's affordances, leading to overall lower trial times in the multimodal condition.

## 4.3. Multimodal signals

Participants in the multimodal condition were not required to produce multimodal signals: they could choose to produce unimodal gestures or vocalizations. Figure 5 summarizes the first descriptions produced by directors in the multimodal condition in relation to the modality in which they were produced and to the type of stimulus they were describing. When describing auditory stimuli, 72% of

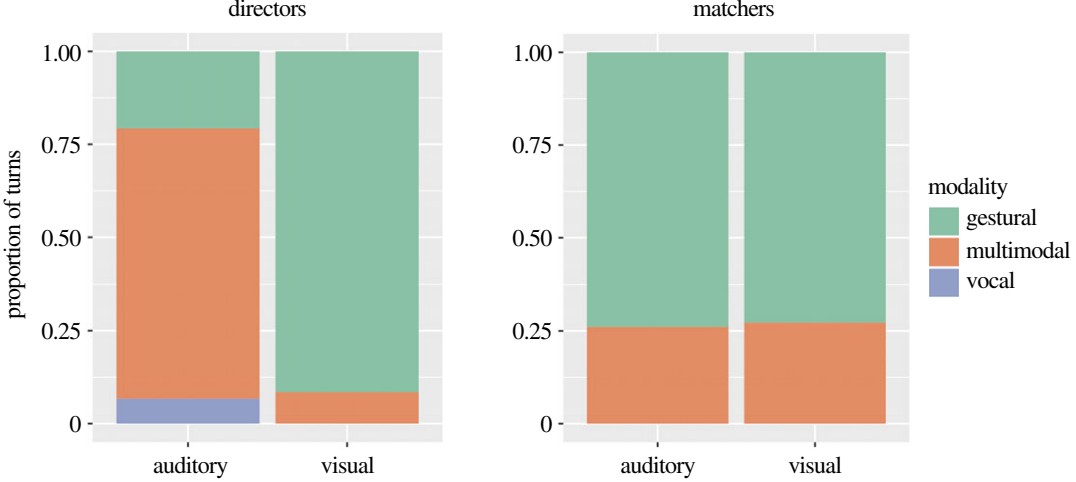

**Figure 5.** The distribution of turns produced in the multimodal signalling condition by stimulus type and signal type. On the left, turns produced by directors ($n = 671$); on the right, turns produced by matchers ($n = 67$). Participants in the multimodal condition had the option to produce multimodal turns, but did not always do so. Unimodal gestural turns are shown in green, unimodal vocal turns in purple, and multimodal turns in orange. Note: there was one turn produced by a director in the visual stimuli condition which consisted of a unimodal vocal signal and a gestural signal. This turn is not shown here.

directors' first turns were multimodal, compared to only 8% when describing visual stimuli. Interestingly, this pattern is not found for matchers: they mainly produce gestures. Recall that the participants switch roles after every trial, so this is not driven by particular participants, but by the roles they are playing.

In order to look more closely at descriptions of auditory stimuli by directors in the multimodal condition, figure 6 shows the ratio between the length of the visual component and the length of the auditory component. The visual component of the multimodal signal is often longer than the auditory component. This shows that, even when communicating about sounds, participants who are able to use both gestures and vocalizations choose to employ gestural signalling extensively in their communication. In fact, gestural signalling is more pronounced not only in terms of its relative frequency to the vocal components of multimodal signals but also temporally. For example, figure 7 shows the relative temporal relation of the production of the auditory and visual components (for multimodal signals produced by directors communicating about auditory signals). There are few vocal signals that appear outside the gestural signal window. Furthermore, there is a bias for the vocal signals to appear later in the turn, with around a third of vocal signals starting at the same time as the gestural signal, but around two-thirds ending at the same time.

In summary, we found that:

— There were no large differences between conditions in terms of accuracy.
— For auditory stimuli, participants in the multimodal condition were more efficient than in either of the unimodal conditions.
— For both visual and auditory stimuli, participants in the multimodal condition improved their efficiency faster than those in the gesture-only condition.
— The result is that, by the end of the experiment, participants in the multimodal condition were more efficient than in the gestural condition for either auditory or visual stimuli.
— Participants in the vocal-only condition describing visual stimuli had shorter trial times, but were also slightly less accurate than the gesture-only condition and interacted less than in the other conditions.
— Participants in the multimodal condition did not always produce multimodal signals. In fact, trials in the multimodal condition were not necessarily shorter if the director produced multimodal descriptions for their first turn. That is, the advantage for the participants in the multimodal condition may derive from additional benefits other than an immediate advantage for describing stimuli, such as having the option to draw on the gestural and vocal modalities in a way that they see fit their communicative goals best (as laid out above; see also Discussion).
— Trials were shorter if they were guessed correctly or were preceded by trials where the matcher responded.

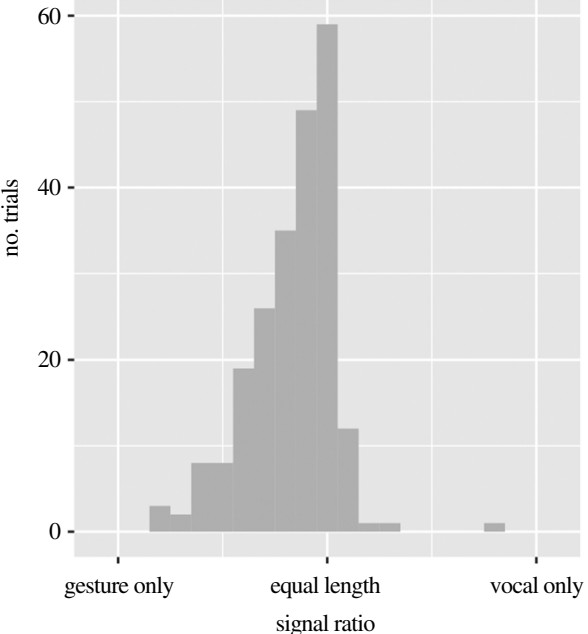

**Figure 6.** Distribution of signals in turns produced by directors in the multimodal condition when describing auditory stimuli. The total time spent producing signals is shown in proportion to the time spent producing vocal signals. A value on the extreme left indicates a ratio of 0 (turn includes only unimodal gestural signals). A value on the extreme right indicates a ratio of 1.0 (turn includes only unimodal vocal signals). The value marked 'equal length' represents a ratio of 0.5 (the same amount of time was dedicated to vocal and gestural signalling).

— In multimodal signals, the vocal component was often shorter than and contained within the gestural component, even when the stimulus to be communicated was auditory in nature.
— There was some evidence that vocal signals were weak when used on their own. For example: vocal-only participants were less efficient than multimodal participants when describing auditory stimuli; vocal-only participants had slightly lower accuracy when describing visual stimuli; and participants in the multimodal condition rarely chose to produce vocal-only signals. The fact that vocal signals were frequently temporally embedded within visual signals may be considered as further corroborating the notion that vocal signals are secondary to visual signals.

## 5. Discussion

Previous experimental studies provided evidence that visible gesture is a particularly expressive means of non-linguistic communication and that multimodality has no advantage over unimodal gestural communication when constructing a communication system [42,103,136,137]. As we note above, the main point of these studies was about motivated (iconic) versus abstract signals, rather than the inherent effectiveness of a particular modality *per se*. However, the present results demonstrate an advantage for communicative settings which offer the possibility to communicate multimodally over purely gestural communication (when signals are non-conventionalized), which we discuss below.

For auditory stimuli, there were no differences between conditions or stimuli types in terms of accuracy. Although ceiling effects may be limiting the ability to differentiate between conditions, this result does not fit the prediction of the direct linkage hypothesis and previous findings that multimodal signals were less accurately understood than gestures. Furthermore, participants in the multimodal condition were more efficient than either of the other two conditions. The average trial time in the multimodal condition was 56% faster (about 4.6 s) than the average trial time in the gestural condition. These results are predicted by the multimodal advantage hypothesis and go against both the gestural advantage hypothesis and the direct linkage hypothesis.

For visual stimuli, accuracy was comparable across the three modality conditions and across stimuli types. On average, participants in the vocal condition were less accurate than participants in the gesture or multimodal condition. This is in line with earlier findings and with the direct linkage hypothesis (participants find it hard to communicate about visual stimuli by vocal means only). Participants in

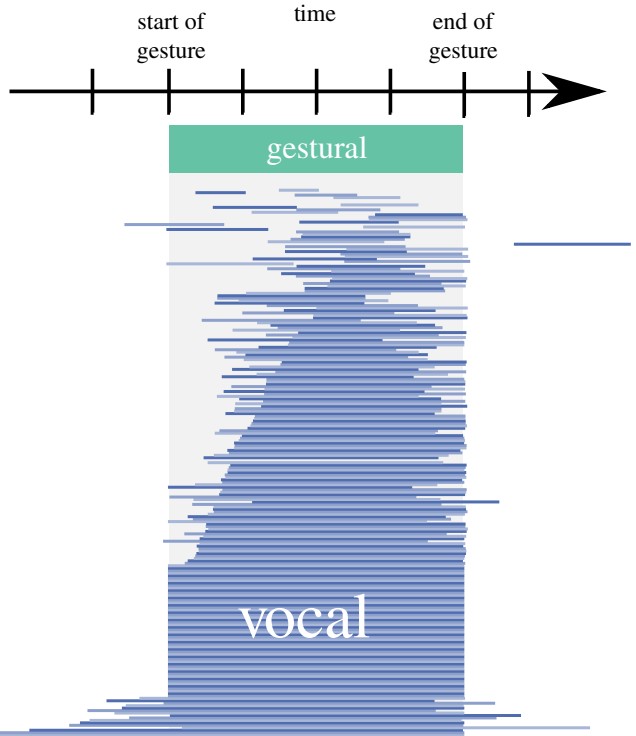

**Figure 7.** The distribution of vocal signals in time, relative to the gestural signal in multimodal turns communicating about acoustic stimuli. Each vocal signal is displayed as a horizontal band (the shades of blue are simply to differentiate different signals). All vocal signals have been scaled so that the corresponding gestural signals are the same length and begin at the same point in time.

the vocal condition were more efficient, though this seems to be at the cost of accuracy. However, the analyses lack the statistical power to tell whether the difference in accuracy is reliable or just due to random differences between participants and stimuli. Participants in the multimodal condition improved their efficiency faster than in the gestural condition. By the final game, participants in the multimodal condition were completing trials about 25% faster (around 2 s) on average than participants in the gestural condition. That is, in contrast to previous studies, we do find that there is evidence of an advantage to be able to choose to communicate multimodally over being restricted to just the gestural modality. This result is in line with the multimodal advantage hypothesis rather than the gestural advantage hypothesis or the direct linkage hypothesis.

More generally our results demonstrate the effectiveness of gesture. For example, in multimodal signals, the vocal component often started later and ended earlier than the visual component. This suggests that gestures were core, if not the primary, components of the signals. Also, when describing visual stimuli in the multimodal condition, participants used mainly unimodal gestures, suggesting that vocalizations were often not needed in addition to gestures. However, when describing auditory stimuli, they used mainly multimodal signals, rather than unimodal vocalizations alone. It was in this latter condition that the multimodal advantage was most pronounced, where the participants had the combined advantage of direct linkage through vocal signals as well as the power of gesture. To summarize, it does appear that gestures can more easily create motivated signals than vocalizations, in line with previous findings from experimental studies using similar paradigms [42,103].

We would like to emphasize that differences in responses to audio and visual stimuli are not necessarily meaningful for theories of language evolution. The audio and visual stimuli were designed with the criteria that they have no conventionalized signals, but they are not matched in several other dimensions (indeed, it would be hard to match stimuli from different domains in terms of distinctiveness or complexity). For example, the audio samples may be easier for participants to link to real multimodal sources, perhaps making multimodal communication more effective for audio stimuli than for visual stimuli. We note that the communicative success for all stimuli was well above chance, and that there was no main effect of stimulus modality for efficiency. In this sense, the stimuli seem to be similarly difficult to communicate, even if the modality of the stimulus may strongly bias which modality is used in communication. However, our main conclusion is not about a difference

between audio and visual stimuli, but about the effectiveness of different modalities, particularly the comparison between gesture-only and multimodality. By the final game, we find that participants in the multimodal condition have achieved greater communicative efficiency than participants in the gesture-only condition for both audio and visual stimuli. This is a key difference between the current study and prior studies. However, it is possible that the choice of stimuli explains the finding that auditory stimuli elicit more multimodal responses than visual stimuli, so we do not draw any strong implications of this result for language evolution. At the very least, then, the choice of stimuli that participants must communicate about can alter the apparent effectiveness of different modalities. In previous experiments, stimuli consisted of items for which there existed conventional signals while in the current experiment stimuli were not associated with conventional labels and contained both stimuli that were suitable for visual signalling and stimuli suitable for vocal signalling. This evened the playing field for gesture and vocalization and meant that participants who could use both modalities had an advantage.

In general, we suspect that the efficiency advantage in the multimodal condition does not derive entirely from multimodality in itself, but from the ability to flexibly deploy modalities to take advantage of the affordances of a given stimulus. This is based on two findings. Firstly, participants in the multimodal condition had the choice to produce multimodal signals, but often produced just unimodal signals. Secondly, whether the signal was multimodal or unimodal did not directly predict trial efficiency. Indeed, even unimodal descriptions in the multimodal condition ended up leading to faster trials than unimodal descriptions in the two unimodal conditions. Therefore, we suggest that the multimodal advantage derived from being able to flexibly deploy modalities by either combining them or using them on their own depending on what best meets current communicative demands. Such an interpretation is very much in line with Kendon's [138] notion of multimodality, which proposed that gesture is best understood 'if we look upon it as an available resource, and try to see how participants deploy it in the light of how they understand how its properties may best meet the current communicational requirements of the interactional situation in which they are taking part' [138, p. 233]. By contrast, participants in the gestural and vocal conditions were forced to encode everything in just one modality, whether the information they intended to convey was best suited to the respective modality or not, thus slowing down communication. In summary, the nature of the referents that communicators need to communicate about will drive the modalities that are used. If there are a mix of different affordances, then multimodally communicating interactants will have an advantage.

## 5.1. Evolution

Previous studies suggest that vocal communication has no advantage over gestural communication, feeding into the dichotomy between 'gesture first' and 'vocal first' hypotheses. In the present study, we find that people tend to use multimodal signals rather than just vocal depictions when the stimulus is auditory. We also find that vocalization is a weaker strategy for communication when used on its own (based on several factors, e.g. lower efficiency when describing auditory stimuli, slightly lower accuracy when describing visual stimuli, participants in the multimodal condition rarely chose to produce vocal-only signals). The fact that vocal signals were frequently temporally embedded within visual signals may be considered as further corroborating the notion that non-conventionalized vocal signals are often secondary to non-conventionalized visual signals. However, in our data, there is a clear advantage to combining vocalizations with gesture. This undermines the dichotomy and suggests a multimodal origin for human language (see also [24–26]; [139], p. 87). For example, Levinson & Holler [22] suggest that the multimodal system of communication modern humans possess emerged out of a stratified evolutionary process. As part of the incremental evolution which culminated in the gesture-plus-speech system in place today, they hypothesize the presence of an early capacity to produce ad hoc gestural displays, most likely on the basis of action sequences such as the ones produced by modern apes [4]. Crucially, such gestural systems are likely to have been enriched with simple vocalizations from a very early stage, with vocalizations being used, first and foremost, to draw attention to the visual displays being produced, but which may also have been iconic in nature [140–142]. Given, then, a minimally powerful system of *multimodal* signalling, human communication would be on track to becoming more complex and expressive. Indeed, as suggested by Levinson & Holler [22], later adaptations would have ensured the development of vocal signalling into a much more powerful and complex medium than in its initial stage, a process crucially influenced by the close interaction of the vocal and visual-gestural modalities. Ultimately, the two expressive modalities

would have continued co-evolving, leading to the fully integrated and dynamic multimodal system humans make use of today. Experimental methods, such as the one used in this study, may help us sharpen these theories.

### 5.1.1. Future work on the role of interaction in language evolution

There is still much to explore regarding how multimodality might confer a benefit in grounding communicative acts. We suggest that interaction may play a role and that experiments similar to this one can help researchers think about this. We make three observations that may be explored further in the future. The first observation is that efficiency improved at a greater rate in the multimodal condition than in the gestural condition (at least for visual stimuli). This suggests that part of the advantage of multimodality is a cumulative effect over many rounds of interaction.

The second observation is that collaboration to address a current problem helped solve future problems. Trials with responses from the matcher were not more likely to be correct and tended to be longer, but they were associated with greater accuracy in subsequent trials. This is in line with a meta-analysis which finds that initiating repair in experimental communication games leads to increased efficiency and accuracy because it enhances the conceptual alignment between participants [116], and analyses of the importance of repair in 'cross-signing' between signers who have no sign language in common [122]. This could be due to better alignment on the signal for the target meaning or (because the matcher must make a forced choice) due to better alignment on the distinctions between the different meanings. Future experiments could explore the effect of matcher responses further.

The third observation relates to participants in the vocal-only condition. There was almost no interaction in this condition (there was only one trial where the matcher responded), perhaps due to the lack of direct visual contact in this condition which prevented the transmission of facial and gestural cues of communication problems (Micklos *et al.* [116] also find fewer repairs in these kinds of experiments when participants are not face-to-face). Instead, the director would produce a vocalization and the matcher would make a guess without attempting to elicit more help. This strategy might have contributed to the reduced trial times, but also reduced accuracy.

These three observations suggest that multimodality may be a key factor in how interaction affects the cultural evolution of language. Especially in the early stages of establishing referential conventions, multimodal signals may help communicators arrive at more effective form-meaning mappings, either because multiple dimensions of the stimulus are represented in the signal or because a single dimension is more robustly signalled via multiple modalities. Combining signals from different modalities may mutually reinforce them, but in addition may help with the interactive negotiation of meaning. In the electronic supplementary material, S3, we give a detailed analysis of interactions between two participants in the multimodal condition. One participant seems to be keen on establishing compositional signals with two gestural components, but their partner sometimes only produces one component. In an extended interaction, the first participant produces two vocalizations, timed with each gesture. This draws attention to the compositional system and their preference for a particular format. Of course, the role of interaction in language evolution may be constrained by the interactive abilities of early human ancestors, though comparative work suggests that at least some basic abilities might have been in place [27,143–146]. In any case, we suggest that the constraints and affordances of interaction should be an important part of future language evolution research.

## 6. Conclusion

The present study explored how the natural affordances of the vocalizations and gestures can be exploited in the context of non-linguistic communication, based on an experimental paradigm of non-linguistic referential communication. Its findings extend previous literature in regard to the role of modality in language emergence. Gestural signalling is indeed a powerful means of improvised communication, but vocal signalling also has a representational potential of its own, that being especially evident in combination with gestural signalling. Indeed, in contradiction to previous experimental literature, we have shown that the ability to use both modalities is a flexible and powerful means of communication, in certain cases being more powerful than unimodal gestural communication. However, this advantage may lie in long-term, cumulative interaction rather than in multimodal signals being immediately easier to understand. Thus, we propose that 'gesture first' theories of language evolution need to be revised to account for the inherently multimodal nature of

modern human communication. Future research should explore more directly the role of improvised multimodal signalling not only in the emergence of referential communicative conventions but also in the development of interactive communication itself.

Ethics. Ethical approval for the study was obtained by Judith Holler from the Ethics Committee for Social Science Faculty (ECSW) of Radboud University, Nijmegen (ECG2013-1403-096). Informed consent for scientific use of the recordings was obtained from all participants.

Data accessibility. The coded data, experimental software, results and analysis are available at GitHub: https://github.com/seannyD/multimodalCommunicationGame and are archived within the Zenodo repository: https://doi.org/10.5281/zenodo.3333408. Data are also available within the electronic supplementary material.

Authors' contributions. V.M.S. conceived of the study. All authors participated in the design of the study and the coding scheme. V.M.S. coordinated the study and carried out the data collection and coding. S.G.R. and V.M.S. carried out the data processing and statistical analysis. V.M.S. drafted the original manuscript. All authors helped write the final manuscript. All authors gave final approval for publication.

Competing interests. We declare we have no competing interests.

Funding. S.G.R. and J.H. were supported by a European Research Council Advanced grant no. 269484 INTERACT to Stephen C. Levinson. S.G.R. was additionally supported by a Leverhulme early career fellowship (ECF-2016-435). We thank the Max Planck Society for additional support.

Acknowledgements. We wish to thank Frédérique Schless for the reliability coding, Gerardo Ortega for sharing data on the use of the 'sleeping' gesture by naive participants, and Ashley Micklos for sharing her data on repair types (table 3). Thanks to four reviewers for comments that substantially helped us improve our paper.

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
