## [Reviewer comments · Royal Society Open Science]

Review History

RSOS-182056.R0 (Original submission)

Review form: Reviewer 1 (Michael Arbib)

Is the manuscript scientifically sound in its present form?

Yes

Are the interpretations and conclusions justified by the results?

No

Is the language acceptable?

Yes

Is it clear how to access all supporting data?

Yes

Do you have any ethical concerns with this paper?

No

Have you any concerns about statistical analyses in this paper?

I do not feel qualified to assess the statistics

Recommendation?

Major revision is needed (please make suggestions in comments)

Comments to the Author(s)

Section 2.1, Modality and the origins of language

The authors note that manual origins theories of language evolution posit that gesture preceded and led to the origin of spoken language. However, their experimental set-up seems to assume that speech was already available when protolanguage first emerged, sidestepping the debate over whether manual protosign emerged before or after the evolution of vocal control and learning (which nonhuman primates lack). For example, in my gestural origins accounts, early protosign provided the scaffolding for the evolution of vocal learning and control and thence protospeech, but thereafter the path toward language was multimodal (Arbib, 2005). This seems to have long ago bridged “the gap between the two opposing sides” (p.3).

Arbib, M. A. (2005). Interweaving Protosign and Protospeech: Further Developments Beyond the Mirror. *Interaction Studies: Social Behavior and Communication in Biological and Artificial Systems*, 6, 145-171.

The authors state (p.3) that “There is increasing evidence for flexible use of vocal calls in primates (e.g. Seyfarth & Cheney, 2010; Slocombe, Zuberbühler, 2005; Seyfarth & Cheney, 2017),” But is that flexibility truly akin to the vocal control AND LEARNING necessary for human spoken language? I am similarly concerned with the claim (p.4) that “the deep connection between vocal and visual behaviour not only in modern humans but possibly too in other species, both extant and extinct, in the phylogenetic line leading to hominins” since it ignores the mismatch between manual and vocal control in all these species.”

In short, Section 2.1 seemed a distraction from the experiment and could be omitted.

Having said all this, I stress that my disappointment with the experiment and the interpretation of the results in this paper is in no way influenced by my preference for protosign to scaffold an expanding spiral with protospeech. The main concern is that the mismatch between the choice of visual and auditory stimuli invalidates the conclusions the authors draw from their data, and that the data are presented at an abstract level that blocks discussion of interesting details.

2.2, Modern human communicative behaviour

It is correct (p.5) that “people of different cultural and linguistic backgrounds employ verbal as well as visual signals when communicating to each other in everyday contexts.” Agreed. This seems to be enough to justify the authors’ enquiry, without the flawed analysis of human origins.

2.3. Emergence and evolution of human language in the wild and in the lab.

The authors note studies in which (p.7) human participants bootstrapped new communication systems and the researchers in those studies reported that communicative success (or comprehension) was superior for communicative acts involving gesture alone compared to vocal signals alone or even multimodal signals.” This is where the story really starts, as the authors seek to establish cases in which multimodal signals show a processing advantage. But this is unsurprising! If I wish to signal that I want a cup of coffee, pantomime will serve well as long as I am in a context of coffee-drinkers. Tea versus coffee is much harder – I might have to pantomime beverage preparation, but in either case, sound might not help. On the other hand, to bring a cat or dog to another’s mind, imitation the animal’s vocalization would be more effective. This seems to be the point of Table 1. It might help the paper to note these naturalistic observations, then assess whether certain studies use stimuli that bias modality use. The authors do note that it is easy to pantomime “apple” and “banana,” but gloss over the challenge of finding a gesture for

“fruit” in general. (Perhaps offering gestures for each prototype could continue until the observer got the meaning, and then an arbitrary “summary gesture” could be introduced? But this is far outside the scope of this experiment.)

pp. 10-11: “The stimuli used in the experiment consisted of sounds and images without existing conventional labels. Auditory stimuli consisted of 8 sounds resembling both generic natural sounds and human-made/artificial sounds. ...and may be described as: air leaking, paper being crumpled, wings flapping ... Visual stimuli, presented in Fig. 1, consisted of 8 images of circles filled with different patterns and shapes (e.g., lines, triangles, etc.).”

There is an immense mismatch here. The authors note (p.12) that “The auditory and visual stimuli cannot be treated as being equally matched” but in fact they are grossly mismatched. Moreover, the claim that “they were designed to capture sounds and visual patterns that can be perceived in the natural environment” seems to be true only of the auditory stimuli.

5. Results:

Lots of graph and statistics but, for me, no insights. For this, I would need a gallery of “typical” vocal, manual and bimodal gestures produced by the subjects, together with a thoughtful analysis. Figure 3 shows the greater accuracy in signaling auditory over visual stimuli. But this is NOT surprising. The auditory stimuli are everyday ones for which we can readily attempt to replicate the sound or pantomime the behavior.

For the visual stimuli, we would need examples to assess how they might (with low accuracy) be paired with auditory signals. I can imagine ways I might want to “sketch” the visual stimuli, but some pairs seem easily confusable, and it is hard to expect vocalizations to distinguish them save through their repetition trial by trial. This really needs to be analyzed.

The graphs have limited value without a more detailed analysis of this kind, with at least the sketch of a theory that bridges between them. I don’t think the visual stimuli tell us much about language origins. Perhaps (reverting to my earlier discussion) a more useful study would have used short videoclips combining sight and sound, with an attempt to characterize where vocal or manual alone could do all the work of communication, and where one could resolve the ambiguity of the other but not suffice alone.

p.20. I did not understand the significance of Table 2 “Table 2 shows the proportion of trials where the matcher produced at least one turn. This was between 5% and 10% for the multimodal and gesture only conditions, but there was only one trial in the vocal only condition where matchers produced a turn.” Perhaps this account of the Table could be expanded.

I have not reviewed the “full data and analyses” provided in the supplementary materials except to check that it was “graph-based” and did not address my concerns. My impression was that the github format was not user-friendly but I did not invest the effort to check this.

6. Discussion

“When it comes to the results for auditory stimuli, there were no differences between conditions in terms of accuracy. This goes against the prediction of the direct linkage hypothesis and previous findings that multimodal signals were less accurately understood than gestures. Furthermore, participants in the multimodal condition were more efficient than either of the other two conditions. The average trial time in the multimodal condition was 56% faster (about 4.6 seconds) than the average trial time in the gestural condition. This supports the multimodal superiority hypothesis and goes against both the gestural superiority hypothesis and the direct linkage hypothesis.” (p.20)

The conclusion supporting the multimodal superiority hypothesis has little or no power. Where the visual stimuli were completely abstract, the auditory stimuli were in fact the sounds of multimodal everyday situations. Thus, once the director recognized them, manual gestures were readily available if the sound alone did not resolve the viewer’s ambiguity.

Review form: Reviewer 2 (Adam Kendon)

Is the manuscript scientifically sound in its present form?

Yes

Are the interpretations and conclusions justified by the results?

Yes

Is the language acceptable?

Yes

Is it clear how to access all supporting data?

Yes

Do you have any ethical concerns with this paper?

No

Have you any concerns about statistical analyses in this paper?

I do not feel qualified to assess the statistics

Recommendation?

Accept with minor revision (please list in comments)

Comments to the Author(s)

See attached file for my review (Appendix A).

Decision letter (RSOS-182056.R0)

27-Feb-2019

Dear Mr Macuch Silva,

The editors assigned to your paper ("Multimodality and the origin of a novel communication system in face-to-face interaction") have now received comments from reviewers. We would like you to revise your paper in accordance with the referee suggestions which can be found below (not including confidential reports to the Editor). Please note this decision does not guarantee eventual acceptance.

Please submit a copy of your revised paper before 22-Mar-2019. Please note that the revision deadline will expire at 00.00am on this date. If we do not hear from you within this time then it will be assumed that the paper has been withdrawn. In exceptional circumstances, extensions may be possible if agreed with the Editorial Office in advance. We do not allow multiple rounds of revision so we urge you to make every effort to fully address all of the comments at this stage. If deemed necessary by the Editors, your manuscript will be sent back to one or more of the original reviewers for assessment. If the original reviewers are not available, we may invite new reviewers.

To revise your manuscript, log into <http://mc.manuscriptcentral.com/rsos> and enter your Author Centre, where you will find your manuscript title listed under "Manuscripts with

Decisions." Under "Actions," click on "Create a Revision." Your manuscript number has been appended to denote a revision. Revise your manuscript and upload a new version through your Author Centre.

- Data accessibility

If you wish to submit your supporting data or code to Dryad (<http://datadryad.org/>), or modify your current submission to dryad, please use the following link:
<http://datadryad.org/submit?journalID=RSOS&manu=RSOS-182056>

- Competing interests

- Authors' contributions

AB carried out the molecular lab work, participated in data analysis, carried out sequence alignments, participated in the design of the study and drafted the manuscript; CD carried out the statistical analyses; EF collected field data; GH conceived of the study, designed the study,

coordinated the study and helped draft the manuscript. All authors gave final approval for publication.

- Acknowledgements

- Funding statement

on behalf of Dr Atsushi Iriki (Associate Editor) and Professor Essi Viding (Subject Editor)
openscience@royalsociety.org

Comments to Author:

Reviewers' Comments to Author:

Reviewer: 1

Comments to the Author(s)

Section 2.1, Modality and the origins of language

The authors note that manual origins theories of language evolution posit that gesture preceded and led to the origin of spoken language. However, their experimental set-up seems to assume that speech was already available when protolanguage first emerged, sidestepping the debate over whether manual protosign emerged before or after the evolution of vocal control and learning (which nonhuman primates lack). For example, in my gestural origins accounts, early protosign provided the scaffolding for the evolution of vocal learning and control and thence protospeech, but thereafter the path toward language was multimodal (Arbib, 2005). This seems to have long ago bridged "the gap between the two opposing sides" (p.3).

Arbib, M. A. (2005). Interweaving Protosign and Protospeech: Further Developments Beyond the Mirror. *Interaction Studies: Social Behavior and Communication in Biological and Artificial Systems*, 6, 145-171.

The authors state (p.3) that "There is increasing evidence for flexible use of vocal calls in primates (e.g. Seyfarth & Cheney, 2010; Slocombe, Zuberbühler, 2005; Seyfarth & Cheney, 2017)," But is that flexibility truly akin to the vocal control AND LEARNING necessary for human spoken language? I am similarly concerned with the claim (p.4) that "the deep connection between vocal and visual behaviour not only in modern humans but possibly too in other species, both extant and extinct, in the phylogenetic line leading to hominins" since it ignores the mismatch between manual and vocal control in all these species."

In short, Section 2.1 seemed a distraction from the experiment and could be omitted.

Having said all this, I stress that my disappointment with the experiment and the interpretation of the results in this paper is in no way influenced by my preference for protosign to scaffold an expanding spiral with protospeech. The main concern is that the mismatch between the choice of

visual and auditory stimuli invalidates the conclusions the authors draw from their data, and that the data are presented at an abstract level that blocks discussion of interesting details.

2.2, Modern human communicative behaviour

It is correct (p.5) that “people of different cultural and linguistic backgrounds employ verbal as well as visual signals when communicating to each other in everyday contexts.” Agreed. This seems to be enough to justify the authors’ enquiry, without the flawed analysis of human origins.

2.3. Emergence and evolution of human language in the wild and in the lab.

The authors note studies in which (p.7) human participants bootstrapped new communication systems and the researchers in those studies reported that communicative success (or comprehension) was superior for communicative acts involving gesture alone compared to vocal signals alone or even multimodal signals.” This is where the story really starts, as the authors seek to establish cases in which multimodal signals show a processing advantage. But this is unsurprising! If I wish to signal that I want a cup of coffee, pantomime will serve well as long as I am in a context of coffee-drinkers. Tea versus coffee is much harder – I might have to pantomime beverage preparation, but in either case, sound might not help. On the other hand, to bring a cat or dog to another’s mind, imitation the animal’s vocalization would be more effective. This seems to be the point of Table 1. It might help the paper to note these naturalistic observations, then assess whether certain studies use stimuli that bias modality use. The authors do note that it is easy to pantomime “apple” and “banana,” but gloss over the challenge of finding a gesture for “fruit” in general. (Perhaps offering gestures for each prototype could continue until the observer got the meaning, and then an arbitrary “summary gesture” could be introduced? But this is far outside the scope of this experiment.)

pp. 10-11: “The stimuli used in the experiment consisted of sounds and images without existing conventional labels. Auditory stimuli consisted of 8 sounds resembling both generic natural sounds and human-made/artificial sounds. ...and may be described as: air leaking, paper being crumpled, wings flapping ... Visual stimuli, presented in Fig. 1, consisted of 8 images of circles filled with different patterns and shapes (e.g., lines, triangles, etc.).”

There is an immense mismatch here. The authors note (p.12) that “The auditory and visual stimuli cannot be treated as being equally matched” but in fact they are grossly mismatched. Moreover, the claim that “they were designed to capture sounds and visual patterns that can be perceived in the natural environment” seems to be true only of the auditory stimuli.

5. Results:

Lots of graph and statistics but, for me, no insights. For this, I would need a gallery of “typical” vocal, manual and bimodal gestures produced by the subjects, together with a thoughtful analysis. Figure 3 shows the greater accuracy in signaling auditory over visual stimuli. But this is NOT surprising. The auditory stimuli are everyday ones for which we can readily attempt to replicate the sound or pantomime the behavior.

For the visual stimuli, we would need examples to assess how they might (with low accuracy) be paired with auditory signals. I can imagine ways I might want to “sketch” the visual stimuli, but some pairs seem easily confusable, and it is hard to expect vocalizations to distinguish them save through their repetition trial by trial. This really needs to be analyzed.

The graphs have limited value without a more detailed analysis of this kind, with at least the sketch of a theory that bridges between them. I don’t think the visual stimuli tell us much about language origins. Perhaps (reverting to my earlier discussion) a more useful study would have used short video clips combining sight and sound, with an attempt to characterize where vocal or manual alone could do all the work of communication, and where one could resolve the ambiguity of the other but not suffice alone.

p.20. I did not understand the significance of Table 2 “Table 2 shows the proportion of trials where the matcher produced at least one turn. This was between 5% and 10% for the multimodal and gesture only conditions, but there was only one trial in the vocal only condition where matchers produced a turn.” Perhaps this account of the Table could be expanded.

I have not reviewed the “full data and analyses” provided in the supplementary materials except to check that it was “graph-based” and did not address my concerns. My impression was that the github format was not user-friendly but I did not invest the effort to check this.

6. Discussion

“When it comes to the results for auditory stimuli, there were no differences between conditions in terms of accuracy. This goes against the prediction of the direct linkage hypothesis and previous findings that multimodal signals were less accurately understood than gestures. Furthermore, participants in the multimodal condition were more efficient than either of the other two conditions. The average trial time in the multimodal condition was 56% faster (about 4.6 seconds) than the average trial time in the gestural condition. This supports the multimodal superiority hypothesis and goes against both the gestural superiority hypothesis and the direct linkage hypothesis.” (p.20)

The conclusion supporting the multimodal superiority hypothesis has little or no power. Where the visual stimuli were completely abstract, the auditory stimuli were in fact the sounds of multimodal everyday situations. Thus, once the director recognized them, manual gestures were readily available if the sound alone did not resolve the viewer’s ambiguity.

Reviewer: 2

Comments to the Author(s)

See attached file for my review

Author's Response to Decision Letter for (RSOS-182056.R0)

See Appendix B.

RSOS-182056.R1 (Revision)

Review form: Reviewer 3

Is the manuscript scientifically sound in its present form?

No

Are the interpretations and conclusions justified by the results?

Yes

Is the language acceptable?

Yes

Is it clear how to access all supporting data?

Yes

Do you have any ethical concerns with this paper?

No

Have you any concerns about statistical analyses in this paper?

Yes

Recommendation?

Major revision is needed (please make suggestions in comments)

Comments to the Author(s)

Macuch Silva et al used a referential communication task to investigate the putative gain of multimodal signals in describing novel visual objects and sounds. The study included three experimental conditions: visible gestures only, non-linguistic vocalizations only, and both (multimodal communication). The authors assessed two main dependent variables, accuracy, i.e., number of correct choices made by the matcher (forced-choice, out of three possible options per trial), and time efficiency in terms of duration until the matcher gave his/her response

The authors tested three main hypotheses:

- (1) direct linkage hypothesis (prediction: interaction stimulus type x modality, accuracy & efficiency increased for visual stimuli in the visual gestural condition relative to the non-linguistic vocalization condition, and vice versa for the sound stimuli).
- (2) Gestural advantage hypothesis
- (3) Multimodal advantage hypothesis

In line with their hypothesis, Macuch Silvia et al. found significant interactions of modality (visual gesture, vocal, multimodal) x stimulus type (visual, auditory) for both variables, accuracy and efficiency.

For the description of visual objects, the participants were less accurate describing visual objects in the non-linguistic vocalization only condition compared to the other two conditions, visible gestures only and multimodal.

For auditory stimuli, participants in the multimodal condition were more efficient than participants in both the vocal and gestural conditions. For visual stimuli, participants in the vocal condition were more efficient than participants in the gestural and multimodal conditions. These latter findings for the dependent variable time efficiency are based on statistical trends (p-values of those were not reported).

Interestingly, the authors also found a significant interaction modality condition by trial length. Suggesting that participants in the multimodal condition increased their time efficiency faster for the description of visual objects. At that point, I was a bit confused about the variable number of trials played and trial length as labeled in the figure. How do these related to each other?

The authors found partial evidence for the direct linkage hypothesis in terms of accuracy for the description of visual objects. Moreover, they showed a benefit of the possibility to use multimodal signals in terms of time efficiency.

Major points

Results

The transparency of the statistical analyses could be improved. Supplemental information about the analyses were made available, but the description of the analyses in the manuscript are sometimes unclear.

The authors should provide p-values for non-significant effects and full stats for significant effects (beta, t or z score, p values etc.). The authors state that effects were tested by model comparisons, but they do not describe the procedure, i.e., whether they started with a full model, testing whether the exclusion of one fixed effect led to a significant difference in the model fit. Or

bottom-up whether the addition of a fixed effect to a null model led to a significant increase in the explained variance.

Moreover, it is unclear how the posthoc comparisons of sublevel were computed. In my view, all this should be describe in the main part of the manuscript.

In general, I would advise the authors to focus their analyses more on the tested hypotheses, i.e., to include mainly the variables of interest in their statistical models. It is always reassuring if a model explain 87% of the variance. However, it would be interesting to know, which variable contributed to what extent? In particular, how much variance was explained by the main variables that are part of the hypotheses? Some variables may not explain any variance and could therefore be excluded.

Including too many variables always bears the risk of overfitting. Indeed, the authors refer to a problem with the model fit on page 19, line 7

“There was more variation in the efficiency data which allowed for a more expansive statistical model.”

Maybe, their models simply contain too main variables in comparison to the number of data points.

The authors should consider to include only the main variables in the results section and to refer to additional analyses including control variables in the supplementary information. Moreover, they should report which control variable was co-varying significantly with the dependent variable and they should drop all variables that have no significant influence. The disadvantage of the specified linear mixed effect models is the sparse random effect structure. I would try to fit models that include not only random intercepts, but also random slopes of the main variables.

Baayen, R. H., Davidson, D. J., & Bates, D. M. (2008). Mixed-effects modeling with crossed random effects for subjects and items. *Journal of Memory and Language*, 59(4), 390–412.

Jaeger, F. (2008). Categorical data analysis: Away from ANOVAs (transformation or not) and towards logit mixed models. *Journal of Memory and Language*, 59(4), 434–446.

<https://doi.org/10.1016/j.jml.2007.11.007>

For example, if the authors would compute a generalized liner mixed effect model for accuracy (0=incorrect; 1= correct response of the matcher) on the trials level, they could maybe fit mixed effect modal that includes a full random effects structure.

Discussion

The paragraph 6.2 Evolution is missing the connection to the work conducted in the current study. The authors were asked to add this paragraph in the first review. In my view, it would be good to connect their work to the work they describe in this paragraph even if their results do not allow strong conclusions.

Minor points

Improve figure reference in the text.

Figure 6, caption is unclear, no reference to the meaning of the different colors.

The authors reported that the variable trial length milliseconds was log transformed, but they did not give a reason for it and they did not state the distribution used in the linear mixed effect model (Gaussian, poisson, binomial etc).

Page 23, line 32 ,indicating that trial length decreased non-linearly towards a plateau (rephrase sentences)

Page 23, line 35, what do the authors mean with a quadratic effect?

Page 27, line 8-10, rephrase sentences

Page 27, line 19-23, in the summary of the paragraph I would add that the modality of the stimulus plays an important role for the modality chosen for communication

Page 27, line 58, I am not sure whether encoding is the correct word in this context

Page 28, lines 32-34, in my view the first sentence implies that real-time interaction was only possible or present in the multi-modal condition. I am not sure if the authors intended to make this point here.

Page 28, line 54 "This is an addition...." until the end of the paragraph. In my view, the authors should skip this part, I do not see this strong link between their findings and the findings by Dingemanse et al (2015).

Review form: Reviewer 4 (Kim Sterelny)

Is the manuscript scientifically sound in its present form?

Yes

Are the interpretations and conclusions justified by the results?

Yes

Is the language acceptable?

Yes

Is it clear how to access all supporting data?

Yes

Do you have any ethical concerns with this paper?

No

Have you any concerns about statistical analyses in this paper?

I do not feel qualified to assess the statistics

Recommendation?

Accept with minor revision (please list in comments)

Comments to the Author(s)

This is a good, clear, well-structured and sensible paper. As the authors note, one of the main divides in current work on the evolution of language is the split between the gesture-first and the vocalisation-first lines of thought. Recently, perhaps belatedly, there has been a shift away from this dichotomy to the idea that the origins of language were multi-modal, just as language is currently multi-modal.

This paper offers real but modest support for this hypothesis through a set of experiments in which communicating and interacting agents had the goal of communicating identifying information about both visual and auditory stimuli, in conditions in which none of these stimuli had conventional labels, in conditions in which some of the dyads were restricted to vocal attempts; others gestural attempts; and a third group had the option to use multi-modal signals

(but were not required to, and often used gesture alone). The authors demonstrate a modest but real advantage to multi-modal communication; not in accuracy but efficiency, and in the improvement of efficiency: multi-modal dyads get better at getting the message across more rapidly.

The issue is interesting and important, and the paper makes a sound contribution. I see that it has already gone through one round of fairly extensive revision, and I have no serious problems with the current version. However, I do have a series of questions/issues: these are just small exercises in fine-tuning, and I would not expect them to take more than a day or two to do.

Minor Points

1. "There is increasing evidence for flexible use of vocal calls in primates (e.g. Seyfarth & Cheney, 2010; Slocombe, Zuberbühler, 2005; Seyfarth & Cheney, 2017), and research showing that it may be easier to recognise and treat vocalisations as symbolic signals compared to gestures (DeLoache, 2002, 2004; Irvine, 2016)."

Perhaps with the exception of the experiments on chimp alarm calling (and even here some elements of the alarm response seemed insensitive to context), my impression was that most of this flexibility was flexibility in response. Does comparative data support the view that there was sufficient topdown control of vocalization for vocalization to play an important role from the very begins of the transition to language? And what is the evidence of the ability to form new signals?

2. "Although it is clear that the behaviour of modern, linguistic, enculturated humans cannot provide direct evidence for the behaviour of early humans, they do provide a useful model and these experimental results have been used to support theories of language evolution. For example, Fay et al. (2013, 2014) made two inferences (see Goldin-Meadow, 2017): first, that gesture has more potential for motivated signs (iconicity, indexicality) than vocalisations; and second that iconic signals can help bootstrap a communication system."

I agree. But there should be some, though perhaps brief, consideration of the obvious problem: these experimental agents have fully modern theories of mind; they have had a huge amount of experience communicating, and hence have trained expectations about what others communicate about; and if language has resulted in any distinctive adaptations for language and communication, these agents have them.

3. "create motivated signals". Perhaps a sentence of explanation on what a motivated signal is.

4. "and the main effects discussed above should not be driven by random differences between participant's abilities (e.g. some might put more effort into their initial descriptions while others rely more on interaction to converge on the correct meaning)".

Perhaps a bit more needs to be said here, for it does not strike me as obvious that individual differences will be minor. Individual histories might play a significant role here: for example, for some people, charades-type games form an important part of their social life, and for others, not at all. It strikes me as possible that a multi-modal system at a communal level might emerge because of variation in individual defaults across the modalities.

5. "A trial could be as short as the director issuing a description followed by the matcher acknowledging understanding"

Am I right in thinking the director has no information about which three stimuli the matcher has to choose between? If so, one might wonder whether that decreases the ecological validity of the experiment: agents often know that they have to communicate X rather than Y or Z, and so have to focus on the most salient differences between X and Y (and X and Z)

6. I shall assume that the statistical procedures were sound, as I have no expertise in that area.

7. "Directors using vocalisations to describe auditory stimuli almost universally attempted to mimic the target stimuli. It is reasonably complex to produce sounds with human vocal chords to match other sounds, but a simple task for the listener to identify the meaning." This is quite important, as it suggests a strong reliance on iconicity as a strategy, and that has quite restricted utility in the vocal modality (and the gestural, but somewhat less so).

8. "In order to look more closely at descriptions of auditory stimuli by directors in the multimodal condition, Fig. 7 shows the ratio between the length of the visual component and the

length of the auditory component. The visual component of the multimodal signal is often longer than the auditory component.”

This needs a little more explanation: why is this ratio important, especially to the evolutionary questions that frame the experimental work?

9. “We suggest that real-time interaction was a critical factor in the improvement of communication during the experiment, based on three observations. The first observation is that the rate of improvement for efficiency was faster in the multimodal condition (at least for visual stimuli). This suggests that there was a cumulative effect over several rounds of interaction.”

This links back to my earlier reservation about the extent to which these experiments can illuminate the early evolution of language. The directors and the matchers are all experts in communicative interaction. But our ancestors forging early forms of language were not.

Decision letter (RSOS-182056.R1)

28-May-2019

Dear Mr Macuch Silva:

Manuscript ID RSOS-182056.R1 entitled "Multimodality and the origin of a novel communication system in face-to-face interaction" which you submitted to Royal Society Open Science, has been reviewed. The comments of the reviewer(s) are included at the bottom of this letter.

Please submit a copy of your revised paper before 20-Jun-2019. Please note that the revision deadline will expire at 00.00am on this date. If we do not hear from you within this time then it will be assumed that the paper has been withdrawn. In exceptional circumstances, extensions may be possible if agreed with the Editorial Office in advance. We do not allow multiple rounds of revision so we urge you to make every effort to fully address all of the comments at this stage. If deemed necessary by the Editors, your manuscript will be sent back to one or more of the original reviewers for assessment. If the original reviewers are not available we may invite new reviewers.

- Ethics statement

If your study uses humans or animals please include details of the ethical approval received, including the name of the committee that granted approval. For human studies please also detail

whether informed consent was obtained. For field studies on animals please include details of all permissions, licences and/or approvals granted to carry out the fieldwork.

- Data accessibility

- Competing interests

- Authors' contributions

- Acknowledgements

- Funding statement

Please note that the journal's editors have made an exception in this case and are allowing a further round of major revision: no further revisions will be possible, so we urge you to make every effort to respond satisfactorily to the reviewers' concerns, and include these responses in your revised manuscript.

Kind regards,
Andrew Dunn

Royal Society Open Science Editorial Office
 Royal Society Open Science
 openscience@royalsociety.org

on behalf of Dr Atsushi Iriki (Associate Editor) and Essi Viding (Subject Editor)
 openscience@royalsociety.org

Reviewer comments to Author:
 Reviewer: 3

Comments to the Author(s)

Macuch Silva et al used a referential communication task to investigate the putative gain of multimodal signals in describing novel visual objects and sounds. The study included three experimental conditions: visible gestures only, non-linguistic vocalizations only, and both (multimodal communication). The authors assessed two main dependent variables, accuracy, i.e., number of correct choices made by the matcher (forced-choice, out of three possible options per trial), and time efficiency in terms of duration until the matcher gave his/her response

The authors tested three main hypotheses:

- (1) direct linkage hypothesis (prediction: interaction stimulus type x modality, accuracy & efficiency increased for visual stimuli in the visual gestural condition relative to the non-linguistic vocalization condition, and vice versa for the sound stimuli).
- (2) Gestural advantage hypothesis
- (3) Multimodal advantage hypothesis

In line with their hypothesis, Macuch Silvia et al. found significant interactions of modality (visual gesture, vocal, multimodal) x stimulus type (visual, auditory) for both variables, accuracy and efficiency.

For the description of visual objects, the participants were less accurate describing visual objects in the non-linguistic vocalization only condition compared to the other two conditions, visible gestures only and multimodal.

For auditory stimuli, participants in the multimodal condition were more efficient than participants in both the vocal and gestural conditions. For visual stimuli, participants in the vocal condition were more efficient than participants in the gestural and multimodal conditions. These latter findings for the dependent variable time efficiency are based on statistical trends (p-values of those were not reported).

Interestingly, the authors also found a significant interaction modality condition by trial length. Suggesting that participants in the multimodal condition increased their time efficiency faster for the description of visual objects. At that point, I was a bit confused about the variable number of trials played and trial length as labeled in the figure. How do these related to each other?

The authors found partial evidence for the direct linkage hypothesis in terms of accuracy for the description of visual objects. Moreover, they showed a benefit of the possibility to use multimodal signals in terms of time efficiency.

Major points

Results

The transparency of the statistical analyses could be improved. Supplemental information about the analyses were made available, but the description of the analyses in the manuscript are sometimes unclear.

The authors should provide p-values for non-significant effects and full stats for significant effects (beta, t or z score, p values etc.). The authors state that effects were tested by model

comparisons, but they do not describe the procedure, i.e., whether they started with a full model, testing whether the exclusion of one fixed effect led to a significant difference in the model fit. Or bottom-up whether the addition of a fixed effect to a null model led to a significant increase in the explained variance.

Moreover, it is unclear how the posthoc comparisons of sublevel were computed. In my view, all this should be describe in the main part of the manuscript.

In general, I would advise the authors to focus their analyses more on the tested hypotheses, i.e., to include mainly the variables of interest in their statistical models. It is always reassuring if a model explain 87% of the variance. However, it would be interesting to know, which variable contributed to what extent? In particular, how much variance was explained by the main variables that are part of the hypotheses? Some variables may not explain any variance and could therefore be excluded.

Including too many variables always bears the risk of overfitting. Indeed, the authors refer to a problem with the model fit on page 19, line 7

“There was more variation in the efficiency data which allowed for a more expansive statistical model.”

Maybe, their models simply contain too main variables in comparison to the number of data points.

The authors should consider to include only the main variables in the results section and to refer to additional analyses including control variables in the supplementary information. Moreover, they should report which control variable was co-varying significantly with the dependent variable and they should drop all variables that have no significant influence. The disadvantage of the specified linear mixed effect models is the sparse random effect structure. I would try to fit models that include not only random intercepts, but also random slopes of the main variables.

Baayen, R. H., Davidson, D. J., & Bates, D. M. (2008). Mixed-effects modeling with crossed random effects for subjects and items. *Journal of Memory and Language*, 59(4), 390–412.

Jaeger, F. (2008). Categorical data analysis: Away from ANOVAs (transformation or not) and towards logit mixed models. *Journal of Memory and Language*, 59(4), 434–446.

<https://doi.org/10.1016/j.jml.2007.11.007>

For example, if the authors would compute a generalized liner mixed effect model for accuracy (0=incorrect; 1= correct response of the matcher) on the trials level, they could maybe fit mixed effect modal that includes a full random effects structure.

Discussion

The paragraph 6.2 Evolution is missing the connection to the work conducted in the current study. The authors were asked to add this paragraph in the first review. In my view, it would be good to connect their work to the work they describe in this paragraph even if their results do not allow strong conclusions.

Minor points

Improve figure reference in the text.

Figure 6, caption is unclear, no reference to the meaning of the different colors.

The authors reported that the variable trial length milliseconds was log transformed, but they did not give a reason for it and they did not state the distribution used in the linear mixed effect model (Gaussian, poisson, binomial etc).

Page 23, line 32 ,indicating that trial length decreased non-linearly towards a plateau (rephrase sentences)

Page 23, line 35, what do the authors mean with a quadratic effect?

Page 27, line 8-10, rephrase sentences

Page 27, line 19-23, in the summary of the paragraph I would add that the modality of the stimulus plays an important role for the modality chosen for communication

Page 27, line 58, I am not sure whether encoding is the correct word in this context

Page 28, lines 32-34, in my view the first sentence implies that real-time interaction was only possible or present in the multi-modal condition. I am not sure if the authors intended to make this point here.

Page 28, line 54 "This is an addition..." until the end of the paragraph. In my view, the authors should skip this part, I do not see this strong link between their findings and the findings by Dingemans et al (2015).

Reviewer: 4

Comments to the Author(s)

This is a good, clear, well-structured and sensible paper. As the authors note, one of the main divides in current work on the evolution of language is the split between the gesture-first and the vocalisation-first lines of thought. Recently, perhaps belatedly, there has been a shift away from this dichotomy to the idea that the origins of language were multi-modal, just as language is currently multi-modal.

This paper offers real but modest support for this hypothesis through a set of experiments in which communicating and interacting agents had the goal of communicating identifying information about both visual and auditory stimuli, in conditions in which none of these stimuli had conventional labels, in conditions in which some of the dyads were restricted to vocal attempts; others gestural attempts; and a third group had the option to use multi-modal signals (but were not required to, and often used gesture alone). The authors demonstrate a modest but real advantage to multi-modal communication; not in accuracy but efficiency, and in the improvement of efficiency: multi-modal dyads get better at getting the message across more rapidly.

The issue is interesting and important, and the paper makes a sound contribution. I see that it has already gone through one round of fairly extensive revision, and I have no serious problems with the current version. However, I do have a series of questions/issues: these are just small exercises in fine-tuning, and I would not expect them to take more than a day or two to do.

Minor Points

1. "There is increasing evidence for flexible use of vocal calls in primates (e.g. Seyfarth & Cheney, 2010; Slocombe, Zuberbühler, 2005; Seyfarth & Cheney, 2017), and research showing that it may be easier to recognise and treat vocalisations as symbolic signals compared to gestures (DeLoache, 2002, 2004; Irvine, 2016)."

Perhaps with the exception of the experiments on chimp alarm calling (and even here some elements of the alarm response seemed insensitive to context), my impression was that most of this flexibility was flexibility in response. Does comparative data support the view that there was sufficient topdown control of vocalization to play an important role from the very begins of the transition to language? And what is the evidence of the ability to form new signals?

2. "Although it is clear that the behaviour of modern, linguistic, enculturated humans cannot provide direct evidence for the behaviour of early humans, they do provide a useful model and

these experimental results have been used to support theories of language evolution. For example, Fay et al. (2013, 2014) made two inferences (see Goldin-Meadow, 2017): first, that gesture has more potential for motivated signs (iconicity, indexicality) than vocalisations; and second that iconic signals can help bootstrap a communication system."

I agree. But there should be some, though perhaps brief, consideration of the obvious problem: these experimental agents have fully modern theories of mind; they have had a huge amount of experience communicating, and hence have trained expectations about what others communicate about; and if language has resulted in any distinctive adaptations for language and communication, these agents have them.

3. "create motivated signals". Perhaps a sentence of explanation on what a motivated signal is.

4. "and the main effects discussed above should not be driven by random differences between participant's abilities (e.g. some might put more effort into their initial descriptions while others rely more on interaction to converge on the correct meaning)".

Perhaps a bit more needs to be said here, for it does not strike me as obvious that individual differences will be minor. Individual histories might play a significant role here: for example, for some people, charades-type games form an important part of their social life, and for others, not at all. It strikes me as possible that a multi-modal system at a communal level might emerge because of variation in individual defaults across the modalities.

5. "A trial could be as short as the director issuing a description followed by the matcher acknowledging understanding"

Am I right in thinking the director has no information about which three stimuli the matcher has to choose between? If so, one might wonder whether that decreases the ecological validity of the experiment: agents often know that they have to communicate X rather than Y or Z, and so have to focus on the most salient differences between X and Y (and X and Z)

6. I shall assume that the statistical procedures were sound, as I have no expertise in that area.

7. "Directors using vocalisations to describe auditory stimuli almost universally attempted to mimic the target stimuli. It is reasonably complex to produce sounds with human vocal chords to match other sounds, but a simple task for the listener to identify the meaning." This is quite important, as it suggests a strong reliance on iconicity as a strategy, and that has quite restricted utility in the vocal modality (and the gestural, but somewhat less so).

8. "In order to look more closely at descriptions of auditory stimuli by directors in the multimodal condition, Fig. 7 shows the ratio between the length of the visual component and the length of the auditory component. The visual component of the multimodal signal is often longer than the auditory component."

This needs a little more explanation: why is this ratio important, especially to the evolutionary questions that frame the experimental work?

9. "We suggest that real-time interaction was a critical factor in the improvement of communication during the experiment, based on three observations. The first observation is that the rate of improvement for efficiency was faster in the multimodal condition (at least for visual stimuli). This suggests that there was a cumulative effect over several rounds of interaction."

This links back to my earlier reservation about the extent to which these experiments can illuminate the early evolution of language. The directors and the matchers are all experts in communicative interaction. But our ancestors forging early forms of language were not.

Author's Response to Decision Letter for (RSOS-182056.R1)

See Appendix C.

RSOS-182056.R2 (Revision)

Review form: Reviewer 3

Is the manuscript scientifically sound in its present form?

No

Are the interpretations and conclusions justified by the results?

No

Is the language acceptable?

Yes

Do you have any ethical concerns with this paper?

No

Have you any concerns about statistical analyses in this paper?

Yes

Recommendation?

Major revision is needed (please make suggestions in comments)

Comments to the Author(s)

Abstract

1) The results suggest that even in the absence of conventional signals, gesture is a powerful mode of communication, but that there are also advantages to multimodality.

->what did they authors intend to express?

- 1) Do gestures have advantages to auditory signal?
- 2) or do gestures have advantages to multimodal signals?
- 3) or do multimodal signals have advantages to gestural signals?

Introduction

2) Pg. 3. L. 30, "...use rigorous statistical methods"

In my view, the authors have not yet applied rigorous statistical methods. See comments below, e.g. about multiple comparison correction for sublevel comparison.

3) Pg. 6 L.58 "For example, gestures enhance neural processing of speech comprehension especially in noisy contexts..."

..., guide visual-spatial attention in healthy and clinical populations (Beattie, Webster, & Ross, 2010; Preisig et al., 2015). Moreover, they help patients with language disorders to express themselves (Hogrefe et al., 2012; van Nispen et al., 2016)

4) Pg. 11 L. 38-50. "In principle, we could find that participants in this condition were more efficient than those in the other conditions without any multimodal signals (simultaneous vocal and gestural signals) being produced. In fact, although participants did produce multimodal signals, we do not find a multimodal advantage in the sense that multimodal signals predict higher efficiency or accuracy."

These sentences are contradicting each other.

Results

5) Pg 24, L. 35-41 "However, for every trial where a matcher responded, subsequent guesses were more likely to be correct (10 trials where the matcher responded raised the probability of a correct guess in subsequent trials by about 4 percentage points, $\beta = 0.12$, $z=2.39$, $\chi^2(1)=7.11$, $p<.01$, pseudo-R²=0.003)"

What was the variable/fixed factor tested?

6) Pg 25, l. 10-16 "...", which is explained by two separate trends (see Fig. 3). For auditory stimuli, participants in the multimodal condition communicated more efficiently than participants in both the vocal and gestural conditions. For visual stimuli, participants in the vocal condition communicated more efficiently than participants in the gestural and multimodal conditions (though they were also less accurate on average, see section 5.1"

Please provide the p-values and the statistical test scores used for the pairwise comparison!

7) Pg. 24, l. 35 Longer trials were guessed less accurately ($\beta = -1.02$, $z=-6.43$, $\chi^2(1)=38.97$, $p<.001$, pseudo-R²=0.05).

Pg. 25 l. 39. Trials were longer if the guess was incorrect (by about 2.5 seconds, $t=2.93$, $\chi^2(1)=13.2$, $p<.001$, pseudo-R²=0.039)

These results are redundant.

8) Pg. 26, l.12-50

This paragraph needs to be connected to linked to the statistical analyses (pg 25, l. 19-35?). Usually, according to the APA standards, descriptive stats are provided before inferential statistics.

Discussion

9) Pg. 29, l. 28-33, The authors state that "This finding is in line with the multimodal advantage hypothesis"

In my view, this conclusion cannot be drawn, because there was no difference between the vocal condition and the multimodal condition

10) Pg. 31. L. 29 "We find ..., and that vocalization is a weaker strategy in communication when used on its own ..."

This interpretation is not supported by the presented data: the authors did not find a main effect modality for the variable accuracy, nor a main effect of modality condition for the variable efficiency.

11) Pg. 32. L24-25,

..., and was generally enhanced in the auditory condition

Response Letter

12) We have also added a handy summary table of results

Table reference?

13) Comparisons between different modality conditions were obtained by fitting equivalent models with different intercepts."

We did this by re-leveling, and have now added a note on this in the main text

Multiple comparison correction?

14) However, we now provide additional 'minimal' models (see above) which fit the reviewer's request, and we show that the results do not differ.

Actually, the provided minimal model does not fit my request.

Could the authors please test two separate models, including only the fixed factors modality (visual vs. auditory) and stimulus type (gestural, vocal, multimodal). These are the main variables the authors based their hypotheses on.

1) A generalized linear mixed effect model (glmer, lme4), fixed effects modality and stimulus type, including random slopes and intercepts for both fixed effects, for the dependent variable accuracy (0=incorrect, 1=correct trial)

2) A linear mixed effect model (lmer, lme4), fixed effects modality and stimulus type, including random slopes and intercepts for both fixed effects, for the dependent variable trial length

Please provide the betas, Z-score/T-score, SEM and p-values (for lmer, p-values will have to be estimated in a separate step)

Review form: Reviewer 4 (Kim Sterelny)

Is the manuscript scientifically sound in its present form?

Yes

Are the interpretations and conclusions justified by the results?

Yes

Is the language acceptable?

Yes

Do you have any ethical concerns with this paper?

No

Have you any concerns about statistical analyses in this paper?

No

Recommendation?

Accept as is

Comments to the Author(s)

The responses to my earlier minor queries are all fine, and as this is an interesting paper on an important issue, I think it should now be published. I think the authors are probably right that a multi-modal view of early language is superior to any unimodal view. One very minor point:

Is this a typo on page 10; "This we attribute to having the possibility to draw on the two modalities flexibly, using sometimes multimodal and sometimes multimodal signals, and using this flexibility for negotiating meaning in interaction", with the repeat of "multimodal"

Decision letter (RSOS-182056.R2)

22-Aug-2019

Dear Mr Macuch Silva:

Manuscript ID RSOS-182056.R2 entitled "Multimodality and the origin of a novel communication system in face-to-face interaction" which you submitted to Royal Society Open Science, has been reviewed. The comments of the reviewer(s) are included at the bottom of this letter.

Please submit a copy of your revised paper before 14-Sep-2019. Please note that the revision deadline will expire at 00.00am on this date. If we do not hear from you within this time then it will be assumed that the paper has been withdrawn. In exceptional circumstances, extensions may be possible if agreed with the Editorial Office in advance. We do not allow multiple rounds of revision so we urge you to make every effort to fully address all of the comments at this stage. If deemed necessary by the Editors, your manuscript will be sent back to one or more of the original reviewers for assessment. If the original reviewers are not available we may invite new reviewers.

- Ethics statement

- Data accessibility

- Competing interests

- Authors' contributions

- Acknowledgements

- Funding statement

on behalf of Dr Atsushi Iriki (Associate Editor) and Essi Viding (Subject Editor)
openscience@royalsociety.org

Reviewer comments to Author:

Reviewer: 4

The responses to my earlier minor queries are all fine, and as this is an interesting paper on an important issue, I think it should now be published. I think the authors are probably right that a multi-modal view of early language is superior to any unimodal view. One very minor point:

Is this a typo on page 10; "This we attribute to having the possibility to draw on the two modalities flexibly, using sometimes multimodal and sometimes multimodal signals, and using this flexibility for negotiating meaning in interaction", with the repeat of "multimodal"

Reviewer: 3
 Comments to the Author(s)

Abstract

1) The results suggest that even in the absence of conventional signals, gesture is a powerful mode of communication, but that there are also advantages to multimodality.

->what did they authors intend to express?

- 1) Do gestures have advantages to auditory signal?
- 2) or do gestures have advantages to multimodal signals?
- 3) or do multimodal signals have advantages to gestural signals?

Introduction

2) Pg. 3. L. 30, "...use rigorous statistical methods"

In my view, the authors have not yet applied rigorous statistical methods. See comments below, e.g. about multiple comparison correction for sublevel comparison.

3) Pg. 6 L.58 "For example, gestures enhance neural processing of speech comprehension especially in noisy contexts...."

..., guide visual-spatial attention in healthy and clinical populations (Beattie, Webster, & Ross, 2010; Preisig et al., 2015). Moreover, they help patients with language disorders to express themselves (Hogrefe et al., 2012; van Nispen et al., 2016)

4) Pg. 11 L. 38-50. "In principle, we could find that participants in this condition were more efficient than those in the other conditions without any multimodal signals (simultaneous vocal and gestural signals) being produced. In fact, although participants did produce multimodal signals, we do not find a multimodal advantage in the sense that multimodal signals predict higher efficiency or accuracy."

These sentences are contradicting each other.

Results

5) Pg 24, L. 35-41 "However, for every trial where a matcher responded, subsequent guesses were more likely to be correct (10 trials where the matcher responded raised the probability of a correct guess in subsequent trials by about 4 percentage points, $\beta = 0.12$, $z=2.39$, $\chi^2(1)=7.11$, $p<.01$, pseudo-R²=0.003)"

What was the variable/fixed factor tested?

6) Pg 25, l. 10-16 "...", which is explained by two separate trends (see Fig. 3). For auditory stimuli, participants in the multimodal condition communicated more efficiently than participants in both the vocal and gestural conditions. For visual stimuli, participants in the vocal condition communicated more efficiently than participants in the gestural and multimodal conditions (though they were also less accurate on average, see section 5.1"

Please provide the p-values and the statistical test scores used for the pairwise comparison!

7) Pg. 24, l. 35 Longer trials were guessed less accurately ($\beta = -1.02$, $z=-6.43$, $\chi^2(1)=38.97$, $p<.001$, pseudo-R²=0.05).

Pg. 25 l. 39. Trials were longer if the guess was incorrect (by about 2.5 seconds, $t=2.93$, $\chi^2(1)=13.2$, $p<.001$, pseudo-R²=0.039)

These results are redundant.

8) Pg. 26, l.12-50

This paragraph needs to be connected to linked to the statistical analyses (pg 25, l. 19-35?). Usually, according to the APA standards, descriptive stats are provided before inferential statistics.

Discussion

9) Pg. 29, l. 28-33, The authors state that “This finding is in line with the multimodal advantage hypothesis”

In my view, this conclusion cannot be drawn, because there was no difference between the vocal condition and the multimodal condition

10) Pg. 31. L. 29 “We find ..., and that vocalization is a weaker strategy in communication when used on its own ...”

This interpretation is not supported by the presented data: the authors did not find a main effect modality for the variable accuracy, nor a main effect of modality condition for the variable efficiency.

11) Pg. 32. L24-25,

..., and was generally enhanced in the auditory condition

Response Letter

12) We have also added a handy summary table of results
Table reference?

13) Comparisons between different modality conditions were obtained by fitting equivalent models with different intercepts.”

We did this by re-leveling, and have now added a note on this in the main text

Multiple comparison correction?

14) However, we now provide additional ‘minimal’ models (see above) which fit the reviewer’s request, and we show that the results do not differ.

Actually, the provided minimal model does not fit my request.

Could the authors please test two separate models, including only the fixed factors modality (visual vs. auditory) and stimulus type (gestural, vocal, multimodal). These are the main variables the authors based their hypotheses on.

1) A generalized linear mixed effect model (glmer, lme4), fixed effects modality and stimulus type, including random slopes and intercepts for both fixed effects, for the dependent variable accuracy (0=incorrect, 1=correct trial)

2) A linear mixed effect model (lmer, lme4), fixed effects modality and stimulus type, including random slopes and intercepts for both fixed effects, for the dependent variable trial length

Please provide the betas, Z-score/T-score, SEM and p-values (for lmer, p-values will have to be estimated in a separate step)

Author's Response to Decision Letter for (RSOS-182056.R2)

See Appendix D.

RSOS-182056.R3 (Revision)

Review form: Reviewer 3

Is the manuscript scientifically sound in its present form?

No

Are the interpretations and conclusions justified by the results?

Yes

Is the language acceptable?

Yes

Do you have any ethical concerns with this paper?

No

Have you any concerns about statistical analyses in this paper?

No

Recommendation?

Accept with minor revision (please list in comments)

Comments to the Author(s)

I thank the authors for the work they made on the manuscript and for the additional information they provided. They addressed the majority of my concerns.

Minor points

Response Letter

7) Ok, please make this transparent for the reader

8) Method for multiple comparison correction/p-adjustment?

10) If the conclusion is drawn in this context, it should be made transparent for the reader.

Abstract

1.46-49

“The results suggest that even in the absence of conventional signals, gesture is a powerful mode of communication compared with vocalisation, but that there are also advantages to multimodality compared with using gesture alone.”

->Do the authors infer that gesture is the more powerful mode of communication than non-linguistic vocalizations? Or do they mean that gestures are as powerful as vocalisations in this context? Please clarify.

Results

-Please report β , z , chi-square for all fixed effects tested in the models accuracy and efficiency

-Pg. 23, l. 39-43

“For auditory stimuli, participants in the multimodal condition communicated more efficiently than participants in both the vocal ($\beta = .69$, $t=4.4$, Satterthwaite $p<.001$) and gestural conditions ($\beta = .25$, $t=1.7$, Satterthwaite $p=.1$)”

->I would not consider $p<.001$ as a statistical trend. According to my understanding, a statistical trend is considered as a p-value in-between 5% and 10%

-Pg. 23 l. 59 and pg. 24 p-adjusted – what was the correction applied? (see comment response letter)

Decision letter (RSOS-182056.R3)

17-Oct-2019

Dear Mr Macuch Silva:

On behalf of the Editors, I am pleased to inform you that your Manuscript RSOS-182056.R3 entitled "Multimodality and the origin of a novel communication system in face-to-face interaction" has been accepted for publication in Royal Society Open Science subject to minor revision in accordance with the referee suggestions. Please find the referees' comments at the end of this email.

The reviewers and Subject Editor have recommended publication, but also suggest some minor revisions to your manuscript. Therefore, I invite you to respond to the comments and revise your manuscript.

- Ethics statement

- Data accessibility

If you wish to submit your supporting data or code to Dryad (<http://datadryad.org/>), or modify your current submission to dryad, please use the following link:
<http://datadryad.org/submit?journalID=RSOS&manu=RSOS-182056.R3>

- **Competing interests**

- **Authors' contributions**

- **Acknowledgements**

- **Funding statement**

Because the schedule for publication is very tight, it is a condition of publication that you submit the revised version of your manuscript before 26-Oct-2019. Please note that the revision deadline will expire at 00.00am on this date. If you do not think you will be able to meet this date please let me know immediately.

When submitting your revised manuscript, you will be able to respond to the comments made by the referees and upload a file "Response to Referees" in "Section 6 - File Upload". You can use this to document any changes you make to the original manuscript. In order to expedite the

processing of the revised manuscript, please be as specific as possible in your response to the referees.

Kind regards,
Anita Kristiansen
Editorial Coordinator
Royal Society Open Science
openscience@royalsociety.org

on behalf of Dr Atsushi Iriki (Associate Editor) and Essi Viding (Subject Editor)
openscience@royalsociety.org

Reviewer comments to Author:
Reviewer: 3

Comments to the Author(s)

I thank the authors for the work they made on the manuscript and for the additional information they provided. They addressed the majority of my concerns.

Minor points

Response Letter

- 7) Ok, please make this transparent for the reader
- 8) Method for multiple comparison correction/p-adjustment?
- 10) If the conclusion is drawn in this context, it should be made transparent for the reader.

Abstract

1.46-49

"The results suggest that even in the absence of conventional signals, gesture is a powerful mode of communication compared with vocalisation, but that there are also advantages to multimodality compared with using gesture alone."

->Do the authors infer that gesture is the more powerful mode of communication than non-linguistic vocalizations? Or do they mean that gestures are as powerful as vocalisations in this context? Please clarify.

Results

-Please report β , z , chi-square for all fixed effects tested in the models accuracy and efficiency

-Pg. 23, l. 39-43

"For auditory stimuli, participants in the multimodal condition communicated more efficiently than participants in both the vocal ($\beta = .69$, $t=4.4$, Satterthwaite $p<.001$) and gestural conditions ($\beta = .25$, $t=1.7$, Satterthwaite $p=.1$)"

->I would not consider $p<.001$ as a statistical trend. According to my understanding, a statistical trend is considered as a p-value in-between 5% and 10%

-Pg. 23 l. 59 and pg. 24 p-adjusted - what was the correction applied? (see comment response letter)

Author's Response to Decision Letter for (RSOS-182056.R3)

See Appendix E.

Decision letter (RSOS-182056.R4)

27-Nov-2019

Dear Mr Macuch Silva,

It is a pleasure to accept your manuscript entitled "Multimodality and the origin of a novel communication system in face-to-face interaction" in its current form for publication in Royal Society Open Science.

on behalf of Dr Atsushi Iriki (Associate Editor) and Professor Essi Viding (Subject Editor)
openscience@royalsociety.org

Appendix A

Review of manuscript RSOS-182056

Description of the Study

The study reported in this paper explores how the gestural modality compares in communicative efficacy with the vocal non-verbal modality and with a combination of the two, in the context of a situation in which one person (the “Director”) tries to indicate the nature of an “object” to a recipient (the “Matcher”) so that the Matcher can indicate what the object is that the Director is trying to convey. The objects were either abstract drawings (visual objects) or they were sounds of various kinds (auditory objects). Director and Matcher in this situation can only use bodily movements or non-verbal verbalizations to communicate. In one condition they can only use body movements; in another condition they can only use non-verbal vocalizations; in a third condition they are free to use either or both together, as they choose.

In the situation devised by the authors the “Matcher” was to identify which “object” the Director was trying to convey, by selecting one out of three possible objects that were made available on each trial on a computer screen for visual objects or via earphones for auditory objects. 30 participants (all Dutch speakers) were divided into 15 pairs who were assigned to one of three conditions: use only visible action, use only vocalisation, or be free to use both modalities as they wished. Within each pair the roles of Matcher and Director were alternated. The trials in the experiment for each pair was organized into blocks, one block given only visual stimuli, another block given only auditory stimuli.

An attempt was made to ensure that the objects to be communicated could not be easily conveyed using conventional means. The visual objects were a set of circles each one filled differently, for example with multiple small circles, hatching, small angular objects, a combination of lines and shapes, and so on. The auditory objects were sounds, presented via headphones, such as bubbling water, air leaking from an inflated balloon, wind blowing in a tree, and so on.

The idea was to try to put people in a situation in which they were not able to use existing conventional code systems with which they might be familiar, forcing them to invent means of conveying the objects presented to them in this study in whatever way they could.

The number of objects correctly identified in each trial was taken as a measure of Accuracy, and the length of time it took for the matcher to arrive at a selection was regarded as a measure of Efficiency.

The experiment described here was considered to be an improvement on similar experiments conducted by other researchers (specifically an experiment by Nicolas Fay and colleagues, published in *Frontiers in Psychology* in 2014) insofar as here, in this experiment, the participants were forced to create utterly novel means of conveying the object. In the Fay study the objects selected for transmission were listed by English words and the possibility that they could be transmitted by using conventional kinds of expression was thus not ruled out - this was especially true for their transmission by gesture.

A theoretical rationale for studies of this sort (which is followed in the present paper) is to try to establish what might have been the more efficacious modality when humans were first developing language: whether this might have been the gestural modality (“gesture first”) or the vocal modality (“speech first”). If modern humans are placed in a situation where they have to communicate *de novo*, the modality which they prefer to use perhaps can be taken to indicate the

modality that might have been preferred when humans first attempted to communicate linguistically. Thus, in the experiment just mentioned by Fay, et al., of which the present study is a refinement, it was found that gesture was far and away the most communicatively efficacious as compared to the nonverbal vocal modality. The authors concluded that gesture is thus an excellent modality to use in creating a new language. This being so, this could be taken as supporting the “gesture first” theory of language origins.

Results.

In the present experiment it was found that Matchers were more accurate when Directors used visual signals to convey visual objects and least accurate when Directors used vocal signals to convey visual objects. When Directors were free to use both visual signals and vocal signals (the “multimodal” condition) Matchers improved in their accuracy in identifying auditory objects, but this made no difference in Matchers’ accuracy when identifying visual objects.

The authors interpret these findings to mean that when the participants are free to use either modality, or both at the same time to communicate (the “multimodal” condition) they can be more efficacious in communication, especially when the objects to be communicated are auditory objects. They conclude, in the light of this, that being able to use either or both modalities as one chooses is the most versatile and flexible approach and, accordingly, this means that “ ‘gesture first’ theories of language evolution need to be revised in order to account for the multimodal nature of modern human communication” (p. 23).

Comments.

(1) The authors offer no examples of what Directors actually did in their efforts to convey visual shapes gesturally or vocally, or auditory shapes vocally or with gesture. Given the attempt in this experiment to ensure that the shapes and sounds used as objects had no features that could easily be conveyed by any sort of conventionalised gestures or vocal noises, it would have been interesting to know what kinds of things the Directors did do to try to convey these objects.

(2) Evidently, within trials, Matchers could respond to the Director with “turns”, but I think it should be clarified as to the nature of these “turns”. It does not seem to me that it is made clear what these “turns” were or what was achieved by them. Evidently the Matchers could enter into discussion with their Directors, but how they could do so needs to be clarified (this was made clear in the Fay et al. 2014 study, it should be done here also) Table 2 purported to provide data about “percentage of trials where the matcher produces at least one turn”. But why should Matchers produce any turns? What were these turns for and do the authors have any comments on the different frequencies of matcher turns in the different conditions?

(3) With regard to the so-called “multimodal” condition, as I understand this, in this condition Directors were free to use gesture or vocalization, either separately or together as they chose. However, in analyzing the results the authors simply treat “multimodal” as one of the conditions as if, in this condition, the Matchers received from their Directors *both* gesture and sound *together*. Yet the Directors were free, in this condition, to choose what modality to use and how to deploy it. Did they in fact always deploy these modalities together, or did it sometimes happen that they only used one modality, or the two modalities in succession or in alternation, rather than together? And did this make a difference? This is not made clear, but it seems to me that it should have been.

(4) It was also found that the speed with which the task was accomplished in each trial increased as trials progressed. Evidently the participants improved in learning how to do the tasks as they repeated them. It was found, however, that when the Director could use “multimodal” signals the task was always accomplished faster than when either only auditory or visual signals were used. The authors say that the “multimodal” condition was faster because “the combination of vocalisations and gesture boosted communication” - however, what does “boosted communication” really mean? As it stands it is a vacuous expression. How did the combination of “gesture” and “vocalization” actually help the Matcher to arrive more quickly at a decision as to which object was intended? Some analysis of how it did so and in relation to what sorts of objects it did so might have been given.

(5) I confess I have failed to understand what Fig. 6 is supposed to show. It looks as if when the Directors had visual stimuli to convey, when they were free to use whatever modality they preferred they used mostly vocalizations - which seems very odd.

Furthermore, Fig. 6 compares Directors’ turns with Matchers’ turns. Yet the idea that the Matchers could engage in turns and why they would do so remains unclear to me.

(6) While one might agree that “‘gesture first’ theories of language evolution need to be revised in order to account for the multimodal nature of modern human communication”, the experiments presented here might not seem particularly necessary or persuasive for this point of view. There are already very many very good reasons that support the view that hominin communication has *always* been “multimodal”. One can remain doubtful that experiments of this sort can throw much light on what hominins were doing or could do, communicatively, at the dawn of any form of communication that might be regarded as somehow “linguistic” (by the way these authors seem to use “communication” interchangeably with “language” or “linguistic communication” - but of course “language” is only one of a diverse array of means of communication. Authors should not forget that “communication” is very broad in meaning and that “linguistic communication” is communication of a very special kind.).

Experiments such as those described here have their value, of course, but they have this for adding to our understanding of how different modalities may differently facilitate the transmission of different kinds of content in modern humans. They may also throw light on the ingenuity humans can display communicatively when they are put into restricted situations of the sort devised for them in these experiments. It must be remembered that the humans that take part in these experiments are already full-fledged language users and, as such, already have at their command capacities for improvisation when a communicative situation might demand it. Is it fair to assume that our non-language using hominin ancestors were as fully equipped in this way? Perhaps some further critical discussion of how experiments of this sort are relevant to language origins issues could usefully be added.

(7) The exposition in this paper is difficult and sometimes we find vacuous or clumsy expressions. The authors seem particularly fond of the verb “bootstrap” - I think they should think carefully about what it really means and see whether, in some cases, it is being used wrongly or unnecessarily. In many cases “help” might be a better verb to use instead.

(8) References to Ackerman et al. 2014; Irvine 2016; Bohn, Call and Tomasello 2018 are not in bibliography. I did not systematically check each in-text reference with what is listed in the bibliography and vice versa but I urge the authors to do this. There can be nothing more

annoying than finding a reference that is not properly included in the bibliography and, generally speaking, one should not include items in the bibliography that are not cited in the text.

(9) Figures in my copy had no numbers. Numbers should be added to ensure they are not inserted wrongly in the printed version.

Appendix B

Response to reviewers

Reviewer comments in this font.

Author responses in this font.

Summary:

We thank the reviewers for their reviews and respond below. Major changes include:

- Inclusion of a qualitative analysis of the communicative strategies of the participants.**
- More nuanced discussion of different theories of how modality relates to language evolution.**

Reviewer 1's main concern is about a mismatch between audio and visual stimuli. We agree that the stimuli are not matched, as acknowledged in the manuscript, but we don't think that this introduces a confound. There is no main effect of stimulus type, and we find a multimodal advantage for both types of stimuli. We have tried to clarify this point below and in the text.

We also received comments from other colleagues, and have clarified various other points.

Reviewer: 1

Comments to the Author(s)

Section 2.1, Modality and the origins of language

The authors note that manual origins theories of language evolution posit that gesture preceded and led to the origin of spoken language. However, their experimental set-up seems to assume that speech was already available when protolanguage first emerged, sidestepping the debate over whether manual protosign emerged before or after the evolution of vocal control and learning (which nonhuman primates lack). For example, in my gestural origins accounts, early protosign provided the scaffolding for the evolution of vocal learning and control and thence protospeech, but thereafter the path toward language was multimodal (Arbib, 2005). This seems to have long ago bridged "the gap between the two opposing sides" (p.3).

Arbib, M. A. (2005). Interweaving Protosign and Protospeech: Further Developments Beyond the Mirror. *Interaction Studies: Social Behavior and Communication in Biological and Artificial Systems*, 6, 145-171.

R1: This point is well taken, though we note that there is a difference between the nuances of original claims and how they are interpreted in the subsequent literature. We conducted a review of papers that cite the experimental work (Fay et al., 2013; 2014), and found that polarisation has crept back into the debate. For example, even Fay et al. (2014) interpret Arbib's work as supporting a unimodal theory of the origins of language (see below). We include quotes from this review in our supporting materials. Our main point is that the experimental evidence has been used to speak to

this debate, and we'd like to supplement it with new data. We now discuss this in the paper:

“We should note that, although “gesture-first” and “speech-first” views are often pitted against each other, in reality many suggest a middle ground. For example, Arbib (2005b) suggests that gesture was important at a very early stage of language evolution because it provided a scaffolding for the evolution of vocal learning, but that communication would have been multimodal as soon as an ability to learn vocalisations was in place. Similarly, Fay, Arbib & Garrod (2013) interpret the findings of their experiment to support a multimodal origin of language. However, we note that the polarisation has crept back into the debate (see Perlman, 2017). For example, the follow-up experimental study by Fay et al. (2014) cites Arbib’s work as supporting one view over the other: “This view [the theory of language origins from non-linguistic vocalisations] is challenged by a competing explanation; that language originated through gesture (Hewes, 1973; Corballis, 2003; Arbib, 2005)” (Fay et al., 2014, pp. 2). We discuss potential reasons for this below.

...

[Fay et al., 2013; 2014] found that communicative success or comprehension was better for gestural rather than vocal communication, and that multimodal communication did not provide any advantage compared to gesture alone. Although it is clear that the behaviour of modern, linguistic, enculturated humans cannot provide direct evidence for the behaviour of early humans, they do provide a useful model and these experimental results have been used to support theories of language evolution. For example, Fay et al. (2013, 2014) made two inferences (see Goldin-Meadow, 2017): first, that gesture has more potential for motivated signs (iconicity, indexicality) than vocalisations; and second that iconic signals can help bootstrap a communication system. They conclude that gesture would have been an important part of early communication systems, to the extent that gesture helps convey meanings through iconicity. Fay et al. (2013) are careful to note that “rather than commit to a gestural origin, we prefer an origin in which humans initially communicated using motivated signs (icons and indices), whether gestural or vocal (i.e., a multimodal origin)” (p. 1365). That is, the results were intended to suggest that motivated (iconic) signs have an advantage over abstract signs when bootstrapping a language, rather than claim that this process did not involve vocalisations or multimodality. However, subsequent studies have interpreted these results in a variety of ways, including that they support a “gesture-first” theory of language evolution (see supporting materials). Part of the reason might be that these experimental studies compared the communicative effectiveness of modalities, rather than directly measure the potential for each modality to create motivated signals (see Lister, et al., 2015; Perlman et al., 2018). In any case, the common understanding is that experimental evidence supports the view that multimodality does not provide any advantage during the process of language emergence above and beyond the gestural modality. Our study seeks to provide further experimental evidence that tests this interpretation.”

The authors state (p.3) that “There is increasing evidence for flexible use of vocal calls in primates (e.g. Seyfarth & Cheney, 2010; Slocombe, Zuberbühler, 2005; Seyfarth & Cheney, 2017),” But is that flexibility truly akin to the vocal control AND LEARNING necessary for

human spoken language? I am similarly concerned with the claim (p.4) that “the deep connection between vocal and visual behaviour not only in modern humans but possibly too in other species, both extant and extinct, in the phylogenetic line leading to hominins” since it ignores the mismatch between manual and vocal control in all these species.”

R2: In terms of the link between vocal and visual behaviour, we are simply citing the work of others. In addition, we follow Fay, Arbib & Garrod (2013) by referring readers to Pollick & de Waal (2007) for evidence of multimodality in ape communication. However, the reviewer is right to point out that there is a difference between vocal communication and *learned* vocal communication. We have added the following to the paper:

“Of course, there are questions about whether the ability to learn vocal signals was present for our ancestors before the emergence of symbolic language (see e.g. Arbib, 2005a). However, Perlman (2017) argues that there is at least some evidence of vocal learning in non-human primates, and that this might plausibly allow a fully multimodal system right from the start.”

For the reviewer’s convenience, here is Perlman’s argument that relates to our study:

“In addition, many researchers of ape communication are also coming to advocate for a more deeply multimodal evolution of language (Tagliapietra et al., 2011; Liebal, Waller, Burrows, & Slocumbe, 2013). For example, Leavens (2003) suggested that, “Because visual and vocal communication seem to be functionally linked in extant apes, language may have been multimodal from its inception” (p. 233). In this respect, Koko’s repertoire of learned vocal and breathing related behaviors may be a revealing example of the capacity of great apes to produce novel behaviors involving multiple actions that are coordinated across modalities. Of hundreds of video-recorded instances, Koko performed the vast majority within a larger behavioral complex that combined vocal and oral articulatory movements with various manual gestures and praxic actions (Perlman & Clark, 2015). For example, she combined blowing (bilabial friction) with various manual gestures, including bringing a single open palm to her mouth, bringing both hands perpendicular to her mouth, and bringing both hands over her mouth.”

In short, Section 2.1 seemed a distraction from the experiment and could be omitted.

R3: We hope that the added discussion above helps make this section more relevant for the experiment.

Having said all this, I stress that my disappointment with the experiment and the interpretation of the results in this paper is in no way influenced by my preference for protosign to scaffold an expanding spiral with protospeech. The main concern is that the mismatch between the choice of visual and auditory stimuli invalidates the conclusions the authors draw from their data, and that the data are presented at an abstract level that blocks discussion of interesting details.

(see response to point 2.3, pp.10-11 below)

2.2, Modern human communicative behaviour

It is correct (p.5) that “people of different cultural and linguistic backgrounds employ verbal as well as visual signals when communicating to each other in everyday contexts.” Agreed. This seems to be enough to justify the authors’ enquiry, without the flawed analysis of human origins.

R4: The intention of this section was also to link the work on multimodality to the literature on cognition and interaction. We have included more details about how this is relevant in later sections.

2.3. Emergence and evolution of human language in the wild and in the lab.

The authors note studies in which (p.7) human participants bootstrapped new communication systems and the researchers in those studies reported that communicative success (or comprehension) was superior for communicative acts involving gesture alone compared to vocal signals alone or even multimodal signals.” This is where the story really starts, as the authors seek to establish cases in which multimodal signals show a processing advantage. But this is unsurprising! If I wish to signal that I want a cup of coffee, pantomime will serve well as long as I am in a context of coffee-drinkers. Tea versus coffee is much harder – I might have to pantomime beverage preparation, but in either case, sound might not help. On the other hand, to bring a cat or dog to another’s mind, imitation the animal’s vocalization would be more effective. This seems to be the point of Table 1. It might help the paper to note these naturalistic observations, then assess whether certain studies use stimuli that bias modality use. The authors do note that it is easy to pantomime “apple” and “banana,” but gloss over the challenge of finding a gesture for “fruit” in general. (Perhaps offering gestures for each prototype could continue until the observer got the meaning, and then an arbitrary “summary gesture” could be introduced? But this is far outside the scope of this experiment.)

R5: This is an interesting problem but, as the reviewer suggests, outside the scope of the current experiment. We’ve referred readers to other experiments that cover these issues:

“It is also possible to study how the role of context, the relative difficulty of disambiguating individual meanings or the need to distinguish particular instances from general categories (Konopka & Brown-Schmidt, 2014; Winters, Kirby & Smith, 2014; Silvey, Kirby & Smith, 2015; Winters, Kirby & Smith, 2018), though this is beyond the scope of the current study.”

pp. 10-11: “The stimuli used in the experiment consisted of sounds and images without existing conventional labels. Auditory stimuli consisted of 8 sounds resembling both generic natural sounds and human-made/artificial sounds. ...and may be described as: air leaking, paper being crumpled, wings flapping ... Visual stimuli, presented in Fig. 1, consisted of 8 images of circles filled with different patterns and shapes (e.g., lines, triangles, etc.).” There is an immense mismatch here. The authors note (p.12) that “The auditory and visual stimuli cannot be treated as being equally matched” but in fact they are grossly mismatched. Moreover, the claim that “they were designed to capture sounds and visual patterns that can be perceived in the natural environment” seems to be true only of the auditory stimuli.

R6: The reviewer suggests that the stimuli are not matched. In the text below, they clarify that the nature of this mismatch is that the auditory stimuli have natural sources, while the visual stimuli do not.

As the reviewer notes, we have acknowledged that the stimuli are not matched. It would be difficult to prove that the confusability was equal between sets. The stimuli were designed to meet one criterion: that they had no conventionalised label. In the manuscript, we labelled the sounds with brief descriptions in English, which make them seem more ‘natural’ than they really are. Several of the sounds are completely synthetic (no source in the natural world), and in pilot tests participants did not agree about the source of the sounds. By “visual patterns” we meant basic visual primitives that one might experience in the world (circles, triangles, parallel lines etc), as opposed to formal, platonic shapes. In pilot tests, people described them as ‘nests’ or ‘bubbles’, so it’s not clear how we would prove that these images had no sources in the minds of the participants. In short, we don’t claim that the stimuli are matched.

Nevertheless, we note that there is no main effect of stimuli type for accuracy. That is, in some sense the stimuli are comparably difficult to communicate. And all stimuli are guessed well above chance even from the first trial, showing that people could readily find auditory ways of communicating the visual stimuli. However, our main prediction is not about the stimuli type, but about the communication modalities (effects that are common to both stimulus sets). In this sense, we don’t see how the reviewer’s concern translates into a confound (see below for further discussion, R8, R13). We hope that the qualitative analyses of the solutions that participants produced help further clarify these points.

5. Results:

Lots of graph and statistics but, for me, no insights. For this, I would need a gallery of “typical” vocal, manual and bimodal gestures produced by the subjects, together with a thoughtful analysis.

R7: We welcome this suggestion. We had wrongly thought that the readers might not be interested in the qualitative results and would find it cumbersome and too lengthy. We now include a gallery of typical signals for different signals and have tried to make this as easily digestible and concise as possible. In order to incorporate the audio, we provide this as a self-contained web page that can be opened in a web browser. A copy of this file can be found here: <http://www.correlation-machine.com/Gallery.html> Also, we have included a turn-by-turn analysis of an interactive sequence which we hope will illustrate some aspects of interaction. We also include data on the proportion of different types of repair from a another study which used the videos from our experiment in a more extensive qualitative analysis (Micklos, Macuch Silva & Fay, 2018).

Figure 3 shows the greater accuracy in signaling auditory over visual stimuli. But this is NOT surprising. The auditory stimuli are everyday ones for which we can readily attempt to replicate the sound or pantomime the behavior.

R8: Figure 3 does not show greater accuracy in signalling auditory over visual stimuli. There is no significant main effect of stimulus type predicting accuracy, and it is not part of our main argument.

We have added to the text: “There is no significant main effect of stimulus type.” to make this clearer.

For the visual stimuli, we would need examples to assess how they might (with low accuracy) be paired with auditory signals. I can imagine ways I might want to “sketch” the visual stimuli, but some pairs seem easily confusable, and it is hard to expect vocalizations to distinguish them save through their repetition trial by trial. This really needs to be analyzed.

R9: We have now added a close analysis of an interaction which shows how visual stimuli were disambiguated. We note that these strategies were also used in disambiguating the sounds (long/short sounds, similar to thick/thin lines). We also note that the communicative success for vocalisations is well above chance even from the first trial, and that participants did not receive feedback: the director could not observe the matcher's guess and the matcher was not told whether their guess was correct. This rules out at least a simple strategy of learning arbitrary signs over repeated trials.

The graphs have limited value without a more detailed analysis of this kind, with at least the sketch of a theory that bridges between them. I don't think the visual stimuli tell us much about language origins. Perhaps (reverting to my earlier discussion) a more useful study would have used short videoclips combining sight and sound, with an attempt to characterize where vocal or manual alone could do all the work of communication, and where one could resolve the ambiguity of the other but not suffice alone.

R10: We thank the reviewer for this suggestion and agree this would be a valuable future study. As the experiment stands, we can't speak to these results.

p.20. I did not understand the significance of Table 2 “Table 2 shows the proportion of trials where the matcher produced at least one turn. This was between 5% and 10% for the multimodal and gesture only conditions, but there was only one trial in the vocal only condition where matchers produced a turn.” Perhaps this account of the Table could be expanded.

R11: We have prefaced this with “Participants interacted more in the multimodal and gestural conditions than in the vocal only condition.” And point to the discussion where we discuss the reason for this in more depth.

I have not reviewed the “full data and analyses” provided in the supplementary materials except to check that it was “graph-based” and did not address my concerns. My impression was that the github format was not user-friendly but I did not invest the effort to check this.

R12: Github is a standard repository format for sharing data, though the reviewer is correct that it only contained the quantitative results. We have updated the README

to explain the structure of the repository more fully. And we now include the main analyses files directly as supporting materials.

6. Discussion

“When it comes to the results for auditory stimuli, there were no differences between conditions in terms of accuracy. This goes against the prediction of the direct linkage hypothesis and previous findings that multimodal signals were less accurately understood than gestures. Furthermore, participants in the multimodal condition were more efficient than either of the other two conditions. The average trial time in the multimodal condition was 56% faster (about 4.6 seconds) than the average trial time in the gestural condition. This supports the multimodal superiority hypothesis and goes against both the gestural superiority hypothesis and the direct linkage hypothesis.” (p.20)

The conclusion supporting the multimodal superiority hypothesis has little or no power. Where the visual stimuli were completely abstract, the auditory stimuli were in fact the sounds of multimodal everyday situations. Thus, once the director recognized them, manual gestures were readily available if the sound alone did not resolve the viewer’s ambiguity.

R13: Reviewer 1 suggests that the difference between multimodal and gestural conditions is entirely driven by the issue with the stimuli. This isn't true: participants in the multimodal condition were faster than the gestural condition for auditory stimuli *and* for visual stimuli.

The reviewer goes on to suggest that: "manual gestures were readily available if the sound alone did not resolve the viewer’s ambiguity.". We agree that that people were using multimodal behaviour combinations to aid their communication. However, note that for the vast majority of multimodal trials, gestures temporally precede or are produced at the same time as vocalisations (see section 5.3 and figure 8). That is, participants are not turning to gesture after vocalisations have failed. If anything, they are producing gestures first and backing them up with vocalisations.

We have improved figure 8 to try to communicate this latter point.

To summarise our response to reviewer 1:

The reviewer has legitimate concerns about how well matched the stimuli are. We don't claim that they are matched. The important issue is whether the mismatch leads to a confound for our conclusions. We don't believe that they do: multimodal communication leads to an efficiency advantage over gestural communication whether participants are communicating about audio stimuli or visual stimuli.

Compare this issue to the finding that auditory stimuli elicit more multimodal responses than visual stimuli. We agree that the reviewer’s concern about the mismatch between stimuli is a valid explanation for this difference. We now make it clear that we make no strong interpretations of this finding.

We have added the following to the discussion:

“We note that the audio and visual stimuli in the experiment were designed with the criteria that they have no conventionalised signals, but that they are not matched in several other dimensions. Indeed, it would be hard to match stimuli from different domains in terms of distinctiveness or complexity, though we note that the communicative success for all stimuli was well above chance, and that there was no main effect of stimulus modality for efficiency. In this sense, the stimuli seem to be similarly difficult to communicate. There are other differences, for example the audio samples may be easier for participants to link to real multimodal sources, perhaps making multimodal communication more effective for audio stimuli than for visual stimuli. However, our main conclusion is not about a difference between audio and visual stimuli, but about the effectiveness of different modalities, particularly the comparison between gesture-only and multimodality. We find that participants in the multimodal condition achieve greater communicative efficiency than participants in the gesture-only condition for both audio and visual stimuli. This is a key difference between the current study and prior studies. However, it is possible that the choice of stimuli explains the finding that auditory stimuli elicit more multimodal responses than visual stimuli, so we do not draw any strong implications of this result for language evolution.”

Reviewer: 2

(1) The authors offer no examples of what Directors actually did in their efforts to convey visual shapes gesturally or vocally, or auditory shapes vocally or with gesture. Given the attempt in this experiment to ensure that the shapes and sounds used as objects had no features that could easily be conveyed by any sort of conventionalised gestures or vocal noises, it would have been interesting to know what kinds of things the Directors did do to try to convey these objects.

R14: Reviewer 1 had a similar point (see R7) and we now discuss this at the start of section 5.

(2) Evidently, within trials, Matchers could respond to the Director with “turns”, but I think it should be clarified as to the nature of these “turns”. It does not seem to me that it is made clear what these “turns” were or what was achieved by them. Evidently the Matchers could enter into discussion with their Directors, but how they could do so needs to be clarified (this was made clear in the Fay et al. 2014 study, it should be done here also) Table 2 purported to provide data about “percentage of trials where the matcher produces at least one turn”. But why should Matchers produce any turns? What were these turns for?

R15: This is a good point. We have now added two sections in the paper: the first covers what the existing literature might predict would happen, and the second covers what happened in our experiment (see paragraph excerpt below). In addition, an example of an extended trial with interaction between director and matcher is included.

“From the conversation analysis literature, we might expect various kinds of responses from the matcher. The preferred response is to show that they understood. Otherwise, the matcher might initialise a “repair sequence” (Schegloff, Jefferson & Sacks, 1977) by signalling that they did not understand and inviting the director to work with them to resolve the issue. There are broadly three types of repair “open”, “restricted” and a “candidate understanding” (see Dingemanse et al., 2015). The matcher can initiate “open” repair by simply signalling that they don’t understand. There are various indices of non-understanding in the vocal (raising pitch to signal question, the schwa vowel from “huh?”, which may be a universal sign for misunderstanding, Dingemanse et al., 2013) and visual modalities (raised eyebrows, furrowed brow etc.). In the visual modality (at least in some signed languages), open repair can also be initialised using a “freeze look”: stopping all motion to contrast with positive backchannel feedback (Manrique, 2015). It is conceivable that similar strategies (freeze look, brow signals) may be used by our participants in the absence of an established communication system.

Restricted repair involves signalling that there is a problem with a particular part of the original signal. In spoken language this can be done using questions (e.g. A:“Sibby’s sister had a baby”, B:“Who’s sister?”), and similar strategies exist in signed languages. Restricted repairs are harder to do without an established communication system, but can be done for example by pointing at the space where (part of) a gesture was produced, or repeating (part of) a vocalisation up to the point where there was a problem. Still, we expect to see fewer examples of this kind of

repair, since it requires some separation of the signal into different parts and we expect most signals to be motivated and holistic.

Finally, matchers might offer a candidate understanding by reproducing the signal or producing an alternative signal (all most likely in combination with other signals marking the action as a repair, to distinguish it from signalling understanding) that the director can confirm or clarify. This is perhaps the most helpful response from the matcher. After repair has been initiated, the director might respond by repeating the full signal, repeating part of the signal, producing an alternative signal or confirming the candidate understanding. Various other strategies are also possible, for example the director can indicate their uncertainty about whether a matcher will understand a referent, similarly to rising intonation in spoken language. This can be done by repeating the signal, holding a gesture, lengthening or slowing the signal, and by using eye-gaze (“try-marking”, Byun et al., 2018).”

We have also included a break-down of different types of repair in our sample, as reported by Micklos, Macuch Silva & Fay (2018).

Do the authors have any comments on the different frequencies of matcher turns in the different conditions?

R16: As argued in the discussion, participants in the vocal-only condition may have interacted less due to the lack of visual contact, which hampered the exchange of any visual cues of communication problems. In the case of the gesture-only and multimodal conditions, when describing visual stimuli matchers in both conditions produced practically the same amount of responses.

(3) With regard to the so-called “multimodal” condition, as I understand this, in this condition Directors were free to use gesture or vocalization, either separately or together as they chose. However, in analyzing the results the authors simply treat “multimodal” as one of the conditions as if, in this condition, the Matchers received from their Directors both gesture and sound together. Yet the Directors were free, in this condition, to choose what modality to use and how to deploy it. Did they in fact always deploy these modalities together, or did it sometimes happen that they only used one modality, or the two modalities in succession or in alternation, rather than together? And did this make a difference? This is not made clear, but it seems to me that it should have been.

R17: The original paper addressed these points, but we have now clarified them. The reviewer is correct that the signals in the multimodal condition could be only gestural, only vocal or a combination. We categorised each turn as “unimodally vocal” (a turn with only vocal signals), “unimodally gestural” (a turn with only gestural signals), “multimodal” (a turn with both vocal and gestural signals that at least partially overlapped in time), or “unimodal mixed” (a turn with both vocal and gestural signals that did not overlap in time). Figure 6 shows the distribution of these turns (though we have fixed some problems with this figure that the reviewer points out, see below), and we discuss the results in section 5.3. There was only one case of a “unimodal mixed” signal, which we also reported. In the statistical model, we also included a factor for whether the director’s first turn was multimodal, providing a way to distinguish between the multimodal condition and the effect of actually multimodal signals. On average,

multimodal signals resulted in shorter trial times, but this was not significant. That's why in section 6.1 we argue that the advantage in the multimodal condition stemmed from the improvement in interactive resolution of problems between participants, rather than the production of more effective descriptive signals.

(4) It was also found that the speed with which the task was accomplished in each trial increased as trials progressed. Evidently the participants improved in learning how to do the tasks as they repeated them. It was found, however, that when the Director could use "multimodal" signals the task was always accomplished faster than when either only auditory or visual signals were used. The authors say that the "multimodal" condition was faster because "the combination of vocalisations and gesture boosted communication" - however, what does "boosted communication" really mean? As it stands it is a vacuous expression. How did the combination of "gesture" and "vocalization" actually help the Matcher to arrive more quickly at a decision as to which object was intended? Some analysis of how it did so and in relation to what sorts of objects it did so might have been given.

R18: We agree with the reviewer. We have re-written this section to address these points (section 6.1, page 30).

(5) I confess I have failed to understand what Fig. 6 is supposed to show. It looks as if when the Directors had visual stimuli to convey, when they were free to use whatever modality they preferred they used mostly vocalizations - which seems very odd. Furthermore, Fig. 6 compares Directors' turns with Matchers' turns. Yet the idea that the Matchers could engage in turns and why they would do so remains unclear to me.

R19: We apologise: At the last moment we decided to change to colourblind-safe colours, but the assignment of colours in the legend was incorrect. We have corrected this. The reviewer's guess is correct: directors used mainly gestures to convey visual stimuli. Please see above for our clarification about matcher turns.

(6) While one might agree that "'gesture first' theories of language evolution need to be revised in order to account for the multimodal nature of modern human communication", the experiments presented here might not seem particularly necessary or persuasive for this point of view. There are already very many very good reasons that support the view that hominin communication has always been "multimodal". One can remain doubtful that experiments of this sort can throw much light on what hominins were doing or could do, communicatively, at the dawn of any form of communication that might be regarded as somehow "linguistic" (by the way these authors seem to use "communication" interchangeably with "language" or "linguistic communication" - but of course "language" is only one of a diverse array of means of communication. Authors should not forget that "communication" is very broad in meaning and that "linguistic communication" is communication of a very special kind.).

Experiments such as those described here have their value, of course, but they have this for adding to our understanding of how different modalities may differently facilitate the transmission of different kinds of content in modern humans. They may also throw light on the ingenuity humans can display communicatively when they are put into restricted

situations of the sort devised for them in these experiments. It must be remembered that the humans that take part in these experiments are already full-fledged language users and, as such, already have at their command capacities for improvisation when a communicative situation might demand it. Is it fair to assume that our non-language using hominin ancestors were as fully equipped in this way? Perhaps some further critical discussion of how experiments of this sort are relevant to language origins issues could usefully be added.

R20: The main point of the paper was to contrast our findings with prior experiments, and we have now included further discussion of these experiments and how they have been interpreted (see also response R1 to reviewer 1, above). Our assumptions are derived from the prior experiments we are mirroring. While aware of the difficulty of using modern humans as models for earlier hominins, they, too, assumed that their results were informative for evolutionary questions. For example:

Fay, Arbib & Garrod (2013):

“Although our study was conducted among modern- day humans (with modern brains and mastery of at least one spoken language), our results may “speak” to vocal and gestural theories of the origin of language (Arbib, 2005; Cheney & Seyfarth, 2005; Corballis, 2003; MacNeilage, 1998). If one accepts that any feature that helped establish language (such as the use of motivated signs) would not have been discarded during the later evolution of the species (p. 384, Deacon, 1997), then our results suggest an important role for gesture.”

Lister, Fay, Ellison & Ohan (2015) Creating a New Communication System: Gesture has the Upper Hand. Proceedings of the Cognitive Science Society Conference: "Because modern humans already possess complex, shared language systems, we are unable to experimentally replicate the context in which language arose. However, comparing communication in the vocal and gestural modalities allows us to make inferences about the characteristics of human communication that equipped our ancestors to develop complex sign systems”.

(7) The exposition in this paper is difficult and sometimes we find vacuous or clumsy expressions. The authors seem particularly fond of the verb “bootstrap” - I think they should think carefully about what it really means and see whether, in some cases, it is being used wrongly or unnecessarily. In many cases “help” might be a better verb to use instead.

R21: We follow the usage commonly found in the experimental semiotic literature, where ‘bootstrapping a language/communication system’ is used to refer to situations in which participants are asked to create new form-meaning mappings in the laboratory without using existing natural languages. For example, “bootstrap” is used in the title of the study that we are trying to emulate (Fay et al., 2013 “How to bootstrap a communication system”). See Galantucci (2017) for a review of experimental semiotics.

Having said this, we have replaced all instances of “bootstrap” with alternatives, except for when we refer to the results of previous experiments that explicitly discuss bootstrapping.

(8) References to Ackerman et al. 2014; Irvine 2016; Bohn, Call and Tomasello 2018 are not in bibliography. I did not systematically check each in-text reference with what is listed in the bibliography and vice versa but I urge the authors to do this. There can be nothing more annoying than finding a reference that is not properly included in the bibliography and, generally speaking, one should not include items in the bibliography that are not cited in the text.

R22: Our apologies, we have added the references into the paper.

(9) Figures in my copy had no numbers. Numbers should be added to ensure they are not inserted wrongly in the printed version.

R23: We've added numbers to the file names of the figures.

Appendix C

Summary

We thank the reviewers for additional review, and respond below. In particular, we thank reviewer 3 for reviewing our statistical approach, and we are mainly in agreement with their suggestions. Responding to their questions lead to us re-writing our statistical scripts and those lead to some changes in the results. The main claims of our paper have not changed, but some exploratory results are different. We apologise for not spotting this earlier, and hope that the reviewers understand that these statistical analyses are complicated, and that we are committed to producing the best analysis we can.

The first change is about the accuracy of trials. In the raw data, it looks like people in the vocal condition are about 10 percentage points less accurate for visual stimuli, and the analysis showed a significant interaction between modality condition and stimulus type. However, the reviewer asked us to include a full random slope structure in the model. When we do this, the interaction is not significant. This suggests that, while accuracy is lower for visual stimuli in the vocal condition, the difference is not greater than might be expected by random (slope) variation between dyads and items. While we don't want to claim that there's **no difference, it's clear that we may not have the statistical power to confirm this trend. This is a minor point that does not change the central argument of our paper, which is about differences in efficiency between multimodal and gestural conditions.**

The second change is about interaction. In the original analysis, we reported that the cumulative number of matcher turns (CMT) within a pair lead to greater efficiency. While a matcher's response slowed down the current trial, it would speed up subsequent trials. However, this was an artefact of the modelling procedure. Our analysis involved building up from a null model. We had entered CMT before entering the trial number. Trial number is a lower bound on CMT, and what we really should have tested is whether CMT predicts efficiency **over and above trial number. When this is done, there is no significant effect of CMT. Indeed, there is no observable pattern when plotting the raw data. The graphs we produced for the paper are the model estimates, but basically what that shows is the effect of number of trials.**

The effect of CMT on **accuracy was tested correctly, and that effect is robust.**

We have done the following:

- **We now state that vocal-only signals for visual stimuli had a lower accuracy on average, though this difference is not greater than one would expect by random effects. We suggest that future experiments could try to confirm this.**
- **We removed figure 5 which showed the model results of the effects of CMT. They are still included in the supporting information.**
- **We have re-framed the interaction section in light of the changes to the statistics.**

In addition, we looked more closely at the trial lengths to try to explain why participants are faster in the multimodal condition.

Below we respond to the rest of the reviewer's comments:

Reviewer questions in this font.

Our responses in this font.

Reviewer: 3

The transparency of the statistical analyses could be improved. Supplemental information about the analyses were made available, but the description of the analyses in the manuscript are sometimes unclear. The authors should provide p-values for non-significant effects and full stats for significant effects (beta, t or z score, p values etc.).

We now provide all the requested stats for the significant effects (note that some beta values for the efficiency results are expressed in milliseconds rather than in model space). We also provide p-values for non-significant effects that we discuss. We feel like the supporting materials provide adequate documentation of the remaining details, and we have also added a handy summary table of results.

The authors state that effects were tested by model comparisons, but they do not describe the procedure, i.e., whether they started with a full model, testing whether the exclusion of one fixed effect led to a significant difference in the model fit. Or bottom-up whether the addition of a fixed effect to a null model led to a significant increase in the explained variance.

We have added the following to the description of the statistical procedure in the main paper:

“We assessed significance by model comparison. Starting with a null model, we add in a-priori selected control variables to produce a baseline model. Then we add each of our main fixed effects, comparing the additional improvement in model fit taking into account the additional number of model parameters using a likelihood ratio test. Comparisons between different modality conditions were obtained by fitting equivalent models with different intercepts.”

“In the supporting materials we show that the results are robust to various alternative approaches, such as using a poisson model or a more minimal fixed effects structure.”

Moreover, it is unclear how the posthoc comparisons of sublevel were computed. In my view, all this should be describe in the main part of the manuscript.

We did this by re-levelling, and have now added a note on this in the main text (see above).

In general, I would advise the authors to focus their analyses more on the tested hypotheses, i.e., to include mainly the variables of interest in their statistical models. It is always reassuring if a model explain 87% of the variance. **(we note that it correctly predicted 87% of the datapoints, which is not the same as variance explained).**

However, it would be interesting to know, which variable contributed to what extent? In particular, how much variance was explained by the main variables that are part of the hypotheses?

We now calculate pseudo-R² for each fixed effect according to Nakagawa & Schielzeth’s (2013) method. We report relevant statistics in the main text and full statistics in the supporting materials.

Nakagawa, S., Schielzeth, H. (2013) A general and simple method for obtaining R² from Generalized Linear Mixed-effects Models. *Methods in Ecology and Evolution* 4: 133–142

Some variables may not explain any variance and could therefore be excluded.

Here we note that the likelihood ratio model comparison test is an explicit test of whether a variable accounts for a significant amount of variance, and we use it to make our modelling decisions.

Including too many variables always bears the risk of overfitting. Indeed, the authors refer to a problem with the model fit on page 19, line 7

“There was more variation in the efficiency data which allowed for a more expansive statistical model.” Maybe, their models simply contain too main variables in comparison

to the number of data points. The authors should consider to include only the main variables in the results section and to refer to additional analyses including control variables in the supplementary information.

In the original analysis for accuracy, we came to the same conclusion as the reviewer. The main model for accuracy only included variables that explained a significant amount of variation in the model comparison procedure (while the SI also shows the more extensive model).

For the efficiency data, we have now included in the supporting materials a similar ‘minimal model’, which only includes variables that explained a significant amount of variation in the main model comparison procedure. The fixed effect estimates for the minimal model and the main model were correlated with $r = 0.99$, and all t-values in the minimal model were stronger than in the main model. This seems to demonstrate that the original model is not overfitted. For the efficiency data, we therefore stick with reporting the full model in the main text (since it is more conservative) and report the minimal model in addition in the SI.

Moreover, they should report which control variable was co-varying significantly with the dependent variable and they should drop all variables that have no significant influence.

The reviewer is describing a procedural approach to identifying control variables, which does not always lead to a good selection (e.g. Shrier & Platt, 2008). Instead, we chose variables which we had a-priori reason to think might lie on a back door path between the dependent and independent variables (Pearl & Mackenzie, 2018). We believe this is a more conservative and principled approach. However, we now provide additional ‘minimal’ models (see above) which fit the reviewer’s request, and we show that the results do not differ.

The disadvantage of the specified linear mixed effect models is the sparse random effect structure. I would try to fit models that include not only random intercepts, but also random slopes of the main variables.

Baayen, R. H., Davidson, D. J., & Bates, D. M. (2008). Mixed-effects modeling with crossed random effects for subjects and items. *Journal of Memory and Language*, 59(4), 390–412.

Jaeger, F. (2008). Categorical data analysis: Away from ANOVAs (transformation or not) and towards logit mixed models. *Journal of Memory and Language*, 59(4), 434–446. <https://doi.org/10.1016/j.jml.2007.11.007>

For example, if the authors would compute a generalized linear mixed effect model for accuracy (0=incorrect; 1= correct response of the matcher) on the trials level, they could maybe fit mixed effect model that includes a full random effects structure.

We note that the previous comment asked us to include fewer variables in the model to address overfitting, while the current comment asks us to include more variables in the model. This is difficult to address, especially when there is a lot of debate about the correct approach. For example, more recent advice includes a move towards simpler models rather than ‘maximal’ models (e.g. Bates, Kliegl, Vasishth, Baayen, 2015).

For efficiency, the original model already included random slopes for the main variables (modality condition and stimulus condition). Random slopes for between-unit factors are not appropriate (Baar, 2013). For this reason, we don’t include e.g. random slopes for modality condition by dyad, since each participant dyad only belongs to one modality condition. Similarly, an item only belongs to one stimuli condition.

The original model for accuracy only included random intercepts (and we take it that the reviewer’s point above mainly refers to the accuracy model). We have now included the same random slope structure for the accuracy model as for the efficiency model. The model converges (from memory, this was not originally the case, but there have been some changes to the *lme4* optimising function since our original analysis).

Our original main conclusion was that there were no significant differences by modality or stimulus type, but there was a significant interaction between modality and stimulus type (visual stimuli in the vocal condition are guessed less accurately). This is clear in the plotted data.

However, when we include random slopes, we find that the interaction is no longer significant. That is, while accuracy is lower for visual stimuli in the vocal condition, the difference is not greater than might be expected by random (slope) variation between dyads.

As a side-note, we also ran a model with all possible random slopes (slopes for dyad: stimulus condition, trial number, trial length, cumulative matcher responses; for item: modality condition, trial number, trial length, cumulative matcher responses). The estimates for the final model and the ‘full’ model are

correlated with $r = 0.99$, and significance is not qualitatively different from the simpler random slopes model described above.

Discussion

The paragraph 6.2 Evolution is missing the connection to the work conducted in the current study. The authors were asked to add this paragraph in the first review. In my view, it would be good to connect their work to the work they describe in this paragraph even if their results do not allow strong conclusions.

We are a little confused by the comment, since this paragraph begins by describing our results, and then stating how they affect theories of language evolution. We note that this paragraph has always been part of the manuscript (not added in the last revision), perhaps there is some mistake in the reviewer's reference? In any case, we have added at the end of the paragraph:

"Experimental methods, such as the one used in this study, may help us sharpen these theories."

Minor points

Improve figure reference in the text.

We have moved the figure references to more prominent positions.

Figure 6, caption is unclear, no reference to the meaning of the different colors.

We have clarified:

Figure 6. The distribution of turns produced in the multimodal signalling condition by stimulus type and signal type. On the left, turns produced by directors (n=671); on the right, turns produced by matchers (n=67). Unimodal gestural turns are shown in green, unimodal vocal turns in purple, and multimodal turns in orange. Note: there was one turn produced by a director in the visual stimuli condition which consisted of a unimodal vocal signal and a unimodal gestural signal. This turn is not shown here.

The authors reported that the variable trial length milliseconds was log transformed, but they did not give a reason for it and they did not state the distribution used in the linear mixed effect model (Gaussian, poisson, binomial etc).

We used log trial length to transform the distribution to gaussian. We have added these details into the text. In the SI, we now include a model where we fit the raw milliseconds using a poisson distribution, and the model estimates for the final

model are highly correlated ($r = 0.97$) and all significant variables remain significant, suggesting that this choice is not important.

Page 23, line 32, indicating that trial length decreased non-linearly towards a plateau (rephrase sentences)

Rephrased to “trial length decreased rapidly with each trial at first, then more slowly towards a steady minimum”

Page 23, line 35, what do the authors mean with a quadratic effect?

When this is first mentioned, we added a gloss (“non-linear”) and we have described the effect: “participants in the multimodal condition improving their efficiency at a greater rate than those in the unimodal conditions”.

Page 27, line 8-10, rephrase sentences

Simplified to “For example, in multimodal signals, the vocal component often started later and ended earlier than the visual component.”.

Page 27, line 19-23, in the summary of the paragraph I would add that the modality of the stimulus plays an important role for the modality chosen for communication

We have added:

“We note that the communicative success for all stimuli was well above chance, and that there was no main effect of stimulus modality for efficiency. In this sense, the stimuli seem to be similarly difficult to communicate, even if the modality of the stimulus may strongly bias which modality is used in communication.”

Page 27, line 58, I am not sure whether encoding is the correct word in this context

We have changed this to “signalling”.

In previous experiments, stimuli consisted of items for which there existed conventional signals while in the current experiment stimuli were not associated with conventional labels and contained both stimuli that were suitable for visual signaling and stimuli suitable for vocal signaling.

Page 28, lines 32-34, in my view the first sentence implies that real-time interaction was only possible or present in the multi-modal condition. I am not sure if the authors intended to make this point here.

We did not mean to make this point, and have added:

“We suggest that real-time interaction, which happened almost exclusively in the gestural and multimodal conditions, was a critical factor in the improvement of communication during the experiment, based on three observations.”

Page 28, line 54 “This is an addition...” until the end of the paragraph. In my view, the authors should skip this part, I do not see this strong link between their findings and the findings by Dingemanse et al (2015).

We have modified the entire passage and have removed the extended discussion that the reviewer points to.

Reviewer: 4

Comments to the Author(s)

I do have a series of questions/issues: these are just small exercises in fine-tuning, and I would not expect them to take more than a day or two to do.

Minor Points

1. “There is increasing evidence for flexible use of vocal calls in primates (e.g. Seyfarth & Cheney, 2010; Slocombe, Zuberbühler, 2005; Seyfarth & Cheney, 2017), and research showing that it may be easier to recognise and treat vocalisations as symbolic signals compared to gestures (DeLoache, 2002, 2004; Irvine, 2016).”

Perhaps with the exception of the experiments on chimp alarm calling (and even here some elements of the alarm response seemed insensitive to context), my impression was that most of this flexibility was flexibility in response. Does comparative data support the view that there was sufficient topdown control of vocalization for vocalization to play an important role from the very begins of the transition to language? And what is the evidence of the ability to form new signals?

Reviewer 1 asked a similar question, and we have included this response in the manuscript (Pearlman, 2017 discusses this point in reference to our question):

“In general, signals used in important contexts such as warning or mating are often multimodal in order to be robust (Ratcliffe & Nydam, 2008; Bushman, 1999; Rigaiil et al., 2013; Dalziell et al., 2013; Ota, Gahr & Soma, 2015). In this light it seems quite plausible that early human communication was inherently multimodal in early stages of evolution, just as it is today. Of course, there are questions about whether the ability to *control* vocal signals and *learn* them was present for our ancestors before the emergence of symbolic language (see e.g. Arbib, 2005a). However, Perlman (2017) argues that there is at least some

evidence of vocal learning in non-human primates, and that this might plausibly allow a fully multimodal system right from the start (see also Kendon, 2017, Fröhlich et al., 2019)."

2. "Although it is clear that the behaviour of modern, linguistic, enculturated humans cannot provide direct evidence for the behaviour of early humans, they do provide a useful model and these experimental results have been used to support theories of language evolution. For example, Fay et al. (2013, 2014) made two inferences (see Goldin-Meadow, 2017): first, that gesture has more potential for motivated signs (iconicity, indexicality) than vocalisations; and second that iconic signals can help bootstrap a communication system."

I agree. But there should be some, though perhaps brief, consideration of the obvious problem: these experimental agents have fully modern theories of mind; they have had a huge amount of experience communicating, and hence have trained expectations about what others communicate about; and if language has resulted in any distinctive adaptations for language and communication, these agents have them.

We feel this point is clear, but have added the following brief note:

"It is clear that the behaviour of modern, linguistic, enculturated humans cannot provide direct evidence for the behaviour of early humans (modern humans have more developed theory of mind, experience with pragmatic communication and possibly other adaptations for linguistic communication). However, they do provide a useful model and these experimental results ..."

3. "create motivated signals". Perhaps a sentence of explanation on what a motivated signal is.

We have clarified this as "signals which are grounded perceptually or contextually as opposed to being abstract or arbitrary (see Lister, et al., 2015; Perlman et al., 2018)."

4. "and the main effects discussed above should not be driven by random differences between participant's abilities (e.g. some might put more effort into their initial descriptions while others rely more on interaction to converge on the correct meaning)". Perhaps a bit more needs to be said here, for it does not strike me as obvious that individual differences will be minor. Individual histories might play a significant role here: for example, for some people, charades-type games form an important part of their social life, and for others, not at all. It strikes me as possible that a multi-modal system

at a communal level might emerge because of variation in individual defaults across the modalities.

This sentence was poorly phrased by us: we meant that the main effects should be robust to controls for random differences between individuals. We do control for individual differences. We have rephrased as:

“and the model should also control for random differences between participant’s abilities (e.g. some might put more effort into their initial descriptions while others rely more on interaction to converge on the correct meaning)”

5. “A trial could be as short as the director issuing a description followed by the matcher acknowledging understanding”

Am I right in thinking the director has no information about which three stimuli the matcher has to choose between? If so, one might wonder whether that decreases the ecological validity of the experiment: agents often know that they have to communicate X rather than Y or Z, and so have to focus on the most salient differences between X and Y (and X and Z)

Indeed, directors do not know during their turn which options are available to matchers. However, over the course of many trials and games, participants develop systems which do focus on relevant structural/ perceptual distinctions between the individual stimuli of a stimulus set. Some of the stimuli are easily confusable with others, thus an effective communicative strategy is to focus on minimal differences between similar stimuli (and the matchers can always ask for clarification between various options). We briefly discuss the role of context in our literature review, covering recent experimental work which has specifically addressed the effects of context. E.g. Tinits et al. (2018) has shown that the degree of openness of the referential context impacts the emergence of systematic structure in evolving communication systems, systematic strategies developing more in open and unstable environments but also in environments where referents are not available to matchers in the moment of communication. Similarly, Winters et al. (2018) have shown that signals evolve differently depending on the predictability of the referential contexts, with more predictable contexts leading to communication systems which are more dependent on context to reduce uncertainty about the intended meanings. We also discuss common ground and context at the end of section 6.1.

However, these effects are not central to our argument. It is not clear to us how the director’s ability to see the context might provide an alternative explanation

for the differences we find between conditions. Moreover, we wanted to avoid tempting the participants to use spatial solutions, or complicate the initial signal by encouraging the director to refer to multiple meanings etc.

6. I shall assume that the statistical procedures were sound, as I have no expertise in that area.

Reviewer 3 has provided some thorough review of the statistics.

7. “Directors using vocalisations to describe auditory stimuli almost universally attempted to mimic the target stimuli. It is reasonably complex to produce sounds with human vocal chords to match other sounds, but a simple task for the listener to identify the meaning.” This is quite important, as it suggests a strong reliance on iconicity as a strategy, and that has quite restricted utility in the vocal modality (and the gestural, but somewhat less so).

We agree with the reviewer that participants were using iconic strategies, but are unsure what they are asking us to do. The reviewer asserts that iconicity has restricted utility in the vocal modality compared to the gestural modality, but this is what the experiment aims to test (and indeed, participants in the vocal condition describe auditory stimuli with high accuracy and comparable efficiency to gesturers). Our point here was more to highlight the imbalance between production and comprehension, not necessarily between modalities.

8. “In order to look more closely at descriptions of auditory stimuli by directors in the multimodal condition, Fig. 7 shows the ratio between the length of the visual component and the length of the auditory component. The visual component of the multimodal signal is often longer than the auditory component.”

This needs a little more explanation: why is this ratio important, especially to the evolutionary questions that frame the experimental work?

The ratio indicates that even when communicating about sounds participants who are able to use both gestures and vocalisations choose to employ gestural signaling extensively in their communication. In fact, gestural signaling is more pronounced both temporally and in terms of its relative frequency to the vocal components of multimodal signals, which suggests that gestures are the primary motor in participants’ multimodal signaling. This provides some support to the model by Levinson and Holler, which highlights the centrality of gestural

signaling in the early stages of evolution of human language. We have added a few sentences clarifying this.

9. “We suggest that real-time interaction was a critical factor in the improvement of communication during the experiment, based on three observations. The first observation is that the rate of improvement for efficiency was faster in the multimodal condition (at least for visual stimuli). This suggests that there was a cumulative effect over several rounds of interaction.”

This links back to my earlier reservation about the extent to which these experiments can illuminate the early evolution of language. The directors and the matchers are all experts in communicative interaction. But our ancestors forging early forms of language were not.

Our statement above is about the results of the experiment, and we take it that the reviewer is not questioning the difference between our conditions (all participants were experts in communicative interaction, but the multimodal participants still improved faster). The only conclusion we make about cumulative interaction and evolution is “we expect the constraints and affordances of interaction to be an important part of the story of language evolution”. We don’t make claims about early human interactional abilities, only that their abilities (whatever they might be) would affect the process of language evolution. And it seems too extreme to suggest that early humans would have had *no* abilities in communicative interaction, since many non-human primates exhibit some basic interactional abilities (Fröhlich et al., 2016; Levinson, 2016; Pika et al., 2018; Rossano, 2018). We take it that the reviewer would like us to simply raise this concern, so we have added:

“Of course, the role of interaction may be constrained by the interactive abilities of early human ancestors, though comparative work suggests that at least some basic abilities might have been in place (Fröhlich et al., 2016; Levinson, 2016; Pika et al., 2018; Rossano, 2018).”

Fröhlich, M., Kuchenbuch, P., Müller, G., Fruth, B., Furuichi, T., Wittig, R. M., & Pika, S. (2016). Unpeeling the layers of language: Bonobos and chimpanzees engage in cooperative turn-taking sequences. *Scientific reports*, 6, 25887.

Levinson, S. C. (2016). Turn-taking in human communication—origins and implications for language processing. *Trends in cognitive sciences*, 20(1), 6-14.

Pika, S., Wilkinson, R., Kendrick, K. H., & Vernes, S. C. (2018). Taking turns: bridging the gap between human and animal communication. *Proceedings of the Royal Society B: Biological Sciences*, 285(1880), 20180598.

Rossano, F. (2018). Social manipulation, turn-taking and cooperation in apes. *Interaction Studies*, 19(1-2), 151-166.

Fröhlich, M., Sievers, C., Townsend, S. W., Gruber, T., & van Schaik, C. P. (2019). Multimodal communication and language origins: integrating gestures and vocalizations. *Biological Reviews*.

Appendix D

Response to reviewers

Editor/Reviewer comments in regular font.

Author response in bold font.

- Ethics statement
- Data accessibility
- Competing interests
- Authors' contributions
- Acknowledgements
- Funding statement

We have included all of these sections.

Reviewer comments to Author:

Reviewer: 4

The responses to my earlier minor queries are all fine, and as this is an interesting paper on an important issue, I think it should now be published. I think the authors are probably right that a multi-modal view of early language is superior to any unimodal view. One very minor point:

Is this a typo on page 10; "This we attribute to having the possibility to draw on the two modalities flexibly, using sometimes multimodal and sometimes multimodal signals, and using this flexibility for negotiating meaning in interaction", with the repeat of "multimodal"

We have corrected this.

Reviewer: 3

Comments to the Author(s)

Abstract

1) The results suggest that even in the absence of conventional signals, gesture is a powerful mode of communication, but that there are also advantages to multimodality.

->what did they authors intend to express?

1) Do gestures have advantages to auditory signal?

- 2) or do gestures have advantages to multimodal signals?
- 3) or do multimodal signals have advantages to gestural signals?

We have clarified this:

“The results suggest that even in the absence of conventional signals, gesture is a powerful mode of communication compared with vocalisation, but that there are also advantages to multimodality compared with using gesture alone.”

Introduction

- 2) Pg. 3. L. 30, “...use rigorous statistical methods”
In my view, the authors have not yet applied rigorous statistical methods. See comments below, e.g. about multiple comparison correction for sublevel comparison.

We address these points below.

- 3) Pg. 6 L.58 “For example, gestures enhance neural processing of speech comprehension especially in noisy contexts...”
..., guide visual-spatial attention in healthy and clinical populations (Beattie, Webster, & Ross, 2010; Preisig et al., 2015). Moreover, they help patients with language disorders to express themselves (Hogrefe et al., 2012; van Nispen et al., 2016)

We thank the reviewer for these extra points and have added them to the text.

- 4) Pg. 11 L. 38-50. “In principle, we could find that participants in this condition were more efficient than those in the other conditions without any multimodal signals (simultaneous vocal and gestural signals) being produced. In fact, although participants did produce multimodal signals, we do not find a multimodal advantage in the sense that multimodal signals predict higher efficiency or accuracy.”
These sentences are contradicting each other.

We have rephrased the whole paragraph to make clearer what we mean:

“Participants in the multimodal condition might gain an advantage for two reasons. Firstly, multimodal signals might be more effective than unimodal signals. A second kind of advantage might be the flexibility to draw on these two modalities to produce unimodal visual or unimodal vocal signals as required. We find that participants do use multimodal signals, but we do not find a multimodal advantage in the first sense that using a multimodal signal predicts greater efficiency or accuracy for a given trial. We do find an advantage of the multimodal condition as a whole in bootstrapping a communication system, which we attribute to an advantage in flexibly deploying modalities.”

Results

5) Pg 24, L. 35-41 “However, for every trial where a matcher responded, subsequent guesses were more likely to be correct (10 trials where the matcher responded raised the probability of a correct guess in subsequent trials by about 4 percentage points, $\beta = 0.12$, $z = 2.39$, $\chi^2(1) = 7.11$, $p < .01$, $\text{pseudo-R}^2 = 0.003$)”
What was the variable/fixed factor tested?

This is the “cumulative number of matcher responses”, which we describe in the paragraph above. We have clarified this in the sentence:

“However, for every trial where a matcher responded, subsequent guesses were more likely to be correct (10 trials where the matcher responded raised the probability of a correct guess in subsequent trials by about 4 percentage points, fixed effect of cumulative number of matcher responses, $\beta = 0.12$, $z = 2.39$, $\chi^2(1) = 7.11$, $p < .01$, $\text{pseudo-R}^2 = 0.003$.)”

6) Pg 25, l. 10-16 ...”, which is explained by two separate trends (see Fig. 3). For auditory stimuli, participants in the multimodal condition communicated more efficiently than participants in both the vocal and gestural conditions. For visual stimuli, participants in the vocal condition communicated more efficiently than participants in the gestural and multimodal conditions (though they were also less accurate on average, see section 5.1”

Please provide the p-values and the statistical test scores used for the pairwise comparison!

We have included the model results:

“For auditory stimuli, participants in the multimodal condition communicated more efficiently than participants in both the vocal ($\beta = .69$, $t = 4.4$, Satterthwaite $p < .001$) and gestural conditions ($\beta = .25$, $t = 1.7$, Satterthwaite $p = .1$). For visual stimuli, participants in the vocal condition communicated more efficiently than participants in the gestural ($\beta = .44$, $t = 3.0$, Satterthwaite $p = .009$) and multimodal conditions ($\beta = .69$, $t = 4.4$, Satterthwaite $p < .001$.)”

Note that we are concerned with explaining the significance of the interaction between stimulus type and modality condition, not the individual comparisons for each modality condition. The main inferential statistic for this interaction is a comparison between models, not within a model.

We prefer to rely on comparisons between models rather than p-values derived from the Satterthwaite approximation for comparisons within a model. But we have included them at the reviewer’s request.

Note that the results above relate to the simple interaction between modality condition and stimulus type. This is not the same as one of our most important

results: that participants in the multimodal condition end up more efficient than the gestural condition *by the end of the experiment* (interaction with trial number).

7) Pg. 24, l. 35 Longer trials were guessed less accurately ($\beta = -1.02$, $z = -6.43$, $\chi^2(1) = 38.97$, $p < .001$, pseudo- $R^2 = 0.05$).

Pg. 25 l. 39. Trials were longer if the guess was incorrect (by about 2.5 seconds, $t = 2.93$, $\chi^2(1) = 13.2$, $p < .001$, pseudo- $R^2 = 0.039$)

These results are redundant.

These results are in agreement, but come from different models. They were included as control variables to avoid possibilities such as participants being more accurate just because they took more time to communicate. In principle they might be different, e.g. trial length could completely determine accuracy, but many factors might determine trial length. We include them for completeness.

8) Pg. 26, l.12-50

This paragraph needs to be connected to linked to the statistical analyses (pg 25, l. 19-35?). Usually, according to the APA standards, descriptive stats are provided before inferential statistics.

This comment concerns the description of Figure 4. This is intended as an explorative visualisation of the data, rather than a confirmatory test of an a priori hypothesis, and is clearly a post-hoc analysis. We were using it to explore the nature of the multimodal advantage. We think it is clear from the visualisation that there are different kinds of advantage for each combination of signal and stimulus type. Therefore, we suggest that any multimodal advantage in efficiency comes from the ability to flexibly deploy signals, rather from an inherent advantage of multimodal signals per se.

To address the reviewer's concern, we have added statistics to support each claim in the main text. While this might be done with a multivariate multiple regression, this becomes complicated due to many contrasts having no data and there being no simple implementation in a mixed modelling framework. Therefore, we have simply performed t-tests for each claim, adjusting the p-value for the number of comparisons. The full methods and statistics are included in the supporting materials.

Discussion

9) Pg. 29, l. 28-33, The authors state that "This finding is in line with the multimodal advantage hypothesis"

In my view, this conclusion cannot be drawn, because there was no difference between the vocal condition and the multimodal condition.

We have clarified that this statement refers to a comparison between the multimodal condition and the gestural condition, not between the vocal and multimodal condition:

“That is, in contrast to previous studies, we do find that there is evidence of an advantage to be able to choose to communicate multimodally over being restricted to just the gestural modality. This result is in line with the multimodal advantage hypothesis rather than the gestural advantage hypothesis or the direct linkage hypothesis.”

10) Pg. 31. L. 29 “We find ..., and that vocalization is a weaker strategy in communication when used on its own ...”

This interpretation is not supported by the presented data: the authors did not find a main effect modality for the variable accuracy, nor a main effect of modality condition for the variable efficiency.

This quote is comparing vocalisation-only to multimodality (“vocalisation is a weaker strategy for communication when used on its own, but it gains power when combined with gesture”). This conclusion is built from many different results. For example, lower efficiency when describing auditory stimuli, slightly lower accuracy when describing visual stimuli, vocal signals being temporally embedded within visual signals etc. We believe our conclusion is justified. We have added a note to the text to make it clear which results our conclusions are based on.

11) Pg. 32. L24-25,
..., and was generally enhanced in the auditory condition

The context of this sentence is in the “future work” section: “We suggest that interaction may play a role, and that experiments similar to this one can help researchers think about this. We make three observations that may be explored further in the future. The first observation is that efficiency improved at a greater rate in the multimodal condition than in the gestural condition (at least for visual stimuli). This suggests that part of the advantage of multimodality is a cumulative effect over many rounds of interaction.”

We take it that the reviewer would like us to add the qualification to the last-but-one sentence. While we acknowledge that there was a difference between stimuli types, this is not the point we are trying to draw attention to. Our point is to compare the multimodal and gestural conditions and suggest that part of

the difference is cumulative over rounds of play, as we clearly state in the quote above.

Response Letter

12) We have also added a handy summary table of results
Table reference?

We decided to take the table out since it over-simplified the results. We should have amended the response letter.

13) Comparisons between different modality conditions were obtained by fitting equivalent models with different intercepts.”

We did this by re-leveling, and have now added a note on this in the main text
Multiple comparison correction?

Releveling is generally not considered comparing multiple different hypotheses. It does not involve multiple independent tests. The re-leveled model is mathematically identical to the original, but the estimates are expressed according to different relations between the variable levels. This helped us understand the interaction effect, but the main statistical tests are based on comparisons between models, not on comparisons between levels within a model. In fact, we did not report any statistics for comparison between levels in the previous version.

We note that, even if it were suitable, adjusting for 2 comparisons would not qualitatively change the significance of any of the tests.

We have clarified that we mean “releveling” in the text (page 19), but have not changed the statistics.

14) However, we now provide additional ‘minimal’ models (see above) which fit the reviewer’s request, and we show that the results do not differ.

Actually, the provided minimal model does not fit my request.

Could the authors please test two separate models, including only the fixed factors modality (visual vs. auditory) and stimulus type (gestural, vocal, multimodal). These are the main variables the authors based their hypotheses on.

(we assume that the reviewer means “modality (*gestural, vocal, multimodal*) and stimulus type (*visual vs. auditory*)”)

1) A generalized linear mixed effect model (glmer, lme4), fixed effects modality and stimulus type, including random slopes and intercepts for both fixed effects, for the dependent variable accuracy (0=incorrect, 1=correct trial)

2) A linear mixed effect model (*lmer*, *lme4*), fixed effects modality and stimulus type, including random slopes and intercepts for both fixed effects, for the dependent variable trial length

Please provide the betas, Z-score/T-score, SEM and p-values (for *lmer*, p-values will have to be estimated in a separate step)

The context of this request, from the previous review, is the reviewer's concern that the model is over-parametarised: "I would advise the authors to focus their analyses more on the tested hypotheses, i.e., to include mainly the variables of interest in their statistical models. ... Maybe, their models simply contain too main variables in comparison to the number of data points. The authors should consider to include only the main variables in the results section and to refer to additional analyses including control variables in the supplementary information."

We responded by providing a model that only included variables that explained significant variance. The reviewer has now clarified that they wanted a model with only fixed effects for modality and stimulus type. While this would make the results section of the main paper simpler, we argue that it would miss out on important variation and possible confounds. For example, the effect of trial number is clear and an important part of our argument. We have a clear statement of the expected effects in our methods section.

In addition, we note that the existing analysis does not find significant main effects for either modality nor stimulus type, and we would not expect models with only these factors to yield significant results, either. It is unclear to us how the results of the models above provide a confound for our conclusions, and in fact may not be conservative enough (as we show below).

Below we provide the models that the reviewer requested. We assume that the reviewer was requesting random slopes for dyad, player and item (as in the main analysis). As in the previous response, we note that some random slopes are not appropriate: "Random slopes for between-unit factors are not appropriate (Baar, 2013). For this reason, we don't include e.g. random slopes for modality condition by dyad, since each participant dyad only belongs to one modality condition. Similarly, an item only belongs to one stimuli condition." All our models test the maximum random effect structure permissible.

We ran the models that the reviewer requested. We note that we prefer model comparison tests to p-values for within-model effects, but have presented them here using the Satterthwaite approximation (in package *lmerTest*).

For accuracy, the main effects of modality condition and stimulus type are not significant (as in the main analysis).

Fixed effects:

	Estimate	Std. Error	z value	Pr(> z)
(Intercept)	3.3105	0.5175	6.397	1.58e-10 ***
modalityConditionvisual	-0.2072	0.4357	-0.476	0.634
modalityConditionvocal	-0.7789	0.4059	-1.919	0.055 .
conditionVisual	-1.0290	0.5974	-1.722	0.085 .

(in fact, they appear marginal, while our main model suggests they are clearly not significant, thus underlining the fact that the other factors included in the fuller models we present contribute to the variance in our data)

For trial length, the effects are not significant (as in the main analysis):

Fixed effects:

	Estimate	Std. Error	df	t value	Pr(> t)
(Intercept)	-0.17692	0.14180	21.25648	-1.248	0.226
modalityConditionvisual	0.27891	0.16887	15.15718	1.652	0.119
modalityConditionvocal	0.11988	0.17032	15.64188	0.704	0.492
conditionVisual	0.08168	0.11763	25.06719	0.694	0.494

We note that this model results in convergence warnings and worse outlying residuals than the main model.

It is unclear whether the reviewer wants us to include the interaction between modality and stimulus type. It is clearly motivated by our hypothesis. When including it, there are significant interaction effects:

Fixed effects:

	Estimate	Std. Error	df	t value	Pr(> t)
(Intercept)	-0.3365	0.1411	20.7431	-2.385	0.02672 *
modalityConditionvisual	0.3922	0.1816	15.2947	2.160	0.04706 *
modalityConditionvocal	0.4596	0.1836	15.9015	2.503	0.02360 *
conditionVisual	0.3557	0.1152	23.4330	3.087	0.00514 **
modalityConditionvisual:conditionVisual	-0.1892	0.1135	16.1406	-1.668	0.11462
modalityConditionvocal:conditionVisual	-0.5823	0.1198	17.3306	-4.858	0.00014 ***

However, the main effects now appear to be significant. Following the reviewer's advice, we should claim that there is a main effect of modality condition and stimulus type. However, we argue that these are spurious correlations that appear because key confounding factors are not controlled for. Our current method in the main analysis is designed to implement these controls. They result in a more complicated picture, but one that fits the data better and is more conservative in its conclusions.

Therefore, we have not included these models in the final paper.

Appendix E

Author response for “Multimodality and the origin of a novel communication system in face-to-face interaction”

We thank the reviewer and editor for their feedback and respond below. Based on some additional feedback, we have made some other minor clarifications to which can be seen in the track-changes document.

Reviewer text in this font.

Our response in this font.

Response Letter

I thank the authors for the work they made on the manuscript and for the additional information they provided. They addressed the majority of my concerns.

Minor points

7) Ok, please make this transparent for the reader

This relates to the apparent redundancy of including accuracy as an independent variable in the efficiency model and efficiency as an independent variable in the accuracy model. We have included the text of our response into the relevant sections of the manuscript:

Page 19 (description of accuracy model):

The fixed factors included: ... trial length (to control for participants being more accurate just because they took more time to communicate).

Page 20 (description of efficiency model):

“Note that accuracy was included in the efficiency model and that efficiency was included in the accuracy model. We expect the effects to agree.”

Page 26 (results for efficiency):

“The latter results agrees with the effect of trial length in the model of accuracy.”

8) Method for multiple comparison correction/p-adjustment?

We used Bonferroni correction. We have specified this in the manuscript (results section page 27), and we have added a longer note to the relevant place in the supporting materials.

10) If the conclusion is drawn in this context, it should be made transparent for the reader.

This relates to a sentence “We find ..., that vocalization is a weaker strategy in communication when used on its own ...”, and we previously responded that this conclusion was built from many different results. We have added our previous response into the text (page 33):

"We also find that vocalisation is a weaker strategy for communication when used on its own (based on several factors e.g. lower efficiency when describing auditory stimuli,

slightly lower accuracy when describing visual stimuli, participants in the multimodal condition rarely chose to produce vocal-only signals). The fact that vocal signals were frequently temporally embedded within visual signals may be considered as further corroborating the notion that non-conventionalised vocal signals are often secondary to non-conventionalised visual signals. However, in our data, there is a clear advantage to combining vocalisations with gesture."

**We have also added this information to the summary section (page 29-30):
"There was some evidence that vocal signals were weak when used on their own. For example: vocal-only participants were less efficient than multimodal participants when describing auditory stimuli; vocal-only participants had slightly lower accuracy when describing visual stimuli; and participants in the multimodal condition rarely chose to produce vocal-only signals. Vocal-only participants describing visual stimuli had shorter trial times, but were also slightly less accurate and did not interact much. The fact that vocal signals were frequently temporally embedded within visual signals may be considered as further corroborating the notion that vocal signals are secondary to visual signals."**

Abstract

1.46-49

"The results suggest that even in the absence of conventional signals, gesture is a powerful mode of communication compared with vocalisation, but that there are also advantages to multimodality compared with using gesture alone."

->Do the authors infer that gesture is the more powerful mode of communication than non-linguistic vocalizations? Or do they mean that gestures are as powerful as vocalisations in this context? Please clarify.

Apologies, we now understand the source of the confusion. We have changed this to:

The results suggest that even in the absence of conventional signals, gesture is a **more powerful mode of communication compared to vocalisation, but that there are also advantages to multimodality compared to using gesture alone.**

Results

-Please report β , z, chi-square for all fixed effects tested in the models accuracy and efficiency

The easiest way to report these in the main manuscript while reducing the number of stats in the text (as also requested by the reviewers) is to put them into a table. We've added a table for the results of the accuracy model and a table for the results of the efficiency model. This allows us to take out some of the stats from the text. These details are also available in the supplementary materials.

-Pg. 23, l. 39-43

“For auditory stimuli, participants in the multimodal condition communicated more efficiently than participants in both the vocal ($\beta=.69$, $t=4.4$, Satterthwaite $p<.001$) and gestural conditions ($\beta=.25$, $t=1.7$, Satterthwaite $p=.1$)”

->I would not consider $p<.001$ as a statistical trend. According to my understanding, a statistical trend is considered as a p-value in-between 5% and 10%

We agree with this definition. The quote is part of the explanation of an interaction effect. The original text is:

“There was no main effect of modality condition nor stimulus type, but there was a significant interaction between the two, which is explained by two separate trends (see Fig. 3). For auditory stimuli, participants in the multimodal condition communicated more efficiently than participants in both the vocal ($\beta=.69$, $t=4.4$, Satterthwaite $p<.001$) and gestural conditions ($\beta=.25$, $t=1.7$, Satterthwaite $p=.1$). For visual stimuli, participants in the vocal condition communicated more efficiently than participants in the gestural ($\beta=.44$, $t=3.0$, Satterthwaite $p=.009$) and multimodal conditions ($\beta = .69$, $t = 4.4$, Satterthwaite $p<.001$), though they were also less accurate on average (see section 5.1).”

We now see that this was unclear. We wanted to explain the interaction effect without making a claim about the differences within stimulus types (there is no main effect of stimulus nor modality condition). But “trend” was the wrong word, so we’ve changed this to “patterns”, and made the scope of the claim more clear:

“... which is explained by **two patterns (see Fig. 3). **Firstly**, for auditory stimuli, participants in the multimodal condition communicated more efficiently than participants in both the vocal ($\beta=.69$, $t=4.4$, Satterthwaite $p<.001$) and gestural conditions ($\beta=.25$, $t=1.7$, Satterthwaite $p=.1$). **Secondly**, for visual stimuli, ...”**

-Pg. 23 l. 59 and pg. 24 p-adjusted – what was the correction applied? (see comment response letter)

We used Bonferroni correction. We have specified this in the manuscript (results section page 27), and we have added a longer note to the relevant place in the supporting materials. In particular, we have updated the summary at the end of the results section (page 29) to more clearly convey the findings.